# TabR: Tabular Deep Learning Meets Nearest Neighbors

**Yury Gorishniy**[*†]  **Ivan Rubachev**[‡†]  **Nikolay Kartashev**[‡†]

**Daniil Shlenskii**[†]  **Akim Kotelnikov**[‡†]  **Artem Babenko**[†‡]

## Abstract

Deep learning (DL) models for tabular data problems (e.g. classification, regression) are currently receiving increasingly more attention from researchers. However, despite the recent efforts, the non-DL algorithms based on gradient-boosted decision trees (GBDT) remain a strong go-to solution for these problems. One of the research directions aimed at improving the position of tabular DL involves designing so-called retrieval-augmented models. For a target object, such models retrieve other objects (e.g. the nearest neighbors) from the available training data and use their features and labels to make a better prediction.

In this work, we present TabR – essentially, a feed-forward network with a custom k-Nearest-Neighbors-like component in the middle. On a set of public benchmarks with datasets up to several million objects, TabR marks a big step forward for tabular DL: it demonstrates the best average performance among tabular DL models, becomes the new state-of-the-art on several datasets, and even outperforms GBDT models on the recently proposed "GBDT-friendly" benchmark (see Figure 1). Among the important findings and technical details powering TabR, the main ones lie in the attention-like mechanism that is responsible for retrieving the nearest neighbors and extracting valuable signal from them. In addition to the higher performance, TabR is simple and significantly more efficient compared to prior retrieval-based tabular DL models. The source code is published: link.

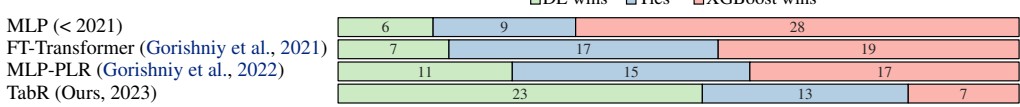

Figure 1: Comparing DL models with XGBoost (Chen and Guestrin, 2016) on 43 regression and classification tasks of middle scale ($\leq 50K$ objects) from "Why do tree-based models still outperform deep learning on typical tabular data?" by Grinsztajn et al. (2022). TabR marks a significant step forward compared to prior tabular DL models and continues the positive trend for the field.

## 1 Introduction

Machine learning (ML) problems on tabular data, where objects are described by a set of heterogeneous features, are ubiquitous in industrial applications in medicine, finance, manufacturing, and other fields. Historically, for these tasks, the models based on gradient-boosted decision trees (GBDT) have been a go-to solution for a long time. However, lately, tabular deep learning (DL) models have been receiving increasingly more attention, and they are becoming more competitive (Klambauer et al., 2017; Popov et al., 2020; Wang et al., 2020; Hazimeh et al., 2020; Huang et al., 2020; Gorishniy et al., 2021; Somepalli et al., 2021; Kossen et al., 2021; Gorishniy et al., 2022).

In particular, several attempts to design a retrieval-augmented tabular DL model have been recently made (Somepalli et al., 2021; Qin et al., 2021; Kossen et al., 2021). For a target object, a retrieval-augmented model retrieves additional objects from the training set (e.g. the target object's nearest

---

[*]Corresponding author: `firstnamelastname@gmail.com`    [†]Yandex    [‡]HSE

neighbors, or even the whole training set) and uses them to improve the prediction for the target object. In fact, the retrieval technique is widely popular in other domains, including natural language processing (Das et al., 2021; Wang et al., 2022; Izacard et al., 2022), computer vision (Jia et al., 2021; Iscen et al., 2022; Long et al., 2022), CTR prediction (Qin et al., 2020; 2021; Du et al., 2022), and others. Compared to purely parametric (i.e. retrieval-free) models, the retrieval-based ones can achieve higher performance and also exhibit several practically important properties, such as the ability for incremental learning and better robustness (Das et al., 2021; Jia et al., 2021).

While multiple retrieval-augmented models for tabular data problems exist, in our experiments, we show that they provide if only minor benefits over the properly tuned multilayer perceptron (MLP; the simplest parametric model), while being significantly more complex and costly. Nevertheless, in this work, we show that, with certain previously overlooked design aspects in mind, it is possible to obtain a retrieval-based tabular architecture that is powerful, simple and substantially more efficient than prior retrieval-based models. We summarize our main contributions as follows:

1. We design TabR – a simple retrieval-augmented tabular DL model which, on a set of public benchmarks, demonstrates the best average performance among DL models, achieves the new state-of-the-art on several datasets and is significantly more efficient than prior deep retrieval-based tabular models.
2. In particular, TabR achieves a notable milestone for tabular DL by outperforming GBDT on the recently proposed benchmark with middle-scale tasks (Grinsztajn et al., 2022), which was originally used to illustrate the superiority of decision-tree-based models over DL models. Tree-based models, in turn, remain a more efficient solution.
3. We highlight the important degrees of freedom of the attention mechanism (the often used module in retrieval-based models) that allow designing better retrieval-based tabular models.

## 2 RELATED WORK

**Gradient boosted decision trees (GBDT)**. GBDT-based ML models are non-DL solutions for supervised problems on tabular data that are popular within the community due to their strong performance and high efficiency. By employing the modern DL building blocks and, in particular, the retrieval technique, our new model successfully competes with GBDT and, in particular, demonstrates that DL models can be superior on non-big data by outperforming GBDT on the recently proposed benchmark with small-to-middle scale tasks (Grinsztajn et al., 2022).

**Tabular deep learning.** Tabular DL is a rapidly developing field with the recent advances covering parametric architectures (Klambauer et al., 2017; Wang et al., 2020; Gorishniy et al., 2021; 2022) (and many others), regularizations (Jeffares et al., 2023), pretraining (Bahri et al., 2021) and other methods (Hollmann et al., 2023). In this work, we focus specifically on architectures. In particular, the recent studies reveal that MLP-like backbones are still competitive (Kadra et al., 2021; Gorishniy et al., 2021; 2022), and that embeddings for continuous features (Gorishniy et al., 2022) significantly reduce the gap between tabular DL and GBDT. In this work, we show that a properly designed retrieval component can boost the performance of tabular DL even further.

**Retrieval-augmented models in general.** Usually, the retrieval-based models are designed as follows. For an input object, first, they retrieve relevant samples from available (training) data. Then, they process the input object together with the retrieved instances to produce the final prediction for the input object. One of the common motivations for designing retrieval-based schemes is the local learning paradigm (Bottou and Vapnik, 1992), and the simplest possible example of such a model is the $k$-nearest neighbors (kNN) algorithm (James et al., 2013). The promise of retrieval-based approaches was demonstrated across various domains, such as natural language processing (Lewis et al., 2020; Guu et al., 2020; Khandelwal et al., 2020; Izacard et al., 2022; Borgeaud et al., 2022), computer vision (Iscen et al., 2022; Long et al., 2022), CTR prediction (Qin et al., 2020; 2021; Du et al., 2022), and others. Additionally, retrieval-augmented models often have useful properties such as better interpretability (Wang and Sabuncu, 2023), robustness (Zhao and Cho, 2018) and others.

**Retrieval-augmented models for tabular data problems.** The classic example of non-deep retrieval-based tabular models are the neighbor-based and kernel methods (James et al., 2013; Nader et al., 2022). There are also deep retrieval-based models applicable to (or directly designed for) tabular data problems (Wilson et al., 2016; Kim et al., 2019; Ramsauer et al., 2021; Kossen et al., 2021;

Somepalli et al., 2021). Notably, some of them omit the retrieval step and use *all* training data points as the "retrieved" instances (Somepalli et al., 2021; Kossen et al., 2021; Schäfl et al., 2022). However, we show that the existing retrieval-based tabular DL models are only marginally better than simple parametric DL models, and that often comes with a cost of using heavy Transformer-like architectures. Compared to prior work, where several layers with *multiple multi-head vanilla attention modules* are often used (Ramsauer et al., 2021; Kossen et al., 2021; Somepalli et al., 2021), our model TabR implements its retrieval component with just *one single-head attention-like module, customized* in a way that makes it better suited for tabular data problems. As a result, TabR substantially outperforms the existing retrieval-based DL models while being significantly more efficient.

## 3 TABR

In this section, we design a new retrieval-augmented deep learning model for tabular data problems.

### 3.1 PRELIMINARIES

**Notation.** For a given supervised learning problem on tabular data, we denote the dataset as $\{(x_i, y_i)\}_{i=1}^n$ where $x_i \in \mathbb{X}$ represents the $i$-th object's features and $y_i \in \mathbb{Y}$ represents the $i$-th object's label. Depending on the context, the $i$ index can be omitted. We consider three types of tasks: binary classification $\mathbb{Y} = \{0, 1\}$, multiclass classification $\mathbb{Y} = \{1, ..., C\}$ and regression $\mathbb{Y} = \mathbb{R}$. For simplicity, in most places, we will assume that $x_i$ contains only numerical (continuous) features, and we will give additional comments on binary and categorical features when necessary. The dataset is split into three disjoint parts: $\overline{1, n} = I_{train} \cup I_{val} \cup I_{test}$, where the "train" part is used for training, the "validation" part is used for early stopping and hyperparameter tuning, and the "test" part is used for the final evaluation. An input object for which a given model makes a prediction is referred to as "input object" or "target object".

When the retrieval technique is used for a given target object, the retrieval is performed within the set of "context candidates" or simply "candidates": $I_{cand} \subseteq I_{train}$. The retrieved objects, in turn, are called "context objects" or simply "context". Optionally, the target object can be included in its own context. In this work, unless otherwise noted, we use the same set of candidates for all input objects and set $I_{cand} = I_{train}$ (which means retrieving from all training objects).

**Experiment setup.** We extensively describe our tuning and evaluation protocols in subsection D.6. The most important points are that, for any given algorithm, on each dataset, following Gorishniy et al. (2022), (1) we perform hyperparameter tuning and early stopping using the *validation* set; (2) for the best hyperparameters, in the main text, we report the metric on the *test* set averaged over 15 random seeds, and provide standard deviations in Appendix E; (3) when comparing any two algorithms, we take the standard deviations into account as described in subsection D.6; (4) to obtain ensembles of models of the same type, we split the 15 random seeds into three disjoint groups (i.e., into three ensembles) each consisting of five models, average predictions within each group, and report the average performance of the obtained three ensembles.

In this work, we mostly use the datasets from prior literature and provide their summary in Table 1 (sometimes, we refer to this set of datasets as "the default benchmark"). Additionally, in subsection 4.2, we use the recently introduced benchmark with middle-scale tasks ($\leq 50K$ objects) (Grinsztajn et al., 2022) where GBDT was reported to be superior to DL solutions.

Table 1: Dataset properties. "RMSE" denotes root-mean-square error, "Acc." denotes accuracy.

|  | CH | CA | HO | AD | DI | OT | HI | BL | WE | CO | MI |
|---|---|---|---|---|---|---|---|---|---|---|---|
| #objects | 10000 | 20640 | 22784 | 48842 | 53940 | 61878 | 98049 | 166821 | 397099 | 581012 | 1200192 |
| #num.features | 7 | 8 | 16 | 6 | 6 | 93 | 28 | 4 | 118 | 10 | 131 |
| #bin.features | 3 | 0 | 0 | 1 | 0 | 0 | 0 | 1 | 1 | 44 | 5 |
| #cat.features | 1 | 0 | 0 | 7 | 3 | 0 | 0 | 4 | 0 | 0 | 0 |
| metric | Acc. | RMSE | RMSE | Acc. | RMSE | Acc. | Acc. | RMSE | RMSE | Acc. | RMSE |
| #classes | 2 | – | – | 2 | – | 9 | 2 | – | – | 7 | – |
| majority class | 79% | – | – | 76% | – | 26% | 52% | – | – | 48% | – |

## 3.2 ARCHITECTURE

To build a retrieval-based tabular DL model, we choose an incremental approach, where we start from a simple retrieval-free architecture, and, step by step, add and improve a retrieval component.

Let's consider a generic feed-forward retrieval-free network $f(x) = P(E(x))$ informally partitioned into two parts: encoder $E : \mathbb{X} \to \mathbb{R}^d$ and predictor $P : \mathbb{R}^d \to \hat{\mathbb{Y}}$. To incrementally make it retrieval-based, we add retrieval module $R$ in a residual branch after $E$ as illustrated in Figure 2, where $\tilde{x} \in \mathbb{R}^d$ is the intermediate representation of the target object, $\{\tilde{x}_i\}_{i \in I_{cand}} \subset \mathbb{R}^d$ are the intermediate representations of the candidates and $\{y_i\}_{i \in I_{cand}} \subset \mathbb{Y}$ are the labels of the candidates.

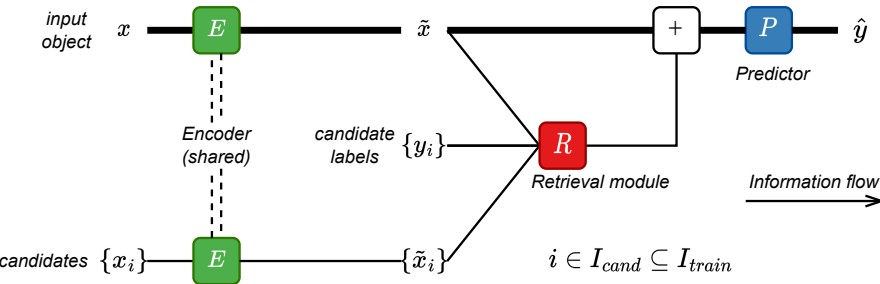

Figure 2: The generic retrieval-based architecture introduced in subsection 3.2 and used to build TabR. First, a target object and its candidates for retrieval are encoded with the same encoder $E$. Then, the retrieval module $R$ enriches the target object's representation by retrieving and processing relevant objects from the candidates. Finally, predictor $P$ makes a prediction. The bold path highlights the structure of the feed-forward retrieval-free model before the addition of the retrieval module $R$.

**Encoder and predictor**. The encoder $E$ and predictor $P$ modules (Figure 2) are not the focus of this work, so we keep them simple as illustrated in Figure 3.

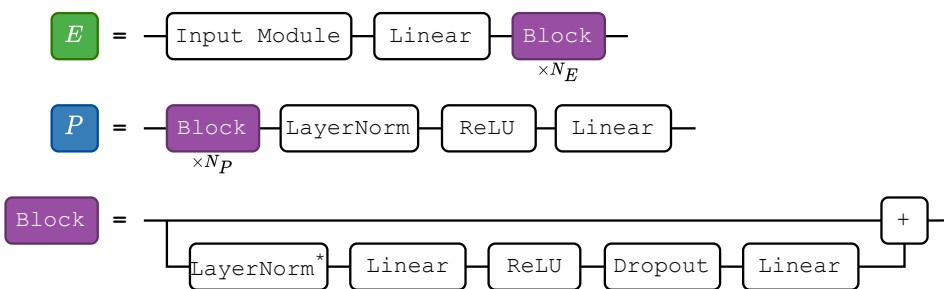

Figure 3: Encoder $E$ and predictor $P$ introduced in Figure 2. $N_E$ and $N_P$ denote the number of `Block` modules in $E$ and $P$, respectively. The `Input Module` encapsulates the input processing routines (feature normalization, one-hot encoding, etc.) and assembles a vector input for the subsequent linear layer. In particular, `Input Module` can contain embeddings for continuous features (Gorishniy et al., 2022). (* `LayerNorm` is omitted in the first `Block` of $E$.)

**Retrieval module**. We define the retrieval module $R$ in the spirit of $k$-nearest neighbors as illustrated in Figure 4. In the figure, the following formal details are omitted for clarity:

1. If the encoder $E$ contains at least one `Block` (i.e. $N_E > 0$), then, before being passed to $R$, $\tilde{x}$ and all $\tilde{x}_i$ are normalized with a shared layer normalization (Ba et al., 2016).
2. Optionally, the target object itself can be unconditionally (i.e. ignoring the top-$m$ operation) added as the $(m+1)$-th object to its set of context objects with the similarity score $\mathcal{S}(\tilde{x}, \tilde{x})$.
3. Dropout is applied to the weights produced by the softmax function.
4. We use $m = 96$ (ablated in subsection A.6) and, unless otherwise noted, $I_{cand} = I_{train}$.

Now, we iterate over possible designs of the similarity module $\mathcal{S}$ and the value module $\mathcal{V}$ (introduced in Figure 4). During this process, we do not use embeddings for numerical features (Gorishniy et al., 2022) in the `Input Module` of the encoder $E$ and set $N_E = 0$, $N_P = 1$ (see Figure 3).

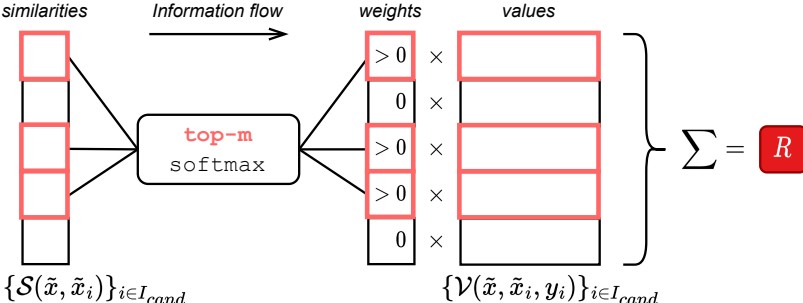

Figure 4: Simplified illustration of the retrieval module $R$ introduced in Figure 2 (the omitted details are provided in the main text). For the target object's representation $\tilde{x}$, the module takes the $m$ nearest neighbors among the candidates $\{\tilde{x}_i\}$ according to the similarity module $S : (\mathbb{R}^d, \mathbb{R}^d) \to \mathbb{R}$ and aggregates their values produced by the value module $\mathcal{V} : (\mathbb{R}^d, \mathbb{R}^d, \mathbb{Y}) \to \mathbb{R}^d$.

**Step-0. The vanilla-attention-like baseline**. The self-attention operation (Vaswani et al., 2017) was often used in prior work to model the interaction between a target object and candidate/context objects (Somepalli et al., 2021; Kossen et al., 2021; Schäfl et al., 2022). Then, instantiating retrieval module $R$ as the vanilla self-attention (modulo the top-$m$ operation) is a reasonable baseline:

$$\mathcal{S}(\tilde{x}, \tilde{x}_i) = W_Q(\tilde{x})^T W_K(\tilde{x}_i) \cdot d^{-1/2} \qquad \mathcal{V}(\tilde{x}, \tilde{x}_i, y_i) = W_V(\tilde{x}_i) \tag{1}$$

where $W_Q$, $W_K$, and $W_V$ are linear layers, and the target object is added as the $(m+1)$-th object to its own context (i.e., ignoring the top-$m$ operation). As reported in Table 2, the Step-0 configuration performs similarly to MLP, which means that using the vanilla self-attention is a suboptimal strategy.

**Step-1. Adding context labels**. A natural attempt to improve the Step-0 configuration is to utilize labels of the context objects, for example, by incorporating them into the value module as follows:

$$\mathcal{S}(\tilde{x}, \tilde{x}_i) = W_Q(\tilde{x})^T W_K(\tilde{x}_i) \cdot d^{-1/2} \qquad \mathcal{V}(\tilde{x}, \tilde{x}_i, y_i) = \underline{W_Y(y_i)} + W_V(\tilde{x}_i) \tag{2}$$

where the difference with Equation 1 is the underlined addition of $W_Y : \mathbb{Y} \to \mathbb{R}^d$, which is an embedding table for classification tasks and a linear layer for regression tasks. Table 2 shows no improvements from using labels, which is counter-intuitive. Perhaps, the similarity module $S$ taken from the vanilla attention does not allow benefiting from such a valuable signal as labels.

**Step-2. Improving the similarity module $S$**. Empirically, we observed that removing the notion of queries (i.e. removing $W_Q$) and using the $L_2$ distance instead of the dot product significantly improves performance on several datasets in Table 2 (subsection A.1 provides more discussion):

$$\mathcal{S}(\tilde{x}, \tilde{x}_i) = \underline{-\|W_K(\tilde{x}) - W_K(\tilde{x}_i)\|^2} \cdot d^{-1/2} \qquad \mathcal{V}(\tilde{x}, \tilde{x}_i, y_i) = W_Y(y_i) + W_V(\tilde{x}_i) \tag{3}$$

where the difference with Equation 2 is underlined. This change is a turning point in our story, which was overlooked in prior work. Crucially, in subsection A.3, we show that removing any of the three ingredients (context labels, key-only representation, $L_2$ distance) results in a performance drop back to the level of MLP. While the $L_2$ distance is unlikely to be the universally best choice (even within the tabular domain), it seems to be a reasonable default choice for tabular data problems.

**Step-3. Improving the value module $\mathcal{V}$**. After improving $S$ on Step-2, we turn to the value module $\mathcal{V}$. Inspired by DNNR (Nader et al., 2022) – the recently proposed generalization of the kNN algorithm, we make $\mathcal{V}$ more expressive by taking the target object's representation $\tilde{x}$ into account:

$$\mathcal{S}(\tilde{x}, \tilde{x}_i) = -\|W_K(\tilde{x}) - W_K(\tilde{x}_i)\|^2 \cdot d^{-1/2} \quad \mathcal{V}(\tilde{x}, \tilde{x}_i, y_i) = W_Y(y_i) + \underline{T(W_K(\tilde{x}) - W_K(\tilde{x}_i))}$$
$$T(\cdot) = \texttt{LinearWithoutBias(Dropout(ReLU(Linear(}\cdot\texttt{))))} \tag{4}$$

where the difference with Equation 3 is underlined. Table 2 shows that the new value module further improves the performance on several datasets. Intuitively, the term $W_Y(y_i)$ (the embedding of the context object's label) can be seen as the "raw" contribution of the $i$-th context object. The term $T(W_K(\tilde{x}) - W_K(\tilde{x}_i))$ can be seen as the "correction" term, where the module $T$ translates the differences in the key space into the differences in the label embedding space. We provide further analysis on the new value module in subsection A.2.

Table 2: The performance of the implementations of the retrieval module $R$, described in subsection 3.2. If a number is underlined, then it is better than the corresponding number from the previous step at least by the standard deviation. Noticeable improvements over MLP start at Step-2. Notation: $\downarrow$ corresponds to RMSE, $\uparrow$ corresponds to accuracy.

| | CH $\uparrow$ | CA $\downarrow$ | HO $\downarrow$ | AD $\uparrow$ | DI $\downarrow$ | OT $\uparrow$ | HI $\uparrow$ | BL $\downarrow$ | WE $\downarrow$ | CO $\uparrow$ |
|---|---|---|---|---|---|---|---|---|---|---|
| MLP | 0.854 | 0.499 | 3.112 | 0.853 | 0.140 | 0.816 | 0.719 | 0.697 | 1.905 | 0.963 |
| (Step-0) The vanilla attention baseline | 0.855 | $\underline{0.484}$ | 3.234 | $\underline{0.857}$ | 0.142 | 0.814 | 0.719 | 0.699 | 1.903 | 0.957 |
| (Step-1) + Context labels | 0.855 | 0.489 | 3.205 | 0.857 | 0.142 | 0.814 | 0.719 | 0.698 | 1.906 | $\underline{0.960}$ |
| (Step-2) + New similarity module $\mathcal{S}$ | $\underline{0.860}$ | $\underline{0.418}$ | $\underline{3.153}$ | 0.858 | $\underline{0.140}$ | 0.813 | 0.720 | $\underline{0.692}$ | $\underline{1.804}$ | $\underline{0.972}$ |
| (Step-3) + New value module $\mathcal{V}$ | 0.859 | $\underline{0.408}$ | 3.158 | $\underline{0.863}$ | $\underline{0.135}$ | 0.810 | 0.722 | 0.692 | 1.814 | $\underline{0.975}$ |
| (Step-4) + Technical tweaks = **TabR** | 0.860 | $\underline{0.403}$ | $\underline{3.067}$ | 0.865 | $\underline{0.133}$ | $\underline{0.818}$ | 0.722 | $\underline{0.690}$ | $\underline{1.747}$ | 0.973 |

**Step-4. TabR**. Finally, empirically, we observed that omitting the scaling term $d^{-1/2}$ in the similarity module and not including the target object to its own context leads to better results on average as reported in Table 2. Both aspects can be considered hyperparameters, and the above notes can be seen as our default recommendations. We call the obtained model "TabR" (Tab $\sim$ tabular, R $\sim$ retrieval). The formal complete description of how TabR implements the retrieval module $R$ is as follows:

$$k = W_K(\tilde{x}), \ k_i = W_K(\tilde{x}_i) \quad \mathcal{S}(\tilde{x}, \tilde{x}_i) = -\|k - k_i\|^2 \quad \mathcal{V}(\tilde{x}, \tilde{x}_i, y_i) = W_Y(y_i) + T(k - k_i) \quad (5)$$

where $W_K$ is a linear layer, $W_Y$ is an embedding table for classification tasks and a linear layer for regression tasks, (by default) a target object is not included in its own context, (by default) the similarity scores are not scaled, and $T(\cdot) = \texttt{LinearWithoutBias(Dropout(ReLU(Linear(\cdot))))}$.

**Limitations.** TabR has standard limitations of retrieval-augmented models, which we describe in Appendix B. We encourage practitioners to review the limitations before using TabR in practice.

## 4 EXPERIMENTS ON PUBLIC BENCHMARKS

In this section, we compare TabR (introduced in section 3) with existing retrieval-based solutions and state-of-the-art parametric models. In addition to the fully-fledged configuration of TabR (with all degrees of freedom available for $E$ and $P$ as described in Figure 3), we also use **TabR-S** ("S" stands for "simple") – a simple configuration, which does not use feature embeddings (Gorishniy et al., 2022), has a linear encoder ($N_E = 0$) and a one-block predictor ($N_P = 1$). We specify when TabR-S is used only in tables, figures, and captions but not in the text. For other details on TabR, including hyperparameter tuning, see subsection D.8.

### 4.1 EVALUATING RETRIEVAL-AUGMENTED DEEP LEARNING MODELS FOR TABULAR DATA

In this section, we compare TabR (section 3) and the existing retrieval-augmented solutions with fully parametric DL models (see Appendix D for implementation details for all algorithms). Table 3 indicates that TabR is the only retrieval-based model that provides a significant performance boost over MLP on many datasets. In particular, the full variation of TabR outperforms MLP-PLR (the modern parametric DL model with the highest average rank from Gorishniy et al. (2022)) on several datasets (CA, OT, BL, WE, CO), and performs on par with it on the rest except for the MI dataset. Regarding the prior retrieval-based solutions, we faced various technical limitations, such as incompatibility with classification problems and scaling issues (e.g., as we show in subsubsection A.4.1, it takes dramatically less time to train TabR than NPT (Kossen et al., 2021) – the closest retrieval-based competitor in Table 3). Notably, the retrieval component is not universally beneficial for all datasets.

The obtained results highlight the retrieval technique and embeddings for numerical features (Gorishniy et al., 2022) (used in MLP-PLR and TabR) as two powerful architectural elements that improve the optimization properties of tabular DL models. Interestingly, the two techniques are not fully orthogonal, but none of them can recover the full power of the other, and it depends on a given dataset whether one should prefer the retrieval, the embeddings, or a combination of both.

**The main takeaway.** TabR becomes a new strong deep learning solution for tabular data problems and demonstrates a good potential of the retrieval-based approach. TabR demonstrates strong average performance and achieves the new state-of-the-art on several datasets.

Table 3: Comparing TabR with existing retrieval-augmented tabular models and parametric DL models. The notation follows Table 2. The bold entries are the best-performing algorithms, which are defined with standard deviations taken into account as described in subsection D.6.

| | CH↑ | CA↓ | HO↓ | AD↑ | DI↓ | OT↑ | HI↑ | BL↓ | WE↓ | CO↑ | MI↓ | Avg. Rank |
|---|---|---|---|---|---|---|---|---|---|---|---|---|
| kNN | 0.837 | 0.588 | 3.744 | 0.834 | 0.256 | 0.774 | 0.665 | 0.712 | 2.296 | 0.927 | 0.764 | 6.0 ± 1.7 |
| DNNR (Nader et al., 2022) | – | 0.430 | 3.210 | – | 0.145 | – | – | 0.704 | 1.913 | – | 0.765 | 4.8 ± 1.9 |
| DKL (Wilson et al., 2016) | – | 0.521 | 3.423 | – | 0.147 | – | – | 0.699 | – | – | – | 6.2 ± 0.5 |
| ANP (Kim et al., 2019) | – | 0.472 | 3.162 | – | 0.140 | – | – | 0.705 | 1.902 | – | – | 4.6 ± 2.5 |
| SAINT (Somepalli et al., 2021) | **0.860** | 0.468 | 3.242 | 0.860 | 0.137 | 0.812 | 0.724 | 0.693 | 1.933 | 0.964 | 0.763 | 3.8 ± 1.5 |
| NPT (Kossen et al., 2021) | 0.858 | 0.474 | 3.175 | 0.853 | 0.138 | 0.815 | 0.721 | 0.692 | 1.947 | 0.966 | 0.753 | 3.6 ± 1.0 |
| MLP | 0.854 | 0.499 | 3.112 | 0.853 | 0.140 | 0.816 | 0.719 | 0.697 | 1.905 | 0.963 | 0.748 | 3.7 ± 1.3 |
| MLP-PLR | 0.860 | 0.476 | **3.056** | **0.870** | 0.134 | 0.819 | **0.729** | 0.687 | 1.860 | 0.970 | **0.744** | 2.0 ± 1.0 |
| TabR-S | 0.860 | **0.403** | **3.067** | 0.865 | **0.133** | 0.818 | 0.722 | 0.690 | 1.747 | 0.973 | 0.750 | 1.9 ± 0.7 |
| TabR | **0.862** | **0.400** | 3.105 | **0.870** | **0.133** | 0.825 | 0.729 | 0.676 | 1.690 | 0.976 | 0.750 | 1.3 ± 0.6 |

## 4.2 COMPARING TABR WITH GRADIENT-BOOSTED DECISION TREES

In this section, we compare TabR with models based on gradient-boosted decision trees (GBDT): XGBoost (Chen and Guestrin, 2016), LightGBM (Ke et al., 2017) and CatBoost (Prokhorenkova et al., 2018). Specifically, we compare ensembles (e.g. an ensemble of TabRs vs. an ensemble of XGBoosts) for a fair comparison since gradient boosting is already an ensembling technique.

**The default benchmark**. Table 4 shows that, on the default benchmark, the tuned TabR provides noticeable improvements over tuned GBDT on several datasets (CH, CA, HO, HI, WE, CO), while being competitive on the rest, except for the MI dataset. The table also demonstrates that TabR has a competitive default configuration (defined in subsection D.8).

**The benchmark from** Grinsztajn et al. (2022). Now, we go further and use the recently proposed benchmark with small-to-middle-scale tasks Grinsztajn et al. (2022). Importantly, this benchmark was originally used to illustrate the superiority of GBDT over parametric DL models on datasets with $\leq 50K$ objects, which makes it an interesting challenge for TabR. We adjust the benchmark to our tuning and evaluation protocols (see subsection C.2 for details) and report the results in Table 5. While MLP-PLR (one of the best parametric DL models) indeed is slightly inferior to GBDT on this set of tasks, TabR makes a significant step forward and outperforms GBDT on average.

Additional analysis: in subsection A.5, we try augmenting XGBoost with a retrieval component; in subsection A.4, we compare training times and batch inference efficiency of TabR and GBDT models.

**The main takeaway**. After the comparison with GBDT, TabR confirms its status of a new strong solution for tabular data problems: it provides strong average performance and can provide a noticeable improvement over GBDT on some datasets.

Table 4: Comparing ensembles of TabR with ensembles of GBDT models. See subsection D.8 to learn how the "default" TabR-S was obtained. The notation follows Table 3.

| | CH↑ | CA↓ | HO↓ | AD↑ | DI↓ | OT↑ | HI↑ | BL↓ | WE↓ | CO↑ | MI↓ | Avg. Rank |
|---|---|---|---|---|---|---|---|---|---|---|---|---|
| | | | | | Tuned hyperparameters | | | | | | | |
| XGBoost | 0.861 | 0.432 | 3.164 | **0.872** | 0.136 | **0.832** | 0.726 | 0.680 | 1.769 | 0.971 | 0.741 | 2.5 ± 0.9 |
| CatBoost | 0.859 | 0.426 | 3.106 | **0.872** | 0.133 | 0.827 | 0.727 | 0.681 | 1.773 | 0.969 | **0.741** | 2.5 ± 1.1 |
| LightGBM | 0.860 | 0.434 | 3.167 | **0.872** | 0.136 | **0.832** | 0.726 | 0.679 | 1.761 | 0.971 | 0.741 | 2.4 ± 0.9 |
| TabR | **0.865** | **0.391** | **3.025** | **0.872** | **0.131** | 0.831 | **0.733** | **0.674** | 1.661 | **0.977** | 0.748 | 1.3 ± 0.9 |
| | | | | | Default hyperparameters | | | | | | | |
| XGBoost | 0.856 | 0.471 | 3.368 | 0.871 | 0.143 | 0.817 | 0.716 | 0.683 | 1.920 | 0.966 | 0.750 | 3.4 ± 0.9 |
| CatBoost | 0.861 | 0.432 | 3.108 | **0.874** | 0.132 | 0.822 | **0.726** | 0.684 | 1.886 | 0.924 | **0.744** | 2.1 ± 0.8 |
| LightGBM | 0.856 | 0.449 | 3.222 | 0.869 | 0.137 | **0.826** | 0.720 | 0.681 | 1.817 | 0.899 | 0.744 | 2.5 ± 0.9 |
| TabR-S | **0.864** | **0.398** | **2.971** | 0.859 | **0.131** | 0.824 | 0.724 | 0.688 | **1.721** | **0.974** | 0.752 | 2.0 ± 1.3 |

Table 5: Comparing ensembles of DL models with ensembles of GBDT models on the benchmark from Grinsztajn et al. (2022) (e.g., an ensemble of MLPs vs ensemble of XGBoosts; note that in Figure 1, we compare *single* models, hence the different numbers). See subsection D.8 for the details on the "default" TabR-S. The default configuration of TabR-S is compared against the default configurations of GBDT models. The comparison is performed in a pairwise manner with standard deviations taken into account as described in subsection D.6.

| | vs. XGBoost Win / Tie / Loss | | | vs. CatBoost Win / Tie / Loss | | | vs. LightGBM Win / Tie / Loss | | |
|---|---|---|---|---|---|---|---|---|---|
| | *Tuned hyperparameters* | | | | | | | | |
| MLP | 6 | 11 | 26 | 6 | 8 | 29 | 5 | 11 | 27 |
| MLP-PLR | 12 | 17 | 14 | 10 | 11 | 22 | 14 | 15 | 14 |
| TabR-S | 21 | 13 | 9 | 17 | 11 | 15 | 21 | 15 | 7 |
| TabR | 26 | 14 | 3 | 23 | 13 | 7 | 26 | 14 | 3 |
| | *Default hyperparameters* | | | | | | | | |
| TabR-S | 28 | 10 | 5 | 17 | 16 | 10 | 25 | 9 | 9 |

## 5 ANALYSIS

### 5.1 FREEZING CONTEXTS FOR FASTER TRAINING OF TABR

In the vanilla formulation of TabR (section 3), for each training batch, the most up-to-date contexts are mined by encoding all the candidates and computing similarities with all of them, which can be prohibitively slow on large datasets. For example, it takes more than 18 hours to train a single TabR on the full "Weather prediction" dataset (Malinin et al., 2021) (3M+ objects; with the default hyperparameters from Table 4). However, as we show in Figure 5, for an average training object, its context (i.e. the top-$m$ candidates and the distribution over them according to the similarity module $\mathcal{S}$) gradually "stabilizes" during the course of training, which gives an opportunity for simple optimization. Namely, after a fixed number of epochs, we can perform "context freeze": i.e., compute the up-to-date contexts for all training (but not validation and test) objects for the one last time and then reuse these contexts for the rest of the training. Table 6 indicates that, on some datasets, this simple technique allows accelerating training of TabR without much loss in metrics, with more noticeable speedups on larger datasets. In particular, on the full "Weather prediction" dataset, we achieve nearly sevenfold speedup (from 18h9min to 3h15min) while maintaining competitive RMSE. See subsection D.2 for implementation details.

Table 6: The performance of TabR-S with the "context freeze" as described in subsection 5.1. TabR-S (CF-$N$) denotes TabR-S with the context freeze applied after $N$ epochs. In parentheses, we provide the fraction of time spent on training compared to the training without freezing (the last row).

| | CA ↓ | DI ↓ | HI ↑ | BL ↓ | WE ↓ | CO ↑ | WE (full) ↓ |
|---|---|---|---|---|---|---|---|
| TabR-S (CF-1) | 0.414 (0.72) | 0.137 (0.47) | **0.718** (0.80) | 0.692 (0.61) | 1.770 (0.57) | **0.973** (0.49) | 1.325 (0.13) |
| TabR-S (CF-4) | **0.409** (0.71) | 0.136 (0.51) | 0.717 (0.73) | **0.691** (0.62) | 1.763 (0.56) | **0.973** (0.59) | – |
| TabR-S | **0.406** (1.00) | **0.133** (1.00) | **0.719** (1.00) | **0.691** (1.00) | **1.755** (1.00) | **0.973** (1.00) | **1.315** (1.00) |

### 5.2 UPDATING TABR WITH NEW TRAINING DATA WITHOUT RETRAINING

Getting access to new unseen training data *after* training a machine learning model (e.g., after collecting yet another portion of daily logs of an application) is a common practical scenario. Technically, TabR allows utilizing the new data *without retraining* by adding the new data to the set of candidates for retrieval. We test this approach on the full "Weather prediction" dataset (Malinin et al., 2021) (3M+ objects). Figure 6 indicates that such "online updates" may be a viable solution for incorporating new data into an already trained TabR. Additionally, this approach can be used to scale TabR to large datasets by training the model on a subset of data and retrieving from the full data. Overall, we consider the conducted experiment as a preliminary exploration and leave a systematic study of continual updates for future work. See subsection D.3 for implementation details.

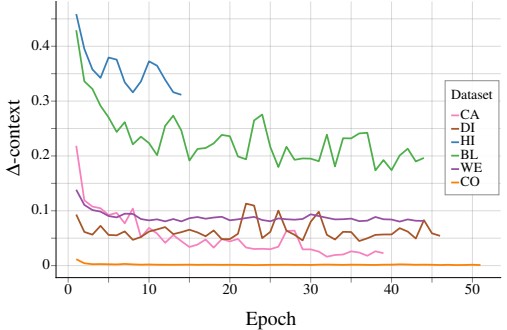

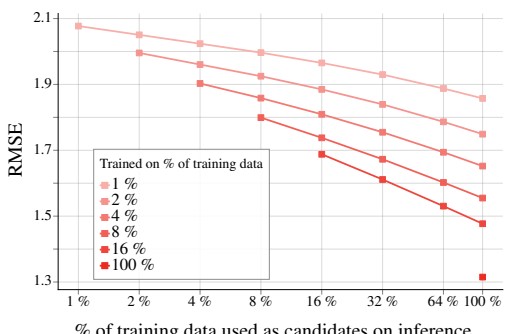

Figure 5: $\Delta$-context (explained below) averaged over training objects until the early stopping while training TabR-S. On a given epoch, for a given object, $\Delta$-context shows the portion of its context (the top-$m$ candidates and their weights) changed compared to the previous epoch (i.e., the lower the value, the smaller the change; see subsection D.2 for formal details). The plot shows that context updates become less intensive during the course of training, which motivates the optimization described in subsection 5.1.

Figure 6: Training TabR-S on various portions of the training data of the full "Weather prediction" dataset and gradually adding the remaining unseen training data to the set of candidates *without retraining* as described in subsection 5.2. For each curve, the leftmost point corresponds to not adding any new data to the set of candidates after the training, and the rightmost point corresponds to adding all unseen training data to the set of candidates.

## 5.3 FURTHER ANALYSIS

In the appendix, we provide a more insightful analysis. A non-exhaustive list of examples:

- in subsection A.1, we analyze the key-only $L_2$-based similarity module $\mathcal{S}$ introduced on Step-2 of subsection 3.2, which was a turning point in our story. We provide intuition behind this specific implementation of $\mathcal{S}$ and perform an in-depth comparison with the similarity module of the vanilla attention (the dot product between queries and keys).
- in subsection A.2, we analyze the value module $\mathcal{V}$ introduced on Step-3 of subsection 3.2. On regression problems, we confirm the correction semantics of the module $T$ from Equation 4.
- in subsubsection A.4.1, we compare training times of TabR with training times of all the baselines. We show that compared to prior retrieval-based tabular models, TabR makes a big step forward in terms of efficiency. While TabR is relatively slower than simple retrieval-free models, within the considered scope of dataset sizes, the absolute training times of TabR are affordable for most practical scenarios.
- in subsection A.8, we highlight additional technical properties of TabR.

## 6 CONCLUSION & FUTURE WORK

In this work, we have demonstrated that retrieval-based deep learning models have great potential in supervised machine learning problems on tabular data. Namely, we have designed TabR – a retrieval-augmented tabular DL architecture that provides strong average performance and achieves the new state-of-the-art on several datasets. Importantly, we have highlighted similarity and value modules as the important details of the attention mechanism which have a significant impact on the performance of attention-based retrieval components.

An important direction for future work is improving the efficiency of retrieval-augmented models to make them faster in general and in particular applicable to tens and hundreds of millions of data points. Also, in this paper, we focused more on the aspect of task performance, so some other properties of TabR remain underexplored. For example, the retrieval nature of TabR provides new opportunities for interpreting the model's predictions through the influence of context objects. Also, TabR may enable better support for continual learning (we scratched the surface of this direction in subsection 5.2). Regarding architecture details, possible directions are improving similarity and value modules, as well as performing multiple rounds of retrieval and interactions with the retrieved instances.

## REPRODUCIBILITY STATEMENT

To make the results and models reproducible and verifiable, **we provide our full codebase, all the results, and step-by-step usage instructions**: link. In particular, (1) *the results and hyperparameters reported the paper is just a summary of the results available at the provided URL* (with minor exceptions); (2) implementations of TabR and all the baselines (except for NPT) are available; (3) the hyperparameter tuning, training and evaluation pipelines are available; (4) the hyperparameters are available; (5) the used datasets and splits are available; (6) hyperparameter tuning and training times are available; (7) the used hardware is available; (8) *within a fixed environment (i.e. fixed hardware and software versions), most of the results are bitwise reproducible*.

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

SUPPLEMENTARY MATERIAL

# A  ADDITIONAL ANALYSIS

## A.1  SIMILARITY MODULE OF TABR

### A.1.1  MOTIVATION

Recall that in Step-2 of subsection 3.2, the change from the similarity module of the vanilla attention to the new key-only $L_2$-driven similarity module was a turning point in our story, where a retrieval-based model started showing noticeable improvements over MLP on several datasets. In fact, in addition to the empirical results (in Table 2 and subsection A.3), this specific similarity module has a reasonable intuitive motivation, which we now provide.

- **First**, aligning two (query and key) representations of target and candidate objects is an additional challenge for the optimization process, and there is no clear motivation for introducing this challenge in our case. And, as demonstrated in subsection A.3, avoiding this challenge is not just beneficial, but rather necessary.
- **Second**, during the design process in subsection 3.2, the similarity module $\mathcal{S}$ operates over *linear transformations* of the input (because, at that point, encoder $E$ is just a linear layer, since we fixed $N_E = 0$). Then, a reasonable similarity measure in the *original* feature space may remain reasonable in the *transformed* feature space. And, for tabular data, $L_2$ is usually a better similarity measure than the dot product in the *original* feature space. Note that the case of shallow/linear encoder is a specific, but very important case: since $E$ is applied to many candidates on each training step, $E$ is better to be lightweight to maintain adequate efficiency.

Combined, the above two points motivate removing query representations and switching to the $L_2$ distance, which leads to the similarity module introduced in Step-2 of subsection 3.2.

### A.1.2  ANALYZING ATTENTION PATTERNS OVER CANDIDATES

In this section, we analyze the similarity module $\mathcal{S}$ introduced in Step-2 of subsection 3.2, which greatly improved the performance on several datasets in Table 2.

Formally, for a given input object, the similarity module defines a distribution over candidates ("weights" in Figure 4) with exactly $m + 1$ non-zero entries ($m$ is the context size; $+1$ comes from adding the target object to its own context in Step-2). Intuitively, the less diverse such distributions are on average, the more frequently *different input objects* are augmented with *similar contexts*. In Table 7, we demonstrate that such distributions are more diverse on average with the new similarity module compared to the one from the vanilla attention. The implementation details are provided in subsection D.4.

Table 7: Entropy of the average distribution over candidates (the averaging is performed over individual distributions for test objects). The distributions are produced by the similarity module as explained in subsubsection A.1.2. The trained Step-1 and Step-2 models are taken directly from Table 2. The similarity module introduced at Step-2 of subsection 3.2 produces more diverse contexts.

|  | CH | CA | HO | AD | DI | OT | HI | WE |
|---|---|---|---|---|---|---|---|---|
| Step-1 | 6.6 | 6.1 | 7.0 | 7.1 | 5.8 | 5.3 | 8.5 | 8.9 |
| Step-2 | 8.4 | 9.0 | 9.3 | 9.7 | 10.3 | 10.1 | 10.5 | 9.5 |
| Uniform | 8.8 | 9.5 | 9.6 | 10.2 | 10.4 | 10.6 | 11.0 | 12.6 |

### A.1.3  CASE STUDIES

In this section, we consider three datasets where the transition to the key-only $L_2$ similarity module from the vanilla dot-product-between-queries-and-keys demonstrated the most impressive performance. Formally, this is the transition from "Step-1" to "Step-2" in Table 2. For each of the three datasets, first, we notice that, for a given input object, there is a domain-specific notion of "good neighbors", i.e., such neighbors that, from a human perspective, are very relevant to the input object

and provide strong hints for making a better prediction for the input object. Then, we show that the new similarity module allows finding and exploiting those natural hints.

**California housing (CA).** On this dataset, the transition from the "vanilla" dot-product-between-queries-and-keys similarity module to the key-only $L_2$ similarity module resulted in a substantial performance boost, as indicated by the difference between "Step-1" and "Step-2" in Table 2. On this dataset, the task is to estimate the prices of houses in California. Intuitively, for a given house from the test set, the prices of the training houses in the geographical neighborhood should be a strong hint for solving the task. Moreover, there are coordinates (longitude and latitude) among the features, which should simplify finding good neighbors. And the "Step-2" model successfully does that, which is not true for the "Step-1" model. Specifically, for an average test object, the "Step-2" model concentrates approximately 7% of the attention mass on the object itself (recall that "Step-2" includes the target object in the context objects) and approximately 77% on the context objects within the 10km radius. The corresponding numbers of the "Step-1" model are 0.07% and 1%.

**Weather prediction (WE).** Here, the story is seemingly similar to the one with the CA dataset analyzed in the previous paragraph, but in fact has a major difference. Again, here, for a given test data point, the dataset contains natural hints in the form of geographical neighbors from the training set which allow making a better weather forecast for a test query; and the "Step-2" model (Table 2) successfully exploits that, while the "Step-1" model cannot pay any meaningful attention to those hints. Specifically, for an average object, the "Step-2" model concentrates approximately 29% of the attention mass on the object itself (recall that "Step-2" includes the target object in the context objects) and approximately 25% on the context objects within the 200km radius. The corresponding numbers of the "Step-1" model are 0.25% and 0.5%. **However**, there is a crucial distinction from the CA case: in the version of the dataset WE that we used, *the features did not contain the coordinates*. In other words, to perform the analysis, *after* the training, we restored the original coordinates for each row from the original dataset and observed that the model *learned* the "correct" notion of "good neighbors" from other features.

**Facebook comments volume (FB).** In this paper, this is the first time when we mention this dataset, which was used in prior work (Gorishniy et al., 2022) and which we also used for some time in this project. Notably, on this dataset, TabR was demonstrating unthinkable improvements over competitors (including GBDT and the best-in-class parametric DL models). Then we noticed a strange pattern: often, for a given input, TabR concentrated an abnormally high percentage of its attention mass on just one context object (a different one for each input object). This is how we discovered that the dataset split that we inherited from Gorishniy et al. (2022) contained a "leak": roughly speaking, for many objects, it was possible to find their almost exact copies in the training set, and the task was dramatically simpler with this kind of hint. In practice, it was dramatically simpler for the TabR, but not for other models. Specifically, for an average object, the "Step-2" model concentrates approximately 20% of the attention mass on the object itself (recall that "Step-2" includes the target object in the context objects) and approximately 35% on its leaked almost-copies. The corresponding numbers of the "Step-1" model are 0.5% and 0.09%.

## A.2 Analyzing the value module of TabR

In this section, we analyze the value module $\mathcal{V}$ of TabR (see Equation 5).

### A.2.1 Motivation

Formally, we note that the output of the value module $\mathcal{V}$ of the vanilla attention (as defined in Equation 3, that is, *before* the Step-3 modification) does not depend on the target object representation $\tilde{x}$. This gives an opportunity for making $\mathcal{V}$ more expressive by taking $\tilde{x}$ into account. While there are numerous technical ways to use this opportunity, we decide to take inspiration from DNNR (Nader et al., 2022) – the recently proposed generalization of the kNN algorithm for regression problems.

*Conceptually*, while kNN captures only local label distributions, DNNR also captures local trends (formally, derivatives). *Technically*, contrary to kNN, in DNNR, a neighbor contributes to the prediction not only its label, but also an additional correction term that depends on the difference between the target object and the neighbor in the original feature space.

This is how we arrive at Equation 4 – the new value module $\mathcal{V}$. Similarly to DNNR, this module builds its output out of two terms:

1. The embedding of the context object's label.
2. The "correction" term, where the module $T$ translates the differences in the key space into the differences in the label embedding space.

### A.2.2 QUANTIFYING THE "CORRECTION" SEMANTICS

Recall the formal definition of the value module $\mathcal{V}$ of TabR (see Equation 5):

$$\mathcal{V}(\tilde{x}, \tilde{x}_i, y_i) = W_Y(y_i) + T(k - k_i) = W_Y(y_i) + T(\Delta k_i) \tag{6}$$

Intuitively, for a given context object, its label $y_i$ can be an important part of its contribution to the prediction. Let's consider regression problems, where, in Equation 6, $y_i \in \mathbb{R}$ is embedded by $W_Y$ to $\tilde{\mathbb{Y}} \subset \mathbb{R}^d$. Since $W_Y$ is a linear layer, $\tilde{\mathbb{Y}}$ is just a line, and each point on this line can be mapped back to the corresponding label from $\mathbb{R}$. Then, the projection of the correction term $T(\Delta k_i)$ on $\tilde{\mathbb{Y}}$ can be translated to the correction of the context label $y_i$:

$$\mathcal{V}(\tilde{x}, \tilde{x}_i, y_i) = W_Y(y_i) + \underline{\text{proj}_{\tilde{\mathbb{Y}}} T(\Delta k_i)} + \text{proj}_{\tilde{\mathbb{Y}}\perp} T(\Delta k_i) = W_Y(y_i + \underline{\Delta y_i}) + \text{proj}_{\tilde{\mathbb{Y}}\perp} T(\Delta k_i) \tag{7}$$

To check whether the underlined correction term $\text{proj}_{\tilde{\mathbb{Y}}} T(\Delta k_i)$ (or $\Delta y_i$) is important, we take a *trained* TabR, and reevaluate it *without retraining* while ignoring this projection (which is equivalent to setting $\Delta y_i = 0$). As a baseline, we also try ignoring the projection of $T(\Delta k_i)$ on a random one-dimensional subspace instead of $\tilde{\mathbb{Y}}$. Table 8 indicates that the correction along $\tilde{\mathbb{Y}}$ plays a vital role for the model. The implementation details are provided in subsection D.5.

Table 8: Evaluating RMSE of trained TabR-S while ignoring projections of $T(\Delta k_i)$ on different one-dimensional subspaces as described in subsection A.2. The first column shows the projection on which one-dimensional subspace is removed from $T(\Delta k_i)$. The first row corresponds to not removing any projections (i.e., the unmodified TabR-S). Ignoring the projection on $\tilde{\mathbb{Y}}$ (the label embedding space) breaks the model while ignoring a random projection does not have much effect.

| | CA $\downarrow$ | HO $\downarrow$ | DI $\downarrow$ | BL $\downarrow$ | WE $\downarrow$ |
|---|---|---|---|---|---|
| – | 0.403 | 3.067 | 0.133 | 0.690 | 1.747 |
| random | 0.403 | 3.071 | 0.133 | 0.690 | 1.754 |
| $\tilde{\mathbb{Y}}$ | 0.465 | 3.649 | 0.364 | 0.695 | 2.003 |

For classification problems, we tested similar hypotheses but did not obtain any interesting results. Perhaps, the value module $\mathcal{V}$ and specifically the $T$ module should be designed differently to better model the nature of classification problems.

### A.3 ABLATION STUDY

Recall that on Step-2 of subsection 3.2, we mentioned that it was crucial that *all* changes from Step-2 compared to Step-0 (using labels + not using queries + using the $L_2$ distance instead of the dot product) are important to provide noticeable improvements over MLP on several datasets. Note that not using queries is equivalent to sharing weights of $W_Q$ and $W_K$: $W_Q = W_K$. Table 9 contains the results of the corresponding experiment and indeed demonstrates that the Step-2 configuration cannot be trivially simplified without loss in metrics (see the CH, CA, BL, WE datasets).

Overall, we hypothesize that both things are important: how valuable the additional signal is (Step-1) and how well we measure the distance from the target object to the source of that valuable signal (Step-2).

### A.4 EFFICIENCY

### A.4.1 COMPARING TRAINING TIMES

While TabR demonstrates strong performance, these benefits do not come for free, since, as with all retrieval-augmented models, the retrieval component of TabR brings additional overhead. In this

Table 9: The ablation study as described in subsection A.3. $W_Q = W_K$ means using only keys and not using queries. Step-2 is the only variation providing noticeable improvements over MLP on the CH, CA, BL, WE datasets.

| | $L_2$, | $W_Q = W_K$, | $W_Y$ | CH | CA | HO | AD | DI | OT | HI | BL | WE | Avg. Rank |
|---|---|---|---|---|---|---|---|---|---|---|---|---|---|
| MLP | | | | 0.854 | 0.499 | 3.112 | 0.853 | 0.140 | 0.816 | 0.719 | 0.697 | 1.905 | $2.4 \pm 1.4$ |
| Step-0 | ✗ | ✗ | ✗ | 0.855 | 0.484 | 3.234 | 0.857 | 0.142 | 0.814 | 0.719 | 0.699 | 1.903 | $2.4 \pm 0.9$ |
| Step-1 | ✗ | ✗ | ✓ | 0.855 | 0.489 | 3.205 | 0.857 | 0.142 | 0.814 | 0.719 | 0.698 | 1.906 | $2.4 \pm 1.2$ |
| | ✗ | ✓ | ✗ | 0.853 | 0.495 | 3.178 | 0.857 | 0.143 | 0.808 | 0.719 | 0.698 | 1.903 | $2.9 \pm 0.8$ |
| | ✗ | ✓ | ✓ | 0.857 | 0.495 | 3.217 | 0.857 | 0.141 | 0.808 | 0.717 | 0.698 | 1.881 | $2.7 \pm 0.7$ |
| | ✓ | ✗ | ✗ | 0.855 | 0.488 | 3.170 | 0.857 | 0.143 | 0.813 | 0.719 | 0.698 | 1.901 | $2.3 \pm 1.0$ |
| | ✓ | ✗ | ✓ | 0.856 | 0.498 | 3.206 | 0.858 | 0.142 | 0.812 | 0.721 | 0.699 | 1.900 | $2.4 \pm 1.1$ |
| | ✓ | ✓ | ✗ | 0.856 | 0.442 | 3.154 | 0.856 | 0.141 | 0.811 | 0.722 | 0.698 | 1.896 | $2.0 \pm 0.7$ |
| Step-2 | ✓ | ✓ | ✓ | 0.860 | 0.418 | 3.153 | 0.858 | 0.140 | 0.813 | 0.720 | 0.692 | 1.804 | $1.2 \pm 0.4$ |

section, we aim to quantify this overhead by comparing the training times of TabR with those of all the baselines. Table 10 shows two important things:

- first, TabR is significantly more efficient (i.e. provides a significantly better trade-off between the downstream performance and training times) than prior retrieval-augmented tabular models. In particular, TabR is significantly (and, sometimes, dramatically) more efficient than NPT (Kossen et al., 2021) – the closest retrieval-based competitor according to Table 3.
- second, within the considered scope of dataset sizes, the absolute training times of TabR will be affordable in practice. Moreover, the reported execution times are achieved with our naive implementation which lacks even some of the basic optimizations.

To sum up, compared to prior work on retrieval-based tabular DL, TabR makes a big step forward in terms of efficiency. TabR is relatively slower than simple models (GBDT, parametric DL models), and improving its efficiency is an important research direction. However, given the room for technical optimizations and techniques similar to context freeze (subsection 5.1), the future of retrieval-based tabular DL looks positive.

Table 10: Training times of the tuned models (from Table 3, Table 4 and Table 6) averaged over the random seeds. The format is hh:mm:ss. TabR-S (CF-4) is TabR-S with the context freeze (subsection 5.1) applied after four epochs. Colors describe the following informal tiers:
■ <5 minutes    ■ <30 minutes    ■ <2 hours    ■ <10 hours    ■ >10 hours

| | CH | CA | HO | AD | DI | OT | HI | BL | WE | CO | MI |
|---|---|---|---|---|---|---|---|---|---|---|---|
| XGBoost | 0:00:01 | 0:00:20 | 0:00:05 | 0:00:05 | 0:00:02 | 0:00:35 | 0:00:15 | 0:00:08 | 0:02:02 | 0:01:55 | 0:03:43 |
| LightGBM | 0:00:00 | 0:00:04 | 0:00:01 | 0:00:01 | 0:00:03 | 0:00:34 | 0:00:10 | 0:00:07 | 0:06:40 | 0:06:22 | 0:06:45 |
| MLP | 0:00:02 | 0:00:18 | 0:00:09 | 0:00:17 | 0:00:15 | 0:00:31 | 0:00:24 | 0:01:38 | 0:00:29 | 0:04:01 | 0:02:09 |
| MLP-PLR | 0:00:03 | 0:00:43 | 0:00:14 | 0:00:24 | 0:00:25 | 0:02:09 | 0:00:17 | 0:00:52 | 0:20:01 | 0:03:32 | 0:30:30 |
| | | | | | *Retrieval-augmented models* | | | | | | |
| TabR-S (CF-4) | 0:00:08 | 0:00:25 | 0:00:30 | 0:00:34 | 0:00:43 | 0:00:57 | 0:01:02 | 0:03:08 | 0:09:08 | 0:23:13 | – |
| TabR-S | 0:00:20 | 0:01:20 | 0:01:23 | 0:03:04 | 0:01:44 | 0:01:17 | 0:02:09 | 0:11:22 | 0:12:11 | 0:49:59 | 0:55:04 |
| TabR | 0:00:16 | 0:00:40 | 0:00:55 | 0:01:30 | 0:01:24 | 0:01:47 | 0:06:22 | 0:04:14 | 1:03:18 | 0:37:03 | 1:46:07 |
| DKL | – | 0:06:15 | 0:03:55 | – | 0:21:59 | – | – | 1:04:10 | – | – | – |
| ANP | – | 0:37:40 | 0:42:16 | – | 2:14:38 | – | – | 1:32:27 | 6:00:11 | – | – |
| SAINT | 0:00:23 | 0:06:04 | 0:01:44 | 0:00:58 | 0:01:55 | 0:05:37 | 0:03:47 | 0:06:22 | 2:55:51 | 6:17:20 | 5:39:37 |
| NPT | 0:08:44 | 0:06:58 | 0:12:21 | 0:11:22 | 0:54:55 | 10:45:42 | 3:26:47 | 0:55:04 | 5:28:56 | 12:05:28 | 8:07:36 |

### A.4.2 COMPARING INFERENCE EFFICIENCY: TABR VS. XGBOOST

In this section, we compare the inference efficiency of TabR and XGBoost. **Importantly,** our current implementation of TabR is naive and lacks even basic optimizations.

Table 11 indicates that the inference speeds are mostly comparable. More nuanced observations are as follows:

- On "non-simple" tasks (CA, OT, WE, CO), TabR is faster (informally, "non-simple" means that XGBoost needs many trees AND/OR XGBoost needs high depth AND/OR a dataset has more features).
- On "simple" tasks, XGBoost is faster (informally, "simple" means that XGBoost is shallow AND/OR dataset has few features).

With the growth of training size (e.g. see the MI dataset in Table 11), TabR may become slower because of the retrieval, however, there is significant room for optimizations:

- Caching candidate key representations instead of recomputing them on each forward pass.
- Performing the search in float16 instead of float32.
- Using approximate search techniques instead of the current brute force.
- Using only a subset of the training data as candidates.
- etc.

Table 11: Inference throughput (batch size 4096) of tuned TabR-S and XGBoost from Table 4 on NVIDIA 2080 Ti. The last row reports the ratio between XGBoost's throughput and TabR-S's throughput.

|  | CH | CA | HO | AD | DI | OT | HI | BL | WE | CO | MI |
|---|---|---|---|---|---|---|---|---|---|---|---|
| #objects | 6400 | 13209 | 14581 | 26048 | 34521 | 39601 | 62751 | 106764 | 296554 | 371847 | 723412 |
| #features | 11 | 8 | 16 | 14 | 9 | 93 | 28 | 9 | 119 | 54 | 136 |
| XGBoost #trees | 121 | 3997 | 1328 | 988 | 802 | 524 | 1040 | 1751 | 3999 | 1258 | 3814 |
| XGBoost maximum tree depth | 5 | 9 | 7 | 10 | 13 | 13 | 11 | 8 | 13 | 12 | 12 |
| XGBoost throughput (obj./sec.) | 2197k | 33k | 179k | 131k | 417k | 19k | 72k | 84k | 15k | 10k | 14k |
| TabR-S throughput (obj./sec.) | 35k | 35k | 55k | 33k | 43k | 40k | 37k | 27k | 34k | 23k | 11k |
| Overhead | 62.3 | 0.9 | 3.3 | 3.9 | 9.6 | 0.5 | 1.9 | 3.1 | 0.5 | 0.4 | 1.2 |

## A.5 AUGMENTING XGBOOST WITH A RETRIEVAL COMPONENT

After the successful results of TabR reported in subsection 4.2, we tried augmenting XGBoost with a simple retrieval component to ensure that we do not miss this opportunity to improve the baselines. Namely, for a given input object, we find $m = 96$ (equal to the context size of TabR) nearest training objects in the original feature space, average their features and labels (the label as-is for regression problems, the one-hot encoding representations for classification problems), concatenate the target object's features with the "average neighbor's" features and label, and the obtained vector is used as the input for XGBoost. The results in Table 12 indicate that this strategy does not lead to any noticeable profit for XGBoost. We tried to vary the number of neighbors but did not achieve any significant improvements.

Table 12: Results for ensembles of tuned models. "XGBoost + retrieval" stands for XGBoost augmented with the "average neighbor's" features and label as described in subsection A.5.

|  | CH ↑ | CA ↓ | HO ↓ | AD ↑ | DI ↓ | OT ↑ | HI ↑ | BL ↓ | WE ↓ | CO ↑ | MI ↓ | Avg. Rank |
|---|---|---|---|---|---|---|---|---|---|---|---|---|
| XGBoost | 0.861 | 0.432 | 3.164 | **0.872** | 0.136 | **0.832** | 0.726 | 0.680 | 1.769 | 0.971 | **0.741** | $1.9 \pm 0.7$ |
| XGBoost + retrieval | 0.855 | 0.436 | 3.134 | 0.871 | 0.133 | 0.815 | 0.724 | 0.687 | 1.788 | 0.962 | 0.743 | $2.5 \pm 0.5$ |
| TabR | **0.865** | **0.391** | **3.025** | **0.872** | **0.131** | 0.831 | **0.733** | **0.674** | **1.661** | **0.977** | 0.748 | $1.2 \pm 0.6$ |

## A.6 HOW THE PERFORMANCE OF TABR DEPENDS ON THE CONTEXT SIZE $m$?

Recall that, throughout the paper, we used the fixed $m = 96$ as the context size (the number of neighbors) for TabR. We evaluate other values of $m$ in Table 13. Crucially, the choice of $m$ must be made based on the performance on *validation* sets (not on the *test* tests). The results indicate that $m = 96$ is a reasonable default value.

## A.7 ADDITIONAL RESULTS FOR THE "CONTEXT FREEZE" TECHNIQUE

We report the extended results for subsection 5.1 in Figure 7, Table 14 and Table 15. For the formal definition of the $\Delta$-context metric, see subsection D.2.

Table 13: The average ranks over datasets from Table 1 of default TabR-S with different values of $m$.

|  | m=1 | m=2 | m=4 | m=8 | m=16 | m=32 | m=64 | m=96 | m=128 | m=256 |
|---|---|---|---|---|---|---|---|---|---|---|
| Validation Set | | | | | | | | | | |
| Avg. Rank | 5 | 4.5 | 4 | 3.25 | 2.75 | 2.12 | 2.12 | 1.88 | 1.88 | 1.88 |
| Rank Std. | 2.45 | 1.87 | 1.73 | 1.71 | 1.56 | 1.45 | 1.45 | 0.93 | 1.05 | 1.36 |
| Test Set | | | | | | | | | | |
| Avg. Rank | 4.12 | 3.62 | 3.5 | 3.12 | 2.38 | 2.12 | 2 | 1.62 | 1.75 | 1.75 |
| Rank Std. | 2.2 | 1.8 | 1.5 | 1.05 | 1.41 | 1.27 | 1.41 | 0.99 | 0.97 | 1.09 |

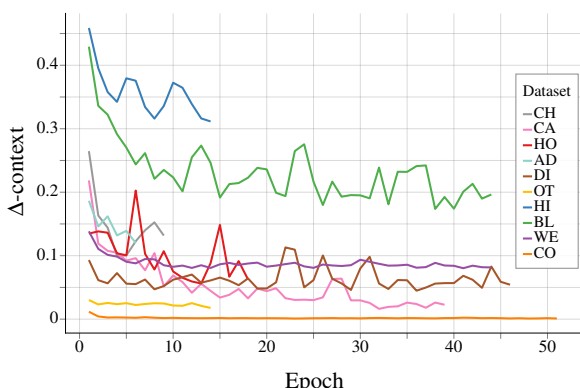

Figure 7: The extended version of Figure 5 with more datasets.

Table 14: The extended version of Table 6. Freezing after 0 epochs means freezing with a randomly initialized model. The speedups are provided in Table 15

|  | CH↑ | CA↓ | HO↓ | AD↑ | DI↓ | OT↑ | HI↑ | BL↓ | WE↓ | CO↑ | WE (full)↓ | Avg. Rank |
|---|---|---|---|---|---|---|---|---|---|---|---|---|
| MLP | 0.854 | 0.499 | 3.112 | 0.853 | 0.140 | **0.816** | **0.719** | 0.697 | 1.905 | 0.963 | – | 2.9 ± 1.5 |
| TabR-S (CF-0) | **0.857** | 0.424 | **3.075** | **0.857** | 0.137 | **0.816** | 0.718 | 0.700 | 1.787 | 0.969 | 1.387 | 2.3 ± 1.4 |
| TabR-S (CF-1) | **0.856** | 0.414 | **3.065** | 0.856 | 0.137 | **0.816** | 0.718 | 0.692 | 1.770 | **0.973** | 1.325 | 1.8 ± 1.0 |
| TabR-S (CF-2) | **0.856** | 0.411 | **3.074** | 0.856 | 0.137 | **0.816** | 0.718 | **0.691** | 1.767 | **0.973** | – | 1.7 ± 0.8 |
| TabR-S (CF-4) | **0.858** | 0.409 | 3.087 | **0.857** | 0.136 | **0.816** | 0.717 | **0.691** | 1.763 | **0.973** | – | 1.3 ± 0.5 |
| TabR-S (CF-8) | **0.858** | 0.407 | 3.118 | **0.857** | 0.135 | **0.817** | 0.719 | **0.691** | 1.761 | **0.973** | – | 1.3 ± 0.5 |
| TabR-S | **0.859** | **0.406** | 3.093 | **0.858** | 0.133 | **0.816** | 0.719 | **0.691** | **1.755** | **0.973** | **1.315** | 1.0 ± 0.0 |

Table 15: Fraction of time spent on training in Table 14, relative to the training time without the context freeze (the last row; the format is hours:minutes:seconds).

|  | CH | CA | HO | AD | DI | OT | HI | BL | WE | CO | WE (full) |
|---|---|---|---|---|---|---|---|---|---|---|---|
| TabR-S (CF-0) | 0.96 | 0.78 | 0.79 | 0.83 | 0.75 | 0.87 | 1.03 | 0.64 | 0.53 | 0.52 | 0.13 |
| TabR-S (CF-1) | 0.88 | 0.72 | 0.89 | 0.89 | 0.47 | 0.80 | 0.80 | 0.61 | 0.57 | 0.49 | 0.13 |
| TabR-S (CF-2) | 0.94 | 0.65 | 0.78 | 0.83 | 0.47 | 0.86 | 0.82 | 0.63 | 0.57 | 0.60 | – |
| TabR-S (CF-4) | 1.01 | 0.71 | 0.73 | 0.73 | 0.51 | 0.97 | 0.73 | 0.62 | 0.56 | 0.59 | – |
| TabR-S (CF-8) | 1.03 | 0.76 | 0.71 | 0.82 | 0.61 | 0.90 | 0.78 | 0.67 | 0.59 | 0.59 | – |
| TabR-S | 0:00:08 | 0:00:36 | 0:00:42 | 0:00:46 | 0:01:25 | 0:00:58 | 0:01:24 | 0:05:03 | 0:16:19 | 0:39:13 | 18:08:39 |

### A.8 Additional technical notes on TabR

We highlight the following technical aspects of TabR:

1. Because of the changes introduced in the Step-3 in subsection 3.2, the value representations $\mathcal{V}(\tilde{x}, \tilde{x}_i, y_i)$ of the candidates cannot be precomputed for a trained model, since they depend on the target object. This implies roughly twice less memory usage when deploying the model to production (since only the key representations and labels have to be deployed for training objects), but $\mathcal{V}(\tilde{x}, \tilde{x}_i, y_i)$ has to be computed in runtime.
2. Despite the attention-like nature of the retrieval module $R$, contrary to prior work, TabR does not suffer from the quadratic complexity w.r.t. the number of candidates, because it computes attention only for the target object, but not for the context objects.
3. In Equation 4, $T$ uses `LinearWithoutBias` in its definition. Strictly speaking, from the perspective of expressiveness, adding a bias (i.e. using `Linear`) would be redundant in the presence of $W_Y$ in Equation 4. And, just in case, we avoid this redundancy (we did not test using the simple `Linear` instead of `LinearWithoutBias`).

## B Limitations & Practical considerations

The following limitations and practical considerations are applicable to retrieval-augmented models in general. TabR itself does not add anything new to this list.

**First**, for a given application, one should carefully evaluate from various perspectives (business logic, legal considerations, ethical aspects, etc.) whether using real training objects for making predictions is reasonable.

**Second**, depending on an application, for a given target object, one may want to retrieve only from a subset of the available data, where the subset is dynamically formed for the target object based on application-specific filters. In terms of subsection 3.1, it means $I_{cand} = I_{cand}(x) \subset I_{train}$.

**Third**, ideally, retrieval during training should simulate retrieval during deployment, otherwise, a retrieval-based model can lead to (highly) suboptimal performance. Examples:

- For time series, during training, TabR must be allowed to retrieve only from the past. Moreover, perhaps, this "past" should also be limited to prevent the retrieval from too old data and too recent data. The decision should be made based on the domain expertise and business logic.
- Let's consider a task where, among all training objects, there are some "related objects". For example, when solving a ranking problem as a point-wise regression, such "related objects" can be obtained as query-document pairs corresponding to the same query, but different documents. In some cases, during training, for a given target object, retrieving from "related objects" can be unfair, because the same will not be possible in production for new objects that do not have "related objects" in the available data. Again, this design decision should be made based on the domain expertise and business logic.

**Lastly**, while TabR is significantly more efficient than prior retrieval-based tabular DL models, the retrieval module $R$ still causes overhead compared to purely parametric models, so TabR may not scale to truly large datasets as-is. We showcase a simple trick to scale TabR to larger datasets in subsection 5.1. We discuss the efficiency aspect in more detail in subsection A.4.

## C Benchmarks

### C.1 The default benchmark

In Table 16, we provide more information on the datasets from Table 1. The datasets include:

- Churn Modeling[1]
- California Housing (real estate data, (Kelley Pace and Barry, 1997))
- House 16H[2]

---

[1]https://www.kaggle.com/shrutimechlearn/churn-modelling
[2]https://www.openml.org/d/574

Table 16: Details on datasets from the main benchmark. "# Num", "# Bin", and "# Cat" denote the number of numerical, binary, and categorical features, respectively. The "Batch size" is the default batch size used to train DL-based models.

| Abbr | Name | # Train | # Validation | # Test | # Num | # Bin | # Cat | Task type | Batch size |
|------|------|---------|--------------|--------|-------|-------|-------|-----------|------------|
| CH | Churn Modelling | 6400 | 1600 | 2000 | 10 | 3 | 1 | Binclass | 128 |
| CA | California Housing | 13209 | 3303 | 4128 | 8 | 0 | 0 | Regression | 256 |
| HO | House 16H | 14581 | 3646 | 4557 | 16 | 0 | 0 | Regression | 256 |
| AD | Adult | 26048 | 6513 | 16281 | 6 | 1 | 8 | Binclass | 256 |
| DI | Diamond | 34521 | 8631 | 10788 | 6 | 0 | 3 | Regression | 512 |
| OT | Otto Group Products | 39601 | 9901 | 12376 | 93 | 0 | 0 | Multiclass | 512 |
| HI | Higgs Small | 62751 | 15688 | 19610 | 28 | 0 | 0 | Binclass | 512 |
| BL | Black Friday | 106764 | 26692 | 33365 | 4 | 1 | 4 | Regression | 512 |
| WE | Shifts Weather (subset) | 296554 | 47373 | 53172 | 118 | 1 | 0 | Regression | 1024 |
| CO | Covertype | 371847 | 92962 | 116203 | 54 | 44 | 0 | Multiclass | 1024 |
| WE (full) | Shifts Weather (full) | 2965542 | 47373 | 531720 | 118 | 1 | 0 | Regression | 1024 |

- Adult (income estimation, (Kohavi, 1996))
- Diamond[3]
- Otto Group Product Classification[4]
- Higgs (simulated physical particles, (Baldi et al., 2014); we use the version with 98K samples available in the OpenML repository (Vanschoren et al., 2014))
- Black Friday[5]
- Weather (temperature, (Malinin et al., 2021)). We take 10% of the dataset for our experiments due to its large size.
- Weather (full) (temperature, (Malinin et al., 2021)). Original splits from the paper.
- Covertype (forest characteristics, (Blackard and Dean., 2000))
- Microsoft (search queries, (Qin and Liu, 2013)). We follow the pointwise approach to learning to rank and treat this ranking problem as a regression problem.

## C.2 THE BENCHMARK FROM GRINSZTAJN ET AL. (2022)

In this section, we describe how exactly we used the benchmark proposed in Grinsztajn et al. (2022).

- We use the same train-val-test splits.
- When there are several splits for one dataset (i.e., when the n-fold-cross-validation was performed in Grinsztajn et al. (2022)), we first treat each of them as separate datasets while tuning and evaluating algorithms as described in Appendix D, but then, we average the metrics over the splits to obtain the final numbers for the dataset. For example, if there are five splits for a given dataset, then we tune and evaluate a given algorithm five times, each of the five tuned configurations is evaluated under 15 random seeds on the corresponding splits, and the reported metric value is the average over $5 * 15 = 75$ runs.
- When there are *multiple* versions of *one* dataset (e.g., the original regression task and the same dataset but converted to the binary classification task or the same dataset, but with the categorical features removed, etc.), we keep only one *original* dataset.
- We removed the "Eye movements" dataset because there is a leak in that dataset.
- We use the tuning and evaluation protocols as described in Appendix D, which was also used in prior works on tabular DL (Gorishniy et al., 2021; 2022). Crucially, we tune hyperparameters of the GBDT models more extensively than most (if not all) prior work in terms of both budget (20 warmup iterations of random sampling followed by 180 iterations of the tree-structured Parzen estimator algorithm) and hyperparameter spaces (see the corresponding sections in Appendix D).

---

[3]https://www.openml.org/d/42225
[4]https://www.kaggle.com/c/otto-group-product-classification-challenge/data
[5]https://www.openml.org/d/41540

# D IMPLEMENTATION DETAILS

## D.1 HARDWARE

We report the used hardware in the results published along with the source code. In a nutshell, the vast majority of experiments on GPU were performed on one NVidia A100 GPU, the remaining small part of GPU experiments was performed on one Nvidia 2080 Ti GPU, and there was also a small portion of runs performed on CPU (e.g. all the experiments on LightGBM).

## D.2 IMPLEMENTATION DETAILS OF SUBSECTION 5.1

In subsection 5.1, we used TabR-S with the default hyperparameters (see subsection D.8). To compute $\Delta$-context, we collect context distributions for training objects *between training epochs*. That is, after the $i$-th training epoch, we pause the training, collect the context distributions for all training objects, and then start the next $(i + 1)$-th training epoch.

$\Delta$-**context.** Intuitively, this heuristic metric describes in a single number how much, for a given input object, the context attention mass was updated compared to the *previous* epoch. Namely, it is a sum of two terms:

1. the `novel` attention mass, i.e. the attention mass coming from the context objects presented on the current epoch, but not presented on the previous epoch
2. the `increased` attention mass, i.e. we take the intersection of the current and the previous context objects and compute the increase of their total attention mass. We set it to 0.0 if actually decreased.

Now, we formally define this metric. For a given input object, let $a \in \mathbb{R}^{|I_{train}|}$ and $b \in \mathbb{R}^{|I_{train}|}$ denote the two distributions over the candidates from the previous and the current epochs, respectively. Let denote the sets of non-zero entries as $A = \{i : a_i > 0\}$ and $B = \{i : a_i > 0\}$. Note that $|A| = |B| = m = 96$. In other words, $A$ and $B$ are the contexts from the two epochs. Then:

$$\Delta\text{-context} = \texttt{novel} + \texttt{increased} \tag{8}$$

$$\texttt{novel} = \sum_{i \in B \setminus A} b_i \tag{9}$$

$$\texttt{increased} = \max \left( \sum_{i \in B \cap A} b_i - \sum_{i \in B \cap A} a_i, 0.0 \right) \tag{10}$$

## D.3 IMPLEMENTATION DETAILS OF SUBSECTION 5.2

In subsection 5.2, we used TabR-S with the default hyperparameters (see subsection D.8).

## D.4 IMPLEMENTATION DETAILS OF SUBSUBSECTION A.1.2

In subsubsection A.1.2, we performed the analysis over exactly the same model checkpoints that we used to assemble the rows "Step-1" and "Step-2" in Table 2.

To reiterate, this is how the entropy in Table 7 is computed:

1. First, we obtain individual distributions over candidates for all test objects. One such distribution contains exactly $(m + 1)$ non-zero entries.
2. Then, we average all individual distributions and obtain the average distribution.
3. Table 7 reports the entropy of the average distribution.

Note that, when obtaining the distribution over candidates, the top-$m$ operation is taken into account. Without that, if the distribution is always uniform regardless of the input object, then the average distribution will also be uniform and with the highest possible entropy, which would be misleading in the context of the story in subsubsection A.1.2.

Lastly, recall that in the Step-1 and Step-2 models, an input object is added to its own context. Then, the edge case when all input objects pay 100% attention only to themselves would lead to the highest

possible entropy, which would be misleading for the story in subsubsection A.1.2. In other words, for the story in subsubsection A.1.2, we should treat the "paying attention to self" behavior similarly for all objects. To achieve that, on the first step of the above recipe, we reassign the attention mass from "self" to a new virtual context object, which is *the same* for all input objects.

### D.5  IMPLEMENTATION DETAILS OF SUBSECTION A.2

To build Table 8, we used TabR-S with the default hyperparameters (see subsection D.8).

### D.6  EXPERIMENT SETUP

For the most part, we simply follow Gorishniy et al. (2022), but we provide all the details for completeness. Note that some of the prior work may differ from the common protocol that we describe below, but we provide the algorithm-specific implementation details further in this section.

**Data preprocessing.** For each dataset, for all DL-based solutions, the same preprocessing was used for fair comparison. For numerical features, by default, we used the quantile normalization from the Scikit-learn package (Pedregosa et al., 2011), with rare exceptions when it turned out to be detrimental (for such datasets, we used the standard normalization or no normalization). For categorical features, we used one-hot encoding. Binary features (i.e. the ones that take only two distinct values) are mapped to $\{0, 1\}$ without any further preprocessing.

**Training neural networks.** For DL-based algorithms, we minimize cross-entropy for classification problems and mean squared error for regression problems. We use the AdamW optimizer (Loshchilov and Hutter, 2019). We do not apply learning rate schedules. We do not use data augmentations. For each dataset, we used a predefined dataset-specific batch size. We continue training until there are `patience + 1` consecutive epochs without improvements on the validation set; we set `patience = 16` for the DL models.

**How we compare algorithms.** For a given dataset, first, we define the "preliminary best" algorithm as the algorithm with the best mean score. Then, we define a set of the best algorithms (i.e. their results are in bold in tables) as follows: a given algorithm is included in the best algorithms if its mean score differs from the mean score of the preliminary best algorithm by no more than the standard deviation of the preliminary best algorithm.

### D.7  EMBEDDINGS FOR NUMERICAL FEATURES

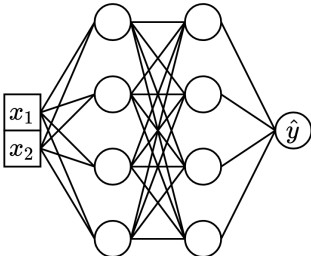
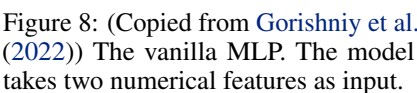
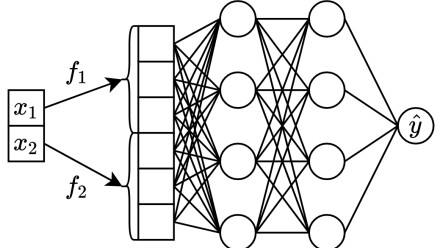

Figure 8: (Copied from Gorishniy et al. (2022)) The vanilla MLP. The model takes two numerical features as input.

Figure 9: (Copied from Gorishniy et al. (2022)) The same MLP as in Figure 8, but now with embeddings for numerical features.

In this work, we actively used embeddings for numerical features from (Gorishniy et al., 2022) (see Figure 8 and Figure 9), the technique which was reported to universally improve DL models. In a nutshell, for a given scalar numerical feature, an embedding module is a trainable module that maps this scalar feature to a vector. Then, the embeddings of all numerical features are concatenated into one flat vector which is passed to further layers. Following the original paper, when we use embeddings for numerical features, the same embedding architecture is used for all numerical features.

In this work, we used the LR (the combination of a linear layer and ReLU) and PLR (the combination of periodic embeddings, a linear layer, and ReLU) embeddings from the original paper. Also, we

introduce the `PLR(lite)` embedding, a simplified version of the `PLR` embedding where the linear layer is shared across all features. We observed it to be significantly more lightweight without critical performance loss.

**Hyperparameters tuning.** For the `LR` embeddings, we tune the embedding dimension in $\mathrm{Uniform}[16, 96]$. For the `PLR` and `PLR(lite)` embeddings, we tune the number of frequencies in $\mathrm{Uniform}[16, 96]$ (in $\mathrm{Uniform}[8, 96]$ for TabR on the datasets from Grinsztajn et al. (2022)), the frequency initialization scale in $\mathrm{LogUniform}[0.01, 100.0]$ and the embedding dimension in $\mathrm{Uniform}[16, 64]$ (in $\mathrm{Uniform}[4, 64]$ for TabR on the datasets from Grinsztajn et al. (2022)).

## D.8   TABR

The implementation, tuning hyperparameters, evaluation hyperparameters, metrics, execution times, hardware and other details are available in the source code. Here, we summarize some of the details for convenience.

**Embeddings for numerical features.** (see subsection D.7) For the non-simple configurations of TabR, on datasets CH, CA, HO, AD, DI, OT, HI, BL, and on all the datasets from Grinsztajn et al. (2022), we used the `PLR(lite)` embeddings as defined in subsection D.7. For other datasets, we used the `LR` embeddings.

**Other details.** We observed that initializing the $W_Y$ module properly may be important for good performance. Please, see the source code.

**Default TabR-S.** The default hyperparameters for TabR-S were obtained at some point in the project by literally averaging the tuned hyperparameters over multiple datasets. The specific set of datasets for averaging included all datasets from Table 16 plus two datasets that used to be a part of the default benchmark, but were excluded later. So, in total, 13 datasets contributed to the default hyperparameters.

Formally, this is not 100% fair to evaluate the obtained default TabR-S on the datasets which contributed to this default hyperparameters as in Table 4. However, we tested the fair leave-one-out approach as well (i.e. for a given dataset, averaging tuned hyperparameters over all datasets except for this one dataset) and did not observe any meaningful changes, so we decided to keep things simple and to have one common set of default hyperparameters for all datasets. Plus, the obtained default TabR-S demonstrates decent performance in Table 5 as well, which illustrates that the obtained default configuration is not strongly "overfitted" to the datasets from Table 16. The specific default hyperparameter values of TabR-S are as follows:

- $d = 265$
- Attention dropout rate $= 0.38920071545944357$
- Dropout rate in `FFN` $= 0.38852797479169876$
- Learning rate $= 0.0003121273641315169$
- Weight decay $= 0.00000122603520006404615$

**Hyperparameters.** The output size of the first linear layer of `FFN` and of $T$ is $2d$. We performed tuning using the tree-structured Parzen Estimator algorithm from the Akiba et al. (2019) library. The same protocol and hyperparameter spaces were used when tuning models in Table 2 and Table 9.

Table 17: The hyperparameter tuning space for TabR. Here (A) = {CH, CA, HO, AD, DI, OT, HI, BL}, (B) = {WE, CO, MI}. For the datasets from Grinsztajn et al. (2022), the tuning space is identical to (A) with the only difference that $d$ is tuned in UniformInt[16, 384].

| Parameter | (Datasets) Distribution | Comment |
|---|---|---|
| Width $d$ | (A,B) UniformInt[96, 384] | |
| Attention dropout rate | (A,B) Uniform[0.0, 0.6] | |
| Dropout rate in FFN | (A,B) Uniform[0.0, 0.6] | |
| Learning rate | (A,B) LogUniform[3e-5, 1e-3] | |
| Weight decay | (A,B) {0, LogUniform[1e-6, 1e-3]} | |
| $N_E$ | (A,B) UniformInt[0, 1] | Const[0] for TabR-S |
| $N_P$ | (A,B) UniformInt[1, 2] | Const[1] for TabR-S |
| # Tuning iterations | (A) 100 (B) 50 | |

## D.9 MLP

The implementation, tuning hyperparameters, evaluation hyperparameters, metrics, execution times, hardware and other details are available in the source code. Here, we summarize some of the details for convenience.

We used the implementation from Gorishniy et al. (2022).

**Hyperparameters.** We use the same hidden dimension throughout the whole network. We performed tuning using the tree-structured Parzen Estimator algorithm from the Akiba et al. (2019) library.

Table 18: The hyperparameter tuning space for MLP

| Parameter | Distribution |
|---|---|
| # layers | UniformInt[1, 6] |
| Width (hidden size) | UniformInt[64, 1024] |
| Dropout rate | {0.0, Uniform[0.0, 0.5]} |
| Learning rate | LogUniform[3e-5, 1e-3] |
| Weight decay | {0, LogUniform[1e-6, 1e-3]} |
| # Tuning iterations | 100 |

## D.10 FT-TRANSFORMER

The implementation, tuning hyperparameters, evaluation hyperparameters, metrics, execution times, hardware and other details are available in the source code. Here, we summarize some of the details for convenience.

We used the implementation from the "rtdl" Python package (version 0.0.13).

**Hyperparameters**. We use the `rtdl.FTTransformer.make_baseline` method to create FT-Transformer, so most of hyperparameters is inherited from this method's signature, and the rest is tuned as shown in the corresponding table.

Table 19: The hyperparameter tuning space for FT-Transformer

| Parameter | Distribution |
|---|---|
| # blocks | $\text{UniformInt}[1, 4]$ |
| $d_{token}$ | $\text{UniformInt}[16, 384]$ |
| Attention dropout rate | $\text{Uniform}[0.0, 0.5]$ |
| FFN hidden dimension expansion rate | $\text{Uniform}[2/3, 8/3]$ |
| FFN dropout rate | $\text{Uniform}[0.0, 0.5]$ |
| Residual dropout rate | $\{0.0, \text{Uniform}[0.0, 0.2]\}$ |
| Learning rate | $\text{LogUniform}[1e\text{-}5, 1e\text{-}3]$ |
| Weight decay | $\{0, \text{LogUniform}[1e\text{-}6, 1e\text{-}4]\}$ |
| # Tuning iterations | 100 |

## D.11 KNN

The implementation, tuning hyperparameters, evaluation hyperparameters, metrics, execution times, hardware and other details are available in the source code. Here, we summarize some of the details for convenience.

The features are preprocessed in the same way as for DL models. The only hyperparameter is the number of neighbors which we tune in $\text{UniformInt}[1, 128]$.

## D.12 DNNR

The implementation, tuning hyperparameters, evaluation hyperparameters, metrics, execution times, hardware and other details are available in the source code. Here, we summarize some of the details for convenience.

We've used the official implementation [6], but to evaluate DNNR on larger datasets with greater hyperparameters variability, we have rewritten parts of the source code to make it more efficient: enabling GPU usage, batched data processing, multiprocessing, where possible. Crucially, we leave the underlying method unchanged. We provide our efficiency-improved DNNR in the source code. There is no support for classification problems, so we evaluate DNNR only on regression problems.

**Hyperparameters.** We performed a grid-search over the main DNNR hyperparameters on all datasets, falling back to defaults (suggested by the authors) due to scaling issues on WE and MI.

Table 20: The hyperparameter grid used for DNNR. Here (A) = {CA, HO}; (B) = {DI, BL, WE, MI}. Notation: $N_f$ – number of features for the dataset.

| Parameter | (Datasets) Parameter grid | Comment |
|---|---|---|
| # neighbors $k$ | (A,B) $[1, 2, 3, \ldots, 128]$ | |
| Learned scaling | (A,B) [No scaling, Trained scaling] | |
| # neighbors used in scaling | (A,B) $[8 \cdot N_f, 2, 3, 4, 8, 16, 32, 64, 128]$ | $8 \cdot N_f$ on WE, MI |
| # epochs used in scaling | 10 | |
| Cat. feature encoding | [one-hot, leave-one-out] | |
| # neighbors for derivative $k'$ | (A) $\text{LinSpace}[2 \cdot N_f, 18 \cdot N_f, 20]$ | |
| | (B) $\text{LinSpace}[2 \cdot N_f, 12 \cdot N_f, 14]$ | |

---

[6] https://github.com/younader/dnnr

### D.13 DKL

The implementation, tuning hyperparameters, evaluation hyperparameters, metrics, execution times, hardware and other details are available in the source code. Here, we summarize some of the details for convenience.

We used DKL implementation from GPyTorch (Gardner et al., 2018). We do not evaluate DKL on WE and MI datasets due to scaling issues (tuning alone takes 1 day and 17 hours, compared to 3 hours for TabR on the medium DI dataset, for example). There is no support for classification problems, thus we evaluate DKL only on regression problems.

**Hyperparameters.** As with MLP we use the same hidden dimension throughout the whole network. And perform tuning using the tree-structured Parzen Estimator algorithm from the Akiba et al. [1] library.

Table 21: The hyperparameter tuning space for DKL

| Parameter | Distribution |
| --- | --- |
| Kernel | $\{\mathrm{rbf}, \mathrm{sm}\}$ |
| # layers | $\mathrm{UniformInt}[1, 4]$ |
| Width (hidden size) | $\mathrm{UniformInt}[64, 768]$ |
| Dropout rate | $\{0.0, \mathrm{Uniform}[0.0, 0.5]\}$ |
| Learning rate | $\mathrm{LogUniform}[1e\text{-}5, 1e\text{-}2]$ |
| Weight decay | $\{0, \mathrm{LogUniform}[1e\text{-}6, 1e\text{-}3]\}$ |
| # Tuning iterations | 100 |

### D.14 ANP

While the original paper introducing ANP did not focus on the tabular data, conceptually, it is very relevant to prior work on retrieval-based tabular DL, so we consider it as one of the baselines.

The implementation, tuning hyperparameters, evaluation hyperparameters, metrics, execution times, hardware and other details are available in the source code. Here, we summarize some of the details for convenience.

We used the Pytorch implementation from an unofficial repository[7] and modified it with respect to the official implementation from Kim et al. (2019). Specifically, we reimplemented `Decoder` class exactly as it was done in Kim et al. (2019) and changed a binary cross-entropy loss with a Gaussian negative log-likelihood loss in `LatentModel` class since it matches with the official implementation.

We do not evaluate ANP on the MI dataset due to scaling issues. Tuning alone on the smaller WE dataset took more than four days for 20(!) iterations (instead of 50-100 used for other algorithms). Also, there is no support for classification problems, thus we evaluate ANP only on regression problems.

We used 100 tuning iterations on CA and HO, 50 on DI, and 20 on BL and WE.

---

[7]https://github.com/soobinseo/Attentive-Neural-Process

Table 22: The hyperparameter tuning space for ANP

| Parameter | Distribution |
|---|---|
| # decoder layers | $\mathrm{UniformInt}[1, 3]$ |
| # cross-attention layers | $\mathrm{UniformInt}[1, 2]$ |
| # self-attention layers | $\mathrm{UniformInt}[1, 2]$ |
| Width (hidden size) | $\mathrm{UniformInt}[64, 384]$ |
| Learning rate | $\mathrm{LogUniform}[3e\text{-}5, 1e\text{-}3]$ |
| Weight decay | $\{0, \mathrm{LogUniform}[1e\text{-}6, 1e\text{-}4]\}$ |

### D.15    NPT

We use the official NPT (Kossen et al., 2021) implementation [8]. We leave the model and training code unchanged and only adjust the datasets and their preprocessing according to our protocols.

We evaluate the `NPT-Base` configuration of the model and follow both `NPT-Base` architecture and optimization hyperparameters. We train NPT for 2000 epochs on CH, CA, AD, HO, 10000 epochs on OT, WE, MI, 15000 epochs on DI, BL, HI and 30000 epochs on CO. For all datasets that don't fit into the A100 80GB GPU, we use batch size 4096 (as suggested in the NPT paper). We also decrease the hidden dim to 32 on WE and MI to avoid the OOM error.

Note that NPT is conceptually equivalent to other transformer-based non-parametric tabular DL solutions: (Somepalli et al., 2021; Schäfl et al., 2022). All three methods use dot-product-based self-attention modules alternating between self-attention between object features and self-attention between objects (for the whole training dataset or its random subset).

### D.16    SAINT

The implementation, tuning hyperparameters, evaluation hyperparameters, metrics, execution times, hardware and other details are available in the source code. Here, we summarize some of the details for convenience.

We use the official implementation of SAINT [9] **with one important fix**. Recall that, in SAINT, a target object interacts with its context objects with intersample attention. In the official implementation of SAINT, *context objects are taken from the same dataset part, as a target object*: for training objects, context objects are taken from the training set, for validation objects – from the validation set, for test objects – from the test set. This is different from the approach described in this paper, where *context objects are always taken from the training set*. Taking context objects from different dataset parts, as in the official implementation of SAINT, may be unwanted because of the following reasons:

1. model can have suboptimal validation and test performance because it is trained to operate when context objects are taken from the training set, but evaluated when context objects are taken from other dataset parts.
2. for a given validation/test object, *the prediction depends on other validation/test objects*. This is not in line with other retrieval-based models, which may result in inconsistent comparisons. Also, in many real-world scenarios, during deployment/test time, input objects should be processed independently, which is not the case for the official implementation of SAINT.

For the above reasons, we slightly modify SAINT such that each individual sample attends only to itself and to context samples from the training set, both during training and evaluation. See the source code for details.

On small datasets (CH, CA, HO, AD, DI, OT, HI, BL) we fix the number of attention heads at 8 and performed hyperparameter tuning using the tree-structured Parzen Estimator algorithm from the Akiba et al. (2019) library.

---

[8] https://github.com/OATML/non-parametric-transformers
[9] https://github.com/somepago/saint

Table 23: The hyperparameter tuning space for SAINT

| Parameter | Distribution |
|---|---|
| Depth | $\text{UniformInt}[1, 4]$ |
| Width | $\text{UniformInt}[4, 32, 4]$ |
| Feed forward multiplier | $\text{Uniform}[^2/_3, ^8/_3]$ |
| Attention dropout | $\text{Uniform}[0, 0.5]$ |
| Feed forward dropout | $\text{Uniform}[0, 0.5]$ |
| Learning rate | $\text{LogUniform}[3e\text{-}5, 1e\text{-}3]$ |
| Weight decay | $\{0, \text{LogUniform}[1e\text{-}6, 1e\text{-}4]\}$ |

On larger datasets (WE, CO, MI) we use slightly modified (for optimizing memory consumption) default configuration from the paper with following fixed hyperparameters:

- `depth` = 4
- `n_heads` = 8
- `dim` = 32
- `ffn_mult` = 4
- `attn_head_dim` = 48
- `attn_dropout` = 0.1
- `ff_dropout` = 0.8
- `learning_rate` = 0.0001
- `weight_decay` = 0.01

## D.17 XGBoost

The implementation, tuning hyperparameters, evaluation hyperparameters, metrics, execution times, hardware and other details are available in the source code. Here, we summarize some of the details for convenience.

In this work, we made our best to tune GBDT models as good as possible to make sure that the comparison is fair, and the conclusions are reliable. Compared to prior work (Gorishniy et al., 2021; 2022), where GBDT is already extensively tuned, we doubled the number of tuning iterations, doubled the number of trees, increased the maximum depth and increased the number of early stopping rounds by 4x.

The following hyperparameters are fixed and not tuned:

- `booster` = "gbtree"
- `n_estimators` = 4000
- `tree_method` = "gpu_hist"
- `early_stopping_rounds` = 200

We performed tuning using the tree-structured Parzen Estimator algorithm from the Akiba et al. (2019) library.

Table 24: The hyperparameter tuning space for XGBoost

| Parameter | Distribution |
|---|---|
| colsample_bytree | Uniform[0.5, 1.0] |
| gamma | {0.0, LogUniform[0.001, 100.0]} |
| lambda | {0.0, LogUniform[0.1, 10.0]} |
| learning_rate | LogUniform[0.001, 1.0] |
| max_depth | UniformInt[3, 14] |
| min_child_weight | LogUniform[0.0001, 100.0] |
| subsample | Uniform[0.5, 1.0 |
| # Tuning iterations | 200 |

## D.18 LIGHTGBM

The implementation, tuning hyperparameters, evaluation hyperparameters, metrics, execution times, hardware and other details are available in the source code. Here, we summarize some of the details for convenience.

In this work, we made our best to tune GBDT models as good as possible to make sure that the comparison is fair, and the conclusions are reliable. Compared to prior work (Gorishniy et al., 2021; 2022), where GBDT is already extensively tuned, we doubled the number of tuning iterations, doubled the number of trees, increased the maximum depth and increased the number of early stopping rounds by 4x.

The following hyperparameters are fixed and not tuned:

- n_estimators = 4000
- early_stopping_rounds = 200

We performed tuning using the tree-structured Parzen Estimator algorithm from the Akiba et al. (2019) library.

Table 25: The hyperparameter tuning space for LightGBM

| Parameter | Distribution |
|---|---|
| feature_fraction | Uniform[0.5, 1.0] |
| lambda_l2 | {0.0, LogUniform[0.1, 10.0]} |
| learning_rate | LogUniform[0.001, 1.0] |
| num_leaves | UniformInt[4, 768] |
| min_sum_hessian_in_leaf | LogUniform[0.0001, 100.0] |
| bagging_fraction | Uniform[0.5, 1.0] |
| # Tuning iterations | 200 |

## D.19 CATBOOST

The implementation, tuning hyperparameters, evaluation hyperparameters, metrics, execution times, hardware and other details are available in the source code. Here, we summarize some of the details for convenience.

In this work, we made our best to tune GBDT models as good as possible to make sure that the comparison is fair, and the conclusions are reliable. Compared to prior work (Gorishniy et al., 2021; 2022), where GBDT is already extensively tuned, we doubled the number of tuning iterations, doubled

the number of trees, increased the maximum depth and increased the number of early stopping rounds by 4x.

The following hyperparameters are fixed and not tuned:

- `n_estimators` = 4000
- `early_stopping_rounds` = 200
- `od_pval` = 0.001

We performed tuning using the tree-structured Parzen Estimator algorithm from the Akiba et al. (2019) library.

Table 26: The hyperparameter tuning space for CatBoost

| Parameter | Distribution |
|---|---|
| `bagging_temperature` | $\text{Uniform}[0.0, 1.0]$ |
| `depth` | $\text{UniformInt}[3, 14]$ |
| `l2_leaf_reg` | $\text{Uniform}[0.1, 10.0]$ |
| `leaf_estimation_iterations` | $\text{Uniform}[1, 10]$ |
| `learning_rate` | $\text{LogUniform}[0.001, 1.0]$ |
| # Tuning iterations | 200 |

# E  EXTENDED RESULTS WITH STANDARD DEVIATIONS

In this section, we provide the extended results with standard deviations for the main results reported in the main text. The results for the default benchmark are in the Table 27. The results for the benchmark from Grinsztajn et al. (2022) are in the Table 28.

Table 27: Extended results for the default benchmark. Results are grouped by datasets and span multiple pages below. Notation: ↓ corresponds to RMSE, ↑ corresponds to accuracy.

### CH ↑

| Method | Single model | Ensemble |
|---|---|---|
| | Tuned Hyperparameters | |
| kNN | $0.837 \pm 0.000$ | – |
| DNNR | – | – |
| DKL | – | – |
| ANP | – | – |
| NPT | $0.858 \pm 0.003$ | – |
| SAINT | $0.860 \pm 0.003$ | – |
| MLP | $0.854 \pm 0.003$ | – |
| MLP-PLR | $0.860 \pm 0.002$ | $0.860 \pm 0.001$ |
| TabR-S | $0.860 \pm 0.002$ | $0.862 \pm 0.002$ |
| TabR | $0.862 \pm 0.002$ | $0.865 \pm 0.001$ |
| CatBoost | $0.858 \pm 0.002$ | $0.859 \pm 0.001$ |
| XGBoost | $0.861 \pm 0.002$ | $0.861 \pm 0.001$ |
| LightGBM | $0.860 \pm 0.001$ | $0.860 \pm 0.000$ |
| | Default hyperparameters | |
| CatBoost | $0.860 \pm 0.002$ | $0.861 \pm 0.001$ |
| XGBoost | $0.855 \pm 0.000$ | $0.856 \pm 0.000$ |
| LightGBM | $0.856 \pm 0.000$ | $0.856 \pm 0.000$ |
| TabR-S | $0.859 \pm 0.003$ | $0.864 \pm 0.001$ |
| | Tuned hyperparameters (Table 2) | |
| step-0 | $0.855 \pm 0.003$ | $0.857 \pm 0.002$ |
| step-1 | $0.855 \pm 0.003$ | $0.858 \pm 0.002$ |
| step-2 | $0.860 \pm 0.002$ | $0.862 \pm 0.003$ |
| step-3 | $0.859 \pm 0.002$ | $0.862 \pm 0.002$ |

### CA ↓

| Method | Single model | Ensemble |
|---|---|---|
| | Tuned Hyperparameters | |
| kNN | $0.588 \pm 0.000$ | – |
| DNNR | $0.430 \pm 0.000$ | – |
| DKL | $0.521 \pm 0.055$ | – |
| ANP | $0.472 \pm 0.007$ | – |
| NPT | $0.474 \pm 0.003$ | – |
| SAINT | $0.468 \pm 0.005$ | – |
| MLP | $0.499 \pm 0.004$ | – |
| MLP-PLR | $0.476 \pm 0.004$ | $0.470 \pm 0.001$ |
| TabR-S | $0.403 \pm 0.002$ | $0.396 \pm 0.001$ |
| TabR | $0.400 \pm 0.003$ | $0.391 \pm 0.002$ |
| CatBoost | $0.429 \pm 0.001$ | $0.426 \pm 0.000$ |
| XGBoost | $0.433 \pm 0.002$ | $0.432 \pm 0.001$ |
| LightGBM | $0.435 \pm 0.002$ | $0.434 \pm 0.001$ |
| | Default hyperparameters | |
| CatBoost | $0.433 \pm 0.001$ | $0.432 \pm 0.001$ |
| XGBoost | $0.471 \pm 0.000$ | $0.471 \pm 0.000$ |
| LightGBM | $0.449 \pm 0.000$ | $0.449 \pm 0.000$ |
| TabR-S | $0.406 \pm 0.003$ | $0.398 \pm 0.001$ |
| | Tuned hyperparameters (Table 2) | |
| step-0 | $0.484 \pm 0.006$ | $0.470 \pm 0.005$ |
| step-1 | $0.489 \pm 0.007$ | $0.474 \pm 0.005$ |
| step-2 | $0.418 \pm 0.002$ | $0.411 \pm 0.000$ |
| step-3 | $0.408 \pm 0.003$ | $0.399 \pm 0.002$ |

### HO ↓

| Method | Single model | Ensemble |
|---|---|---|
| | Tuned Hyperparameters | |
| kNN | $3.744 \pm 0.000$ | – |
| DNNR | $3.210 \pm 0.000$ | – |
| DKL | $3.423 \pm 0.393$ | – |
| ANP | $3.162 \pm 0.028$ | – |
| NPT | $3.175 \pm 0.032$ | – |
| SAINT | $3.242 \pm 0.059$ | – |
| MLP | $3.112 \pm 0.036$ | – |
| MLP-PLR | $3.056 \pm 0.021$ | $2.993 \pm 0.019$ |
| TabR-S | $3.067 \pm 0.040$ | $2.996 \pm 0.027$ |
| TabR | $3.105 \pm 0.041$ | $3.025 \pm 0.010$ |
| CatBoost | $3.117 \pm 0.013$ | $3.106 \pm 0.002$ |
| XGBoost | $3.177 \pm 0.010$ | $3.164 \pm 0.007$ |
| LightGBM | $3.177 \pm 0.009$ | $3.167 \pm 0.005$ |
| | Default hyperparameters | |
| CatBoost | $3.122 \pm 0.011$ | $3.108 \pm 0.002$ |
| XGBoost | $3.368 \pm 0.000$ | $3.368 \pm 0.000$ |
| LightGBM | $3.222 \pm 0.000$ | $3.222 \pm 0.000$ |
| TabR-S | $3.093 \pm 0.060$ | $2.971 \pm 0.017$ |
| | Tuned hyperparameters (Table 2) | |
| step-0 | $3.234 \pm 0.053$ | $3.144 \pm 0.034$ |
| step-1 | $3.205 \pm 0.056$ | $3.104 \pm 0.043$ |
| step-2 | $3.153 \pm 0.031$ | $3.117 \pm 0.012$ |
| step-3 | $3.158 \pm 0.017$ | $3.117 \pm 0.006$ |

### AD ↑

| Method | Single model | Ensemble |
|---|---|---|
| | Tuned Hyperparameters | |
| kNN | $0.834 \pm 0.000$ | – |
| DNNR | – | – |
| DKL | – | – |
| ANP | – | – |
| NPT | $0.853 \pm 0.010$ | – |
| SAINT | $0.860 \pm 0.002$ | – |
| MLP | $0.853 \pm 0.001$ | – |
| MLP-PLR | $0.870 \pm 0.002$ | $0.873 \pm 0.001$ |
| TabR-S | $0.865 \pm 0.002$ | $0.868 \pm 0.002$ |
| TabR | $0.870 \pm 0.001$ | $0.872 \pm 0.001$ |
| CatBoost | $0.871 \pm 0.001$ | $0.872 \pm 0.001$ |
| XGBoost | $0.872 \pm 0.001$ | $0.872 \pm 0.000$ |
| LightGBM | $0.871 \pm 0.001$ | $0.872 \pm 0.000$ |
| | Default hyperparameters | |
| CatBoost | $0.873 \pm 0.001$ | $0.874 \pm 0.001$ |
| XGBoost | $0.871 \pm 0.000$ | $0.871 \pm 0.000$ |
| LightGBM | $0.869 \pm 0.000$ | $0.869 \pm 0.000$ |
| TabR-S | $0.858 \pm 0.001$ | $0.859 \pm 0.000$ |
| | Tuned hyperparameters (Table 2) | |
| step-0 | $0.857 \pm 0.002$ | $0.858 \pm 0.000$ |
| step-1 | $0.857 \pm 0.002$ | $0.860 \pm 0.000$ |
| step-2 | $0.858 \pm 0.002$ | $0.862 \pm 0.001$ |
| step-3 | $0.863 \pm 0.002$ | $0.866 \pm 0.001$ |

**DI ↓**

| Method | Single model | Ensemble |
|---|---|---|
| | Tuned Hyperparameters | |
| kNN | $0.256 \pm 0.000$ | – |
| DNNR | $0.145 \pm 0.000$ | – |
| DKL | $0.147 \pm 0.005$ | – |
| ANP | $0.140 \pm 0.001$ | – |
| NPT | $0.138 \pm 0.001$ | – |
| SAINT | $0.137 \pm 0.002$ | – |
| MLP | $0.140 \pm 0.001$ | – |
| MLP-PLR | $0.134 \pm 0.001$ | $0.133 \pm 0.000$ |
| TabR-S | $0.133 \pm 0.001$ | $0.131 \pm 0.000$ |
| TabR | $0.133 \pm 0.001$ | $0.131 \pm 0.000$ |
| CatBoost | $0.134 \pm 0.001$ | $0.133 \pm 0.000$ |
| XGBoost | $0.137 \pm 0.000$ | $0.136 \pm 0.000$ |
| LightGBM | $0.136 \pm 0.000$ | $0.136 \pm 0.000$ |
| | Default hyperparameters | |
| CatBoost | $0.133 \pm 0.000$ | $0.132 \pm 0.000$ |
| XGBoost | $0.143 \pm 0.000$ | $0.143 \pm 0.000$ |
| LightGBM | $0.137 \pm 0.000$ | $0.137 \pm 0.000$ |
| TabR-S | $0.133 \pm 0.001$ | $0.131 \pm 0.000$ |
| | Tuned hyperparameters (Table 2) | |
| step-0 | $0.142 \pm 0.001$ | $0.139 \pm 0.001$ |
| step-1 | $0.142 \pm 0.002$ | $0.138 \pm 0.000$ |
| step-2 | $0.140 \pm 0.001$ | $0.139 \pm 0.001$ |
| step-3 | $0.135 \pm 0.001$ | $0.133 \pm 0.001$ |

**OT ↑**

| Method | Single model | Ensemble |
|---|---|---|
| | Tuned Hyperparameters | |
| kNN | $0.774 \pm 0.000$ | – |
| DNNR | – | – |
| DKL | – | – |
| ANP | – | – |
| NPT | $0.815 \pm 0.002$ | – |
| SAINT | $0.812 \pm 0.002$ | – |
| MLP | $0.816 \pm 0.003$ | – |
| MLP-PLR | $0.819 \pm 0.002$ | $0.822 \pm 0.002$ |
| TabR-S | $0.818 \pm 0.002$ | $0.824 \pm 0.001$ |
| TabR | $0.825 \pm 0.002$ | $0.831 \pm 0.001$ |
| CatBoost | $0.825 \pm 0.001$ | $0.827 \pm 0.000$ |
| XGBoost | $0.830 \pm 0.001$ | $0.832 \pm 0.001$ |
| LightGBM | $0.830 \pm 0.001$ | $0.832 \pm 0.001$ |
| | Default hyperparameters | |
| CatBoost | $0.820 \pm 0.001$ | $0.822 \pm 0.001$ |
| XGBoost | $0.817 \pm 0.000$ | $0.817 \pm 0.000$ |
| LightGBM | $0.826 \pm 0.000$ | $0.826 \pm 0.000$ |
| TabR-S | $0.816 \pm 0.002$ | $0.824 \pm 0.000$ |
| | Tuned hyperparameters (Table 2) | |
| step-0 | $0.814 \pm 0.002$ | $0.823 \pm 0.002$ |
| step-1 | $0.814 \pm 0.002$ | $0.824 \pm 0.001$ |
| step-2 | $0.813 \pm 0.002$ | $0.818 \pm 0.001$ |
| step-3 | $0.810 \pm 0.002$ | $0.814 \pm 0.001$ |

**HI ↑**

| Method | Single model | Ensemble |
|---|---|---|
| | Tuned Hyperparameters | |
| kNN | $0.665 \pm 0.000$ | – |
| DNNR | – | – |
| DKL | – | – |
| ANP | – | – |
| NPT | $0.721 \pm 0.003$ | – |
| SAINT | $0.724 \pm 0.002$ | – |
| MLP | $0.719 \pm 0.002$ | – |
| MLP-PLR | $0.729 \pm 0.002$ | $0.735 \pm 0.000$ |
| TabR-S | $0.722 \pm 0.001$ | $0.726 \pm 0.001$ |
| TabR | $0.729 \pm 0.001$ | $0.733 \pm 0.001$ |
| CatBoost | $0.726 \pm 0.001$ | $0.727 \pm 0.001$ |
| XGBoost | $0.725 \pm 0.002$ | $0.726 \pm 0.001$ |
| LightGBM | $0.726 \pm 0.001$ | $0.726 \pm 0.001$ |
| | Default hyperparameters | |
| CatBoost | $0.725 \pm 0.001$ | $0.726 \pm 0.001$ |
| XGBoost | $0.716 \pm 0.000$ | $0.716 \pm 0.000$ |
| LightGBM | $0.720 \pm 0.000$ | $0.720 \pm 0.000$ |
| TabR-S | $0.719 \pm 0.002$ | $0.724 \pm 0.000$ |
| | Tuned hyperparameters (Table 2) | |
| step-0 | $0.719 \pm 0.002$ | $0.727 \pm 0.000$ |
| step-1 | $0.719 \pm 0.002$ | $0.724 \pm 0.001$ |
| step-2 | $0.720 \pm 0.002$ | $0.723 \pm 0.001$ |
| step-3 | $0.722 \pm 0.002$ | $0.724 \pm 0.000$ |

**BL ↓**

| Method | Single model | Ensemble |
|---|---|---|
| | Tuned Hyperparameters | |
| kNN | $0.712 \pm 0.000$ | – |
| DNNR | $0.704 \pm 0.000$ | – |
| DKL | $0.699 \pm 0.001$ | – |
| ANP | $0.705 \pm 0.005$ | – |
| NPT | $0.692 \pm 0.001$ | – |
| SAINT | $0.693 \pm 0.001$ | – |
| MLP | $0.697 \pm 0.001$ | – |
| MLP-PLR | $0.687 \pm 0.000$ | $0.684 \pm 0.000$ |
| TabR-S | $0.690 \pm 0.000$ | $0.688 \pm 0.000$ |
| TabR | $0.676 \pm 0.001$ | $0.674 \pm 0.001$ |
| CatBoost | $0.682 \pm 0.000$ | $0.681 \pm 0.000$ |
| XGBoost | $0.681 \pm 0.000$ | $0.680 \pm 0.000$ |
| LightGBM | $0.680 \pm 0.000$ | $0.679 \pm 0.000$ |
| | Default hyperparameters | |
| CatBoost | $0.685 \pm 0.000$ | $0.684 \pm 0.000$ |
| XGBoost | $0.683 \pm 0.000$ | $0.683 \pm 0.000$ |
| LightGBM | $0.681 \pm 0.000$ | $0.681 \pm 0.000$ |
| TabR-S | $0.691 \pm 0.000$ | $0.688 \pm 0.000$ |
| | Tuned hyperparameters (Table 2) | |
| step-0 | $0.699 \pm 0.001$ | $0.694 \pm 0.001$ |
| step-1 | $0.698 \pm 0.001$ | $0.693 \pm 0.001$ |
| step-2 | $0.692 \pm 0.001$ | $0.690 \pm 0.000$ |
| step-3 | $0.692 \pm 0.001$ | $0.688 \pm 0.000$ |

| WE ↓ | | |
|---|---|---|
| Method | Single model | Ensemble |
| *Tuned Hyperparameters* | | |
| kNN | $2.296 \pm 0.000$ | – |
| DNNR | $1.913 \pm 0.000$ | – |
| DKL | – | – |
| ANP | $1.902 \pm 0.009$ | – |
| NPT | $1.947 \pm 0.006$ | – |
| SAINT | $1.933 \pm 0.028$ | – |
| MLP | $1.905 \pm 0.005$ | – |
| MLP-PLR | $1.860 \pm 0.002$ | $1.833 \pm 0.002$ |
| TabR-S | $1.747 \pm 0.002$ | $1.718 \pm 0.001$ |
| TabR | $1.690 \pm 0.003$ | $1.661 \pm 0.002$ |
| CatBoost | $1.807 \pm 0.002$ | $1.773 \pm 0.001$ |
| XGBoost | $1.784 \pm 0.001$ | $1.769 \pm 0.001$ |
| LightGBM | $1.771 \pm 0.001$ | $1.761 \pm 0.001$ |
| *Default hyperparameters* | | |
| CatBoost | $1.895 \pm 0.001$ | $1.886 \pm 0.000$ |
| XGBoost | $1.920 \pm 0.000$ | $1.920 \pm 0.000$ |
| LightGBM | $1.845 \pm 0.003$ | $1.817 \pm 0.001$ |
| TabR-S | $1.755 \pm 0.002$ | $1.721 \pm 0.002$ |
| *Tuned hyperparameters (Table 2)* | | |
| step-0 | $1.903 \pm 0.004$ | $1.835 \pm 0.004$ |
| step-1 | $1.906 \pm 0.003$ | $1.845 \pm 0.001$ |
| step-2 | $1.804 \pm 0.003$ | $1.754 \pm 0.001$ |
| step-3 | $1.814 \pm 0.003$ | $1.765 \pm 0.001$ |

| CO ↑ | | |
|---|---|---|
| Method | Single model | Ensemble |
| *Tuned Hyperparameters* | | |
| kNN | $0.927 \pm 0.000$ | – |
| DNNR | – | – |
| DKL | – | – |
| ANP | – | – |
| NPT | $0.966 \pm 0.001$ | – |
| SAINT | $0.964 \pm 0.010$ | – |
| MLP | $0.963 \pm 0.001$ | – |
| MLP-PLR | $0.970 \pm 0.001$ | $0.974 \pm 0.000$ |
| TabR-S | $0.973 \pm 0.000$ | $0.974 \pm 0.000$ |
| TabR | $0.976 \pm 0.000$ | $0.977 \pm 0.000$ |
| CatBoost | $0.968 \pm 0.000$ | $0.969 \pm 0.000$ |
| XGBoost | $0.971 \pm 0.000$ | $0.971 \pm 0.000$ |
| LightGBM | $0.971 \pm 0.000$ | $0.971 \pm 0.000$ |
| *Default hyperparameters* | | |
| CatBoost | $0.923 \pm 0.000$ | $0.924 \pm 0.000$ |
| XGBoost | $0.966 \pm 0.000$ | $0.966 \pm 0.000$ |
| LightGBM | $0.884 \pm 0.016$ | $0.899 \pm 0.005$ |
| TabR-S | $0.973 \pm 0.001$ | $0.974 \pm 0.000$ |
| *Tuned hyperparameters (Table 2)* | | |
| step-0 | $0.957 \pm 0.002$ | $0.965 \pm 0.001$ |
| step-1 | $0.960 \pm 0.002$ | $0.967 \pm 0.001$ |
| step-2 | $0.972 \pm 0.000$ | $0.973 \pm 0.000$ |
| step-3 | $0.975 \pm 0.001$ | $0.976 \pm 0.000$ |

| MI ↓ | | |
|---|---|---|
| Method | Single model | Ensemble |
| *Tuned Hyperparameters* | | |
| kNN | $0.764 \pm 0.000$ | – |
| DNNR | $0.765 \pm 0.000$ | – |
| DKL | – | – |
| ANP | – | – |
| NPT | $0.753 \pm 0.001$ | – |
| SAINT | $0.763 \pm 0.007$ | – |
| MLP | $0.748 \pm 0.000$ | – |
| MLP-PLR | $0.744 \pm 0.000$ | $0.743 \pm 0.000$ |
| TabR-S | $0.750 \pm 0.001$ | $0.749 \pm 0.000$ |
| TabR | $0.750 \pm 0.001$ | $0.748 \pm 0.000$ |
| CatBoost | $0.741 \pm 0.000$ | $0.741 \pm 0.000$ |
| XGBoost | $0.741 \pm 0.000$ | $0.741 \pm 0.000$ |
| LightGBM | $0.742 \pm 0.000$ | $0.741 \pm 0.000$ |
| *Default hyperparameters* | | |
| CatBoost | $0.745 \pm 0.000$ | $0.744 \pm 0.000$ |
| XGBoost | $0.750 \pm 0.000$ | $0.750 \pm 0.000$ |
| LightGBM | $0.747 \pm 0.000$ | $0.744 \pm 0.000$ |
| TabR-S | $0.757 \pm 0.001$ | $0.752 \pm 0.001$ |

Table 28: Extended results for Grinsztajn et al. (2022) benchmark. Results are grouped by datasets and span multiple pages below. Notation: ↓ corresponds to RMSE, ↑ corresponds to accuracy.

### Ailerons ↓

| Method | Single model | Ensemble |
|---|---|---|
| *Tuned Hyperparameters* | | |
| MLP | $1.624 \pm 0.035$ | $1.620 \pm 0.037$ |
| MLP-PLR | $1.591 \pm 0.021$ | $1.582 \pm 0.019$ |
| TabR-S | $1.620 \pm 0.030$ | $1.595 \pm 0.022$ |
| TabR | $1.615 \pm 0.035$ | $1.585 \pm 0.042$ |
| CatBoost | $1.533 \pm 0.034$ | $1.527 \pm 0.037$ |
| XGBoost | $1.571 \pm 0.041$ | $1.565 \pm 0.040$ |
| LightGBM | $1.581 \pm 0.038$ | $1.577 \pm 0.040$ |
| *Default hyperparameters* | | |
| TabR-S | $1.615 \pm 0.029$ | $1.599 \pm 0.029$ |
| CatBoost | $1.542 \pm 0.041$ | $1.538 \pm 0.043$ |
| XGBoost | $1.644 \pm 0.046$ | $1.644 \pm 0.048$ |
| LightGBM | $1.594 \pm 0.051$ | $1.594 \pm 0.053$ |

### Bike Sharing Demand ↓

| Method | Single model | Ensemble |
|---|---|---|
| *Tuned Hyperparameters* | | |
| MLP | $45.702 \pm 0.756$ | $43.203 \pm 0.132$ |
| MLP-PLR | $42.615 \pm 0.415$ | $41.470 \pm 0.324$ |
| TabR-S | $43.637 \pm 0.681$ | $42.339 \pm 0.415$ |
| TabR | $42.649 \pm 0.939$ | $41.227 \pm 0.615$ |
| CatBoost | $40.927 \pm 0.232$ | $40.552 \pm 0.090$ |
| XGBoost | $42.766 \pm 0.126$ | $42.606 \pm 0.039$ |
| LightGBM | $42.503 \pm 0.190$ | $42.342 \pm 0.149$ |
| *Default hyperparameters* | | |
| TabR-S | $43.486 \pm 0.573$ | $42.369 \pm 0.354$ |
| CatBoost | $42.848 \pm 0.256$ | $42.626 \pm 0.243$ |
| XGBoost | $45.100 \pm 0.381$ | $45.100 \pm 0.410$ |
| LightGBM | $43.089 \pm 0.103$ | $43.089 \pm 0.111$ |

### Brazilian houses ↓

| Method | Single model | Ensemble |
|---|---|---|
| *Tuned Hyperparameters* | | |
| MLP | $0.049 \pm 0.018$ | $0.046 \pm 0.021$ |
| MLP-PLR | $0.043 \pm 0.019$ | $0.040 \pm 0.022$ |
| TabR-S | $0.049 \pm 0.015$ | $0.045 \pm 0.017$ |
| TabR | $0.045 \pm 0.016$ | $0.041 \pm 0.017$ |
| CatBoost | $0.047 \pm 0.031$ | $0.046 \pm 0.033$ |
| XGBoost | $0.054 \pm 0.027$ | $0.053 \pm 0.029$ |
| LightGBM | $0.060 \pm 0.025$ | $0.059 \pm 0.027$ |
| *Default hyperparameters* | | |
| TabR-S | $0.052 \pm 0.016$ | $0.048 \pm 0.018$ |
| CatBoost | $0.043 \pm 0.027$ | $0.042 \pm 0.029$ |
| XGBoost | $0.052 \pm 0.025$ | $0.052 \pm 0.027$ |
| LightGBM | $0.071 \pm 0.021$ | $0.071 \pm 0.022$ |

### Higgs ↑

| Method | Single model | Ensemble |
|---|---|---|
| *Tuned Hyperparameters* | | |
| MLP | $0.723 \pm 0.002$ | $0.725 \pm 0.001$ |
| MLP-PLR | $0.728 \pm 0.001$ | $0.730 \pm 0.001$ |
| TabR-S | $0.725 \pm 0.001$ | $0.728 \pm 0.000$ |
| TabR | $0.730 \pm 0.001$ | $0.733 \pm 0.000$ |
| CatBoost | $0.729 \pm 0.000$ | $0.730 \pm 0.000$ |
| XGBoost | $0.729 \pm 0.001$ | $0.730 \pm 0.000$ |
| LightGBM | $0.727 \pm 0.001$ | $0.728 \pm 0.000$ |
| *Default hyperparameters* | | |
| TabR-S | $0.722 \pm 0.001$ | $0.727 \pm 0.001$ |
| CatBoost | $0.727 \pm 0.001$ | $0.728 \pm 0.001$ |
| XGBoost | $0.718 \pm 0.000$ | $0.718 \pm 0.000$ |
| LightGBM | $0.721 \pm 0.000$ | $0.721 \pm 0.000$ |

### KDDCup09 upselling ↑

| Method | Single model | Ensemble |
|---|---|---|
| *Tuned Hyperparameters* | | |
| MLP | $0.776 \pm 0.011$ | $0.782 \pm 0.009$ |
| MLP-PLR | $0.797 \pm 0.009$ | $0.802 \pm 0.010$ |
| TabR-S | $0.784 \pm 0.014$ | $0.786 \pm 0.017$ |
| TabR | $0.791 \pm 0.012$ | $0.803 \pm 0.008$ |
| CatBoost | $0.799 \pm 0.012$ | $0.801 \pm 0.012$ |
| XGBoost | $0.793 \pm 0.011$ | $0.795 \pm 0.010$ |
| LightGBM | $0.793 \pm 0.012$ | $0.797 \pm 0.011$ |
| *Default hyperparameters* | | |
| TabR-S | $0.772 \pm 0.013$ | $0.781 \pm 0.013$ |
| CatBoost | $0.804 \pm 0.008$ | $0.804 \pm 0.006$ |
| XGBoost | $0.794 \pm 0.008$ | $0.794 \pm 0.009$ |
| LightGBM | $0.789 \pm 0.007$ | $0.789 \pm 0.007$ |

### MagicTelescope ↑

| Method | Single model | Ensemble |
|---|---|---|
| *Tuned Hyperparameters* | | |
| MLP | $0.853 \pm 0.006$ | $0.857 \pm 0.004$ |
| MLP-PLR | $0.860 \pm 0.007$ | $0.863 \pm 0.007$ |
| TabR-S | $0.868 \pm 0.006$ | $0.873 \pm 0.004$ |
| TabR | $0.864 \pm 0.005$ | $0.868 \pm 0.002$ |
| CatBoost | $0.859 \pm 0.007$ | $0.859 \pm 0.008$ |
| XGBoost | $0.855 \pm 0.009$ | $0.859 \pm 0.011$ |
| LightGBM | $0.855 \pm 0.008$ | $0.856 \pm 0.009$ |
| *Default hyperparameters* | | |
| TabR-S | $0.868 \pm 0.006$ | $0.871 \pm 0.005$ |
| CatBoost | $0.860 \pm 0.007$ | $0.860 \pm 0.008$ |
| XGBoost | $0.856 \pm 0.011$ | $0.856 \pm 0.012$ |
| LightGBM | $0.859 \pm 0.009$ | $0.859 \pm 0.010$ |

**Mercedes Benz Greener Manufacturing ↓**

| Method | Single model | Ensemble |
|---|---|---|
| | Tuned Hyperparameters | |
| MLP | $8.383 \pm 0.854$ | $8.336 \pm 0.888$ |
| MLP-PLR | $8.383 \pm 0.854$ | $8.336 \pm 0.888$ |
| TabR-S | $8.351 \pm 0.815$ | $8.269 \pm 0.840$ |
| TabR | $8.319 \pm 0.819$ | $8.244 \pm 0.844$ |
| CatBoost | $8.163 \pm 0.819$ | $8.155 \pm 0.844$ |
| XGBoost | $8.218 \pm 0.817$ | $8.209 \pm 0.846$ |
| LightGBM | $8.208 \pm 0.823$ | $8.162 \pm 0.857$ |
| | Default hyperparameters | |
| TabR-S | $8.290 \pm 0.838$ | $8.223 \pm 0.865$ |
| CatBoost | $8.167 \pm 0.825$ | $8.164 \pm 0.848$ |
| XGBoost | $8.371 \pm 0.787$ | $8.371 \pm 0.810$ |
| LightGBM | $8.280 \pm 0.845$ | $8.280 \pm 0.869$ |

**MiamiHousing2016 ↓**

| Method | Single model | Ensemble |
|---|---|---|
| | Tuned Hyperparameters | |
| MLP | $0.161 \pm 0.003$ | $0.157 \pm 0.003$ |
| MLP-PLR | $0.150 \pm 0.002$ | $0.147 \pm 0.002$ |
| TabR-S | $0.142 \pm 0.002$ | $0.139 \pm 0.002$ |
| TabR | $0.139 \pm 0.002$ | $0.136 \pm 0.002$ |
| CatBoost | $0.142 \pm 0.002$ | $0.141 \pm 0.003$ |
| XGBoost | $0.144 \pm 0.003$ | $0.143 \pm 0.003$ |
| LightGBM | $0.146 \pm 0.002$ | $0.145 \pm 0.003$ |
| | Default hyperparameters | |
| TabR-S | $0.141 \pm 0.002$ | $0.139 \pm 0.002$ |
| CatBoost | $0.142 \pm 0.003$ | $0.141 \pm 0.003$ |
| XGBoost | $0.160 \pm 0.003$ | $0.160 \pm 0.003$ |
| LightGBM | $0.152 \pm 0.004$ | $0.152 \pm 0.004$ |

**MiniBooNE ↑**

| Method | Single model | Ensemble |
|---|---|---|
| | Tuned Hyperparameters | |
| MLP | $0.947 \pm 0.001$ | $0.948 \pm 0.001$ |
| MLP-PLR | $0.947 \pm 0.001$ | $0.949 \pm 0.000$ |
| TabR-S | $0.949 \pm 0.001$ | $0.950 \pm 0.000$ |
| TabR | $0.948 \pm 0.001$ | $0.949 \pm 0.000$ |
| CatBoost | $0.945 \pm 0.001$ | $0.946 \pm 0.001$ |
| XGBoost | $0.944 \pm 0.001$ | $0.945 \pm 0.000$ |
| LightGBM | $0.942 \pm 0.001$ | $0.943 \pm 0.000$ |
| | Default hyperparameters | |
| TabR-S | $0.947 \pm 0.001$ | $0.950 \pm 0.001$ |
| CatBoost | $0.945 \pm 0.001$ | $0.945 \pm 0.000$ |
| XGBoost | $0.942 \pm 0.000$ | $0.942 \pm 0.000$ |
| LightGBM | $0.944 \pm 0.000$ | $0.944 \pm 0.000$ |

**OnlineNewsPopularity ↓**

| Method | Single model | Ensemble |
|---|---|---|
| | Tuned Hyperparameters | |
| MLP | $0.862 \pm 0.001$ | $0.860 \pm 0.000$ |
| MLP-PLR | $0.862 \pm 0.001$ | $0.860 \pm 0.000$ |
| TabR-S | $0.868 \pm 0.001$ | $0.863 \pm 0.001$ |
| TabR | $0.862 \pm 0.001$ | $0.859 \pm 0.000$ |
| CatBoost | $0.853 \pm 0.000$ | $0.853 \pm 0.000$ |
| XGBoost | $0.854 \pm 0.000$ | $0.854 \pm 0.000$ |
| LightGBM | $0.855 \pm 0.000$ | $0.854 \pm 0.000$ |
| | Default hyperparameters | |
| TabR-S | $0.870 \pm 0.001$ | $0.864 \pm 0.000$ |
| CatBoost | $0.855 \pm 0.000$ | $0.854 \pm 0.000$ |
| XGBoost | $0.874 \pm 0.000$ | $0.874 \pm 0.000$ |
| LightGBM | $0.862 \pm 0.000$ | $0.862 \pm 0.000$ |

**SGEMM GPU kernel performance ↓**

| Method | Single model | Ensemble |
|---|---|---|
| | Tuned Hyperparameters | |
| MLP | $0.016 \pm 0.000$ | $0.016 \pm 0.000$ |
| MLP-PLR | $0.015 \pm 0.000$ | $0.015 \pm 0.000$ |
| TabR-S | $0.017 \pm 0.001$ | $0.016 \pm 0.000$ |
| TabR | $0.015 \pm 0.000$ | $0.015 \pm 0.000$ |
| CatBoost | $0.017 \pm 0.000$ | $0.017 \pm 0.000$ |
| XGBoost | $0.017 \pm 0.000$ | $0.017 \pm 0.000$ |
| LightGBM | $0.017 \pm 0.000$ | $0.017 \pm 0.000$ |
| | Default hyperparameters | |
| TabR-S | $0.017 \pm 0.001$ | $0.016 \pm 0.000$ |
| CatBoost | $0.017 \pm 0.000$ | $0.017 \pm 0.000$ |
| XGBoost | $0.017 \pm 0.000$ | $0.017 \pm 0.000$ |
| LightGBM | $0.017 \pm 0.000$ | $0.017 \pm 0.000$ |

**analcatdata supreme ↓**

| Method | Single model | Ensemble |
|---|---|---|
| | Tuned Hyperparameters | |
| MLP | $0.078 \pm 0.009$ | $0.077 \pm 0.010$ |
| MLP-PLR | $0.079 \pm 0.008$ | $0.077 \pm 0.008$ |
| TabR-S | $0.080 \pm 0.007$ | $0.076 \pm 0.005$ |
| TabR | $0.081 \pm 0.009$ | $0.075 \pm 0.005$ |
| CatBoost | $0.078 \pm 0.007$ | $0.073 \pm 0.002$ |
| XGBoost | $0.080 \pm 0.013$ | $0.077 \pm 0.011$ |
| LightGBM | $0.078 \pm 0.012$ | $0.077 \pm 0.011$ |
| | Default hyperparameters | |
| TabR-S | $0.077 \pm 0.007$ | $0.074 \pm 0.007$ |
| CatBoost | $0.071 \pm 0.004$ | $0.071 \pm 0.004$ |
| XGBoost | $0.076 \pm 0.006$ | $0.076 \pm 0.006$ |
| LightGBM | $0.073 \pm 0.006$ | $0.073 \pm 0.006$ |

### bank-marketing ↑

| Method | Single model | Ensemble |
|---|---|---|
| *Tuned Hyperparameters* | | |
| MLP | $0.786 \pm 0.006$ | $0.790 \pm 0.004$ |
| MLP-PLR | $0.795 \pm 0.005$ | $0.798 \pm 0.004$ |
| TabR-S | $0.800 \pm 0.005$ | $0.802 \pm 0.004$ |
| TabR | $0.802 \pm 0.009$ | $0.804 \pm 0.010$ |
| CatBoost | $0.803 \pm 0.007$ | $0.806 \pm 0.008$ |
| XGBoost | $0.801 \pm 0.008$ | $0.803 \pm 0.008$ |
| LightGBM | $0.801 \pm 0.008$ | $0.801 \pm 0.007$ |
| *Default hyperparameters* | | |
| TabR-S | $0.800 \pm 0.006$ | $0.801 \pm 0.005$ |
| CatBoost | $0.803 \pm 0.009$ | $0.803 \pm 0.009$ |
| XGBoost | $0.800 \pm 0.009$ | $0.800 \pm 0.009$ |
| LightGBM | $0.803 \pm 0.004$ | $0.803 \pm 0.004$ |

### black friday ↓

| Method | Single model | Ensemble |
|---|---|---|
| *Tuned Hyperparameters* | | |
| MLP | $0.369 \pm 0.000$ | $0.367 \pm 0.000$ |
| MLP-PLR | $0.363 \pm 0.000$ | $0.363 \pm 0.000$ |
| TabR-S | $0.364 \pm 0.000$ | $0.363 \pm 0.000$ |
| TabR | $0.362 \pm 0.002$ | $0.359 \pm 0.001$ |
| CatBoost | $0.361 \pm 0.000$ | $0.360 \pm 0.000$ |
| XGBoost | $0.360 \pm 0.000$ | $0.360 \pm 0.000$ |
| LightGBM | $0.360 \pm 0.000$ | $0.360 \pm 0.000$ |
| *Default hyperparameters* | | |
| TabR-S | $0.364 \pm 0.000$ | $0.363 \pm 0.000$ |
| CatBoost | $0.361 \pm 0.000$ | $0.361 \pm 0.000$ |
| XGBoost | $0.362 \pm 0.000$ | $0.362 \pm 0.000$ |
| LightGBM | $0.361 \pm 0.000$ | $0.361 \pm 0.000$ |

### california ↓

| Method | Single model | Ensemble |
|---|---|---|
| *Tuned Hyperparameters* | | |
| MLP | $0.149 \pm 0.002$ | $0.146 \pm 0.001$ |
| MLP-PLR | $0.138 \pm 0.001$ | $0.135 \pm 0.000$ |
| TabR-S | $0.124 \pm 0.001$ | $0.121 \pm 0.000$ |
| TabR | $0.122 \pm 0.001$ | $0.120 \pm 0.000$ |
| CatBoost | $0.129 \pm 0.000$ | $0.128 \pm 0.000$ |
| XGBoost | $0.131 \pm 0.001$ | $0.130 \pm 0.000$ |
| LightGBM | $0.131 \pm 0.001$ | $0.130 \pm 0.000$ |
| *Default hyperparameters* | | |
| TabR-S | $0.124 \pm 0.001$ | $0.122 \pm 0.000$ |
| CatBoost | $0.129 \pm 0.000$ | $0.129 \pm 0.000$ |
| XGBoost | $0.141 \pm 0.000$ | $0.141 \pm 0.000$ |
| LightGBM | $0.135 \pm 0.000$ | $0.135 \pm 0.000$ |

### compass ↑

| Method | Single model | Ensemble |
|---|---|---|
| *Tuned Hyperparameters* | | |
| MLP | $0.768 \pm 0.005$ | $0.776 \pm 0.006$ |
| MLP-PLR | $0.783 \pm 0.007$ | $0.796 \pm 0.006$ |
| TabR-S | $0.863 \pm 0.003$ | $0.870 \pm 0.003$ |
| TabR | $0.871 \pm 0.003$ | $0.879 \pm 0.001$ |
| CatBoost | $0.771 \pm 0.004$ | $0.775 \pm 0.003$ |
| XGBoost | $0.819 \pm 0.005$ | $0.822 \pm 0.003$ |
| LightGBM | $0.771 \pm 0.003$ | $0.773 \pm 0.003$ |
| *Default hyperparameters* | | |
| TabR-S | $0.865 \pm 0.004$ | $0.870 \pm 0.001$ |
| CatBoost | $0.758 \pm 0.002$ | $0.760 \pm 0.001$ |
| XGBoost | $0.751 \pm 0.000$ | $0.751 \pm 0.000$ |
| LightGBM | $0.762 \pm 0.004$ | $0.762 \pm 0.004$ |

### covertype ↑

| Method | Single model | Ensemble |
|---|---|---|
| *Tuned Hyperparameters* | | |
| MLP | $0.929 \pm 0.001$ | $0.934 \pm 0.001$ |
| MLP-PLR | $0.944 \pm 0.002$ | $0.950 \pm 0.001$ |
| TabR-S | $0.953 \pm 0.000$ | $0.954 \pm 0.000$ |
| TabR | $0.957 \pm 0.000$ | $0.958 \pm 0.000$ |
| CatBoost | $0.938 \pm 0.000$ | $0.939 \pm 0.000$ |
| XGBoost | $0.940 \pm 0.000$ | $0.940 \pm 0.000$ |
| LightGBM | $0.939 \pm 0.000$ | $0.939 \pm 0.000$ |
| *Default hyperparameters* | | |
| TabR-S | $0.952 \pm 0.000$ | $0.953 \pm 0.000$ |
| CatBoost | $0.912 \pm 0.000$ | $0.913 \pm 0.000$ |
| XGBoost | $0.927 \pm 0.000$ | $0.927 \pm 0.000$ |
| LightGBM | $0.936 \pm 0.000$ | $0.936 \pm 0.000$ |

### cpu act ↓

| Method | Single model | Ensemble |
|---|---|---|
| *Tuned Hyperparameters* | | |
| MLP | $2.712 \pm 0.207$ | $2.544 \pm 0.052$ |
| MLP-PLR | $2.270 \pm 0.048$ | $2.214 \pm 0.059$ |
| TabR-S | $2.298 \pm 0.053$ | $2.223 \pm 0.050$ |
| TabR | $2.128 \pm 0.078$ | $2.063 \pm 0.050$ |
| CatBoost | $2.124 \pm 0.049$ | $2.109 \pm 0.050$ |
| XGBoost | $2.524 \pm 0.353$ | $2.472 \pm 0.379$ |
| LightGBM | $2.222 \pm 0.089$ | $2.207 \pm 0.092$ |
| *Default hyperparameters* | | |
| TabR-S | $2.285 \pm 0.045$ | $2.214 \pm 0.032$ |
| CatBoost | $2.185 \pm 0.088$ | $2.162 \pm 0.091$ |
| XGBoost | $2.910 \pm 0.463$ | $2.910 \pm 0.486$ |
| LightGBM | $2.274 \pm 0.128$ | $2.274 \pm 0.135$ |

### credit ↑

| Method | Single model | Ensemble |
|---|---|---|
| *Tuned Hyperparameters* | | |
| MLP | $0.772 \pm 0.004$ | $0.774 \pm 0.003$ |
| MLP-PLR | $0.774 \pm 0.004$ | $0.775 \pm 0.006$ |
| TabR-S | $0.773 \pm 0.004$ | $0.774 \pm 0.004$ |
| TabR | $0.772 \pm 0.004$ | $0.775 \pm 0.003$ |
| CatBoost | $0.773 \pm 0.003$ | $0.775 \pm 0.004$ |
| XGBoost | $0.770 \pm 0.003$ | $0.771 \pm 0.003$ |
| LightGBM | $0.769 \pm 0.003$ | $0.773 \pm 0.003$ |
| *Default hyperparameters* | | |
| TabR-S | $0.772 \pm 0.005$ | $0.774 \pm 0.005$ |
| CatBoost | $0.771 \pm 0.005$ | $0.773 \pm 0.002$ |
| XGBoost | $0.772 \pm 0.002$ | $0.772 \pm 0.002$ |
| LightGBM | $0.771 \pm 0.003$ | $0.771 \pm 0.003$ |

### diamonds ↓

| Method | Single model | Ensemble |
|---|---|---|
| *Tuned Hyperparameters* | | |
| MLP | $0.091 \pm 0.002$ | $0.086 \pm 0.000$ |
| MLP-PLR | $0.087 \pm 0.001$ | $0.084 \pm 0.001$ |
| TabR-S | $0.083 \pm 0.001$ | $0.082 \pm 0.000$ |
| TabR | $0.083 \pm 0.001$ | $0.081 \pm 0.000$ |
| CatBoost | $0.084 \pm 0.000$ | $0.083 \pm 0.000$ |
| XGBoost | $0.085 \pm 0.000$ | $0.084 \pm 0.000$ |
| LightGBM | $0.085 \pm 0.000$ | $0.085 \pm 0.000$ |
| *Default hyperparameters* | | |
| TabR-S | $0.084 \pm 0.001$ | $0.082 \pm 0.001$ |
| CatBoost | $0.084 \pm 0.000$ | $0.084 \pm 0.000$ |
| XGBoost | $0.088 \pm 0.000$ | $0.088 \pm 0.000$ |
| LightGBM | $0.086 \pm 0.000$ | $0.086 \pm 0.000$ |

### electricity ↑

| Method | Single model | Ensemble |
|---|---|---|
| *Tuned Hyperparameters* | | |
| MLP | $0.832 \pm 0.004$ | $0.841 \pm 0.002$ |
| MLP-PLR | $0.841 \pm 0.004$ | $0.849 \pm 0.000$ |
| TabR-S | $0.924 \pm 0.003$ | $0.929 \pm 0.001$ |
| TabR | $0.937 \pm 0.002$ | $0.942 \pm 0.000$ |
| CatBoost | $0.880 \pm 0.002$ | $0.882 \pm 0.001$ |
| XGBoost | $0.890 \pm 0.001$ | $0.891 \pm 0.001$ |
| LightGBM | $0.887 \pm 0.001$ | $0.887 \pm 0.001$ |
| *Default hyperparameters* | | |
| TabR-S | $0.887 \pm 0.004$ | $0.893 \pm 0.002$ |
| CatBoost | $0.875 \pm 0.001$ | $0.877 \pm 0.000$ |
| XGBoost | $0.882 \pm 0.000$ | $0.882 \pm 0.000$ |
| LightGBM | $0.890 \pm 0.000$ | $0.890 \pm 0.000$ |

### elevators ↓

| Method | Single model | Ensemble |
|---|---|---|
| *Tuned Hyperparameters* | | |
| MLP | $0.005 \pm 0.000$ | $0.005 \pm 0.000$ |
| MLP-PLR | $0.002 \pm 0.000$ | $0.002 \pm 0.000$ |
| TabR-S | $0.005 \pm 0.000$ | $0.005 \pm 0.000$ |
| TabR | $0.002 \pm 0.000$ | $0.002 \pm 0.000$ |
| CatBoost | $0.002 \pm 0.000$ | $0.002 \pm 0.000$ |
| XGBoost | $0.002 \pm 0.000$ | $0.002 \pm 0.000$ |
| LightGBM | $0.002 \pm 0.000$ | $0.002 \pm 0.000$ |
| *Default hyperparameters* | | |
| TabR-S | $0.005 \pm 0.000$ | $0.005 \pm 0.000$ |
| CatBoost | $0.002 \pm 0.000$ | $0.002 \pm 0.000$ |
| XGBoost | $0.002 \pm 0.000$ | $0.002 \pm 0.000$ |
| LightGBM | $0.002 \pm 0.000$ | $0.002 \pm 0.000$ |

### fifa ↓

| Method | Single model | Ensemble |
|---|---|---|
| *Tuned Hyperparameters* | | |
| MLP | $0.803 \pm 0.013$ | $0.801 \pm 0.015$ |
| MLP-PLR | $0.794 \pm 0.011$ | $0.792 \pm 0.012$ |
| TabR-S | $0.790 \pm 0.012$ | $0.786 \pm 0.012$ |
| TabR | $0.791 \pm 0.014$ | $0.787 \pm 0.016$ |
| CatBoost | $0.783 \pm 0.012$ | $0.782 \pm 0.011$ |
| XGBoost | $0.780 \pm 0.011$ | $0.780 \pm 0.011$ |
| LightGBM | $0.781 \pm 0.012$ | $0.779 \pm 0.012$ |
| *Default hyperparameters* | | |
| TabR-S | $0.790 \pm 0.013$ | $0.786 \pm 0.012$ |
| CatBoost | $0.782 \pm 0.012$ | $0.781 \pm 0.013$ |
| XGBoost | $0.790 \pm 0.012$ | $0.790 \pm 0.013$ |
| LightGBM | $0.780 \pm 0.011$ | $0.780 \pm 0.011$ |

### house 16H ↓

| Method | Single model | Ensemble |
|---|---|---|
| *Tuned Hyperparameters* | | |
| MLP | $0.598 \pm 0.012$ | $0.587 \pm 0.004$ |
| MLP-PLR | $0.594 \pm 0.003$ | $0.589 \pm 0.002$ |
| TabR-S | $0.608 \pm 0.016$ | $0.590 \pm 0.006$ |
| TabR | $0.629 \pm 0.024$ | $0.599 \pm 0.000$ |
| CatBoost | $0.599 \pm 0.005$ | $0.596 \pm 0.003$ |
| XGBoost | $0.591 \pm 0.007$ | $0.585 \pm 0.004$ |
| LightGBM | $0.575 \pm 0.002$ | $0.573 \pm 0.001$ |
| *Default hyperparameters* | | |
| TabR-S | $0.603 \pm 0.015$ | $0.583 \pm 0.003$ |
| CatBoost | $0.591 \pm 0.002$ | $0.590 \pm 0.001$ |
| XGBoost | $0.589 \pm 0.000$ | $0.589 \pm 0.000$ |
| LightGBM | $0.593 \pm 0.000$ | $0.593 \pm 0.000$ |

### house sales ↓

| Method | Single model | Ensemble |
|--------|--------------|----------|
| *Tuned Hyperparameters* | | |
| MLP | $0.181 \pm 0.001$ | $0.178 \pm 0.000$ |
| MLP-PLR | $0.169 \pm 0.001$ | $0.168 \pm 0.000$ |
| TabR-S | $0.169 \pm 0.001$ | $0.166 \pm 0.000$ |
| TabR | $0.164 \pm 0.001$ | $0.161 \pm 0.000$ |
| CatBoost | $0.167 \pm 0.000$ | $0.167 \pm 0.000$ |
| XGBoost | $0.169 \pm 0.000$ | $0.169 \pm 0.000$ |
| LightGBM | $0.169 \pm 0.000$ | $0.169 \pm 0.000$ |
| *Default hyperparameters* | | |
| TabR-S | $0.169 \pm 0.001$ | $0.167 \pm 0.000$ |
| CatBoost | $0.167 \pm 0.000$ | $0.167 \pm 0.000$ |
| XGBoost | $0.179 \pm 0.000$ | $0.179 \pm 0.000$ |
| LightGBM | $0.173 \pm 0.000$ | $0.173 \pm 0.000$ |

### houses ↓

| Method | Single model | Ensemble |
|--------|--------------|----------|
| *Tuned Hyperparameters* | | |
| MLP | $0.233 \pm 0.002$ | $0.227 \pm 0.001$ |
| MLP-PLR | $0.228 \pm 0.002$ | $0.224 \pm 0.000$ |
| TabR-S | $0.199 \pm 0.001$ | $0.196 \pm 0.000$ |
| TabR | $0.201 \pm 0.002$ | $0.197 \pm 0.000$ |
| CatBoost | $0.216 \pm 0.001$ | $0.214 \pm 0.000$ |
| XGBoost | $0.219 \pm 0.001$ | $0.217 \pm 0.000$ |
| LightGBM | $0.219 \pm 0.001$ | $0.217 \pm 0.000$ |
| *Default hyperparameters* | | |
| TabR-S | $0.200 \pm 0.001$ | $0.197 \pm 0.001$ |
| CatBoost | $0.216 \pm 0.000$ | $0.216 \pm 0.000$ |
| XGBoost | $0.234 \pm 0.000$ | $0.234 \pm 0.000$ |
| LightGBM | $0.226 \pm 0.000$ | $0.226 \pm 0.000$ |

### isolet ↓

| Method | Single model | Ensemble |
|--------|--------------|----------|
| *Tuned Hyperparameters* | | |
| MLP | $2.223 \pm 0.189$ | $2.037 \pm 0.106$ |
| MLP-PLR | $2.224 \pm 0.156$ | $2.030 \pm 0.103$ |
| TabR-S | $1.976 \pm 0.174$ | $1.763 \pm 0.152$ |
| TabR | $1.992 \pm 0.181$ | $1.748 \pm 0.143$ |
| CatBoost | $2.867 \pm 0.014$ | $2.848 \pm 0.002$ |
| XGBoost | $2.757 \pm 0.047$ | $2.729 \pm 0.037$ |
| LightGBM | $2.701 \pm 0.030$ | $2.690 \pm 0.029$ |
| *Default hyperparameters* | | |
| TabR-S | $1.995 \pm 0.156$ | $1.754 \pm 0.106$ |
| CatBoost | $2.895 \pm 0.020$ | $2.863 \pm 0.013$ |
| XGBoost | $3.368 \pm 0.010$ | $3.368 \pm 0.011$ |
| LightGBM | $2.953 \pm 0.056$ | $2.953 \pm 0.058$ |

### jannis ↑

| Method | Single model | Ensemble |
|--------|--------------|----------|
| *Tuned Hyperparameters* | | |
| MLP | $0.785 \pm 0.003$ | $0.787 \pm 0.002$ |
| MLP-PLR | $0.799 \pm 0.003$ | $0.804 \pm 0.001$ |
| TabR-S | $0.798 \pm 0.002$ | $0.802 \pm 0.002$ |
| TabR | $0.805 \pm 0.002$ | $0.811 \pm 0.001$ |
| CatBoost | $0.798 \pm 0.002$ | $0.801 \pm 0.001$ |
| XGBoost | $0.797 \pm 0.002$ | $0.800 \pm 0.001$ |
| LightGBM | $0.796 \pm 0.002$ | $0.797 \pm 0.001$ |
| *Default hyperparameters* | | |
| TabR-S | $0.795 \pm 0.002$ | $0.800 \pm 0.001$ |
| CatBoost | $0.795 \pm 0.001$ | $0.797 \pm 0.000$ |
| XGBoost | $0.783 \pm 0.000$ | $0.783 \pm 0.000$ |
| LightGBM | $0.794 \pm 0.000$ | $0.794 \pm 0.000$ |

### kdd ipums la 97-small ↑

| Method | Single model | Ensemble |
|--------|--------------|----------|
| *Tuned Hyperparameters* | | |
| MLP | $0.880 \pm 0.007$ | $0.880 \pm 0.006$ |
| MLP-PLR | $0.883 \pm 0.005$ | $0.883 \pm 0.005$ |
| TabR-S | $0.880 \pm 0.008$ | $0.882 \pm 0.008$ |
| TabR | $0.883 \pm 0.005$ | $0.884 \pm 0.005$ |
| CatBoost | $0.879 \pm 0.009$ | $0.880 \pm 0.010$ |
| XGBoost | $0.883 \pm 0.009$ | $0.883 \pm 0.008$ |
| LightGBM | $0.879 \pm 0.007$ | $0.880 \pm 0.007$ |
| *Default hyperparameters* | | |
| TabR-S | $0.877 \pm 0.006$ | $0.878 \pm 0.007$ |
| CatBoost | $0.879 \pm 0.007$ | $0.881 \pm 0.007$ |
| XGBoost | $0.883 \pm 0.010$ | $0.883 \pm 0.011$ |
| LightGBM | $0.884 \pm 0.005$ | $0.884 \pm 0.005$ |

### medical charges ↓

| Method | Single model | Ensemble |
|--------|--------------|----------|
| *Tuned Hyperparameters* | | |
| MLP | $0.082 \pm 0.000$ | $0.081 \pm 0.000$ |
| MLP-PLR | $0.081 \pm 0.000$ | $0.081 \pm 0.000$ |
| TabR-S | $0.081 \pm 0.000$ | $0.081 \pm 0.000$ |
| TabR | $0.081 \pm 0.000$ | $0.081 \pm 0.000$ |
| CatBoost | $0.082 \pm 0.000$ | $0.082 \pm 0.000$ |
| XGBoost | $0.082 \pm 0.000$ | $0.082 \pm 0.000$ |
| LightGBM | $0.082 \pm 0.000$ | $0.082 \pm 0.000$ |
| *Default hyperparameters* | | |
| TabR-S | $0.082 \pm 0.000$ | $0.081 \pm 0.000$ |
| CatBoost | $0.082 \pm 0.000$ | $0.082 \pm 0.000$ |
| XGBoost | $0.084 \pm 0.000$ | $0.084 \pm 0.000$ |
| LightGBM | $0.083 \pm 0.000$ | $0.083 \pm 0.000$ |

### nyc-taxi-green-dec-2016 ↓

| Method | Single model | Ensemble |
|---|---|---|
| Tuned Hyperparameters | | |
| MLP | $0.397 \pm 0.001$ | $0.391 \pm 0.001$ |
| MLP-PLR | $0.368 \pm 0.002$ | $0.364 \pm 0.000$ |
| TabR-S | $0.358 \pm 0.022$ | $0.338 \pm 0.003$ |
| TabR | $0.372 \pm 0.009$ | $0.350 \pm 0.003$ |
| CatBoost | $0.365 \pm 0.001$ | $0.363 \pm 0.000$ |
| XGBoost | $0.379 \pm 0.000$ | $0.379 \pm 0.000$ |
| LightGBM | $0.369 \pm 0.000$ | $0.368 \pm 0.000$ |
| Default hyperparameters | | |
| TabR-S | $0.389 \pm 0.001$ | $0.385 \pm 0.000$ |
| CatBoost | $0.366 \pm 0.000$ | $0.366 \pm 0.000$ |
| XGBoost | $0.386 \pm 0.000$ | $0.386 \pm 0.000$ |
| LightGBM | $0.372 \pm 0.000$ | $0.372 \pm 0.000$ |

### particulate-matter-ukair-2017 ↓

| Method | Single model | Ensemble |
|---|---|---|
| Tuned Hyperparameters | | |
| MLP | $0.377 \pm 0.001$ | $0.374 \pm 0.000$ |
| MLP-PLR | $0.367 \pm 0.001$ | $0.366 \pm 0.000$ |
| TabR-S | $0.361 \pm 0.000$ | $0.359 \pm 0.000$ |
| TabR | $0.360 \pm 0.000$ | $0.358 \pm 0.000$ |
| CatBoost | $0.365 \pm 0.000$ | $0.364 \pm 0.000$ |
| XGBoost | $0.364 \pm 0.000$ | $0.364 \pm 0.000$ |
| LightGBM | $0.364 \pm 0.000$ | $0.363 \pm 0.000$ |
| Default hyperparameters | | |
| TabR-S | $0.361 \pm 0.001$ | $0.359 \pm 0.000$ |
| CatBoost | $0.366 \pm 0.000$ | $0.366 \pm 0.000$ |
| XGBoost | $0.368 \pm 0.000$ | $0.368 \pm 0.000$ |
| LightGBM | $0.366 \pm 0.000$ | $0.366 \pm 0.000$ |

### phoneme ↑

| Method | Single model | Ensemble |
|---|---|---|
| Tuned Hyperparameters | | |
| MLP | $0.851 \pm 0.014$ | $0.861 \pm 0.013$ |
| MLP-PLR | $0.866 \pm 0.012$ | $0.875 \pm 0.012$ |
| TabR-S | $0.878 \pm 0.010$ | $0.884 \pm 0.005$ |
| TabR | $0.877 \pm 0.009$ | $0.885 \pm 0.007$ |
| CatBoost | $0.883 \pm 0.012$ | $0.890 \pm 0.005$ |
| XGBoost | $0.868 \pm 0.017$ | $0.877 \pm 0.016$ |
| LightGBM | $0.870 \pm 0.013$ | $0.873 \pm 0.013$ |
| Default hyperparameters | | |
| TabR-S | $0.877 \pm 0.007$ | $0.880 \pm 0.003$ |
| CatBoost | $0.879 \pm 0.011$ | $0.881 \pm 0.012$ |
| XGBoost | $0.870 \pm 0.016$ | $0.870 \pm 0.016$ |
| LightGBM | $0.874 \pm 0.007$ | $0.874 \pm 0.007$ |

### pol ↓

| Method | Single model | Ensemble |
|---|---|---|
| Tuned Hyperparameters | | |
| MLP | $5.659 \pm 0.543$ | $5.143 \pm 0.579$ |
| MLP-PLR | $2.615 \pm 0.137$ | $2.445 \pm 0.073$ |
| TabR-S | $6.071 \pm 0.537$ | $5.558 \pm 0.404$ |
| TabR | $2.577 \pm 0.169$ | $2.326 \pm 0.058$ |
| CatBoost | $3.632 \pm 0.101$ | $3.551 \pm 0.090$ |
| XGBoost | $4.296 \pm 0.064$ | $4.255 \pm 0.049$ |
| LightGBM | $4.232 \pm 0.337$ | $4.188 \pm 0.311$ |
| Default hyperparameters | | |
| TabR-S | $6.200 \pm 0.396$ | $5.804 \pm 0.248$ |
| CatBoost | $4.479 \pm 0.051$ | $4.400 \pm 0.039$ |
| XGBoost | $5.249 \pm 0.183$ | $5.249 \pm 0.197$ |
| LightGBM | $4.382 \pm 0.195$ | $4.382 \pm 0.210$ |

### rl ↑

| Method | Single model | Ensemble |
|---|---|---|
| Tuned Hyperparameters | | |
| MLP | $0.671 \pm 0.013$ | $0.677 \pm 0.013$ |
| MLP-PLR | $0.744 \pm 0.019$ | $0.767 \pm 0.027$ |
| TabR-S | $0.874 \pm 0.008$ | $0.880 \pm 0.006$ |
| TabR | $0.884 \pm 0.016$ | $0.891 \pm 0.013$ |
| CatBoost | $0.790 \pm 0.007$ | $0.793 \pm 0.005$ |
| XGBoost | $0.797 \pm 0.012$ | $0.799 \pm 0.012$ |
| LightGBM | $0.781 \pm 0.010$ | $0.787 \pm 0.007$ |
| Default hyperparameters | | |
| TabR-S | $0.871 \pm 0.008$ | $0.876 \pm 0.007$ |
| CatBoost | $0.785 \pm 0.010$ | $0.790 \pm 0.004$ |
| XGBoost | $0.775 \pm 0.003$ | $0.775 \pm 0.003$ |
| LightGBM | $0.778 \pm 0.003$ | $0.778 \pm 0.003$ |

### road-safety ↑

| Method | Single model | Ensemble |
|---|---|---|
| Tuned Hyperparameters | | |
| MLP | $0.786 \pm 0.001$ | $0.789 \pm 0.000$ |
| MLP-PLR | $0.785 \pm 0.002$ | $0.789 \pm 0.001$ |
| TabR-S | $0.840 \pm 0.001$ | $0.844 \pm 0.000$ |
| TabR | $0.837 \pm 0.001$ | $0.843 \pm 0.000$ |
| CatBoost | $0.801 \pm 0.001$ | $0.802 \pm 0.000$ |
| XGBoost | $0.810 \pm 0.002$ | $0.813 \pm 0.000$ |
| LightGBM | $0.798 \pm 0.001$ | $0.800 \pm 0.000$ |
| Default hyperparameters | | |
| TabR-S | $0.791 \pm 0.003$ | $0.796 \pm 0.003$ |
| CatBoost | $0.792 \pm 0.001$ | $0.793 \pm 0.000$ |
| XGBoost | $0.796 \pm 0.000$ | $0.796 \pm 0.000$ |
| LightGBM | $0.803 \pm 0.000$ | $0.803 \pm 0.000$ |

### sulfur ↓

| Method | Single model | Ensemble |
|---|---|---|
| *Tuned Hyperparameters* | | |
| MLP | $0.022 \pm 0.002$ | $0.021 \pm 0.002$ |
| MLP-PLR | $0.020 \pm 0.002$ | $0.019 \pm 0.003$ |
| TabR-S | $0.022 \pm 0.002$ | $0.021 \pm 0.002$ |
| TabR | $0.022 \pm 0.003$ | $0.020 \pm 0.003$ |
| CatBoost | $0.019 \pm 0.002$ | $0.019 \pm 0.002$ |
| XGBoost | $0.020 \pm 0.002$ | $0.020 \pm 0.002$ |
| LightGBM | $0.020 \pm 0.002$ | $0.020 \pm 0.002$ |
| *Default hyperparameters* | | |
| TabR-S | $0.021 \pm 0.003$ | $0.021 \pm 0.002$ |
| CatBoost | $0.019 \pm 0.002$ | $0.019 \pm 0.003$ |
| XGBoost | $0.022 \pm 0.002$ | $0.022 \pm 0.002$ |
| LightGBM | $0.021 \pm 0.001$ | $0.021 \pm 0.001$ |

### superconduct ↓

| Method | Single model | Ensemble |
|---|---|---|
| *Tuned Hyperparameters* | | |
| MLP | $10.724 \pm 0.062$ | $10.455 \pm 0.005$ |
| MLP-PLR | $10.566 \pm 0.058$ | $10.334 \pm 0.028$ |
| TabR-S | $10.884 \pm 0.107$ | $10.480 \pm 0.028$ |
| TabR | $10.384 \pm 0.056$ | $10.137 \pm 0.023$ |
| CatBoost | $10.242 \pm 0.022$ | $10.212 \pm 0.006$ |
| XGBoost | $10.161 \pm 0.020$ | $10.141 \pm 0.002$ |
| LightGBM | $10.163 \pm 0.012$ | $10.155 \pm 0.005$ |
| *Default hyperparameters* | | |
| TabR-S | $10.812 \pm 0.110$ | $10.423 \pm 0.046$ |
| CatBoost | $10.263 \pm 0.028$ | $10.222 \pm 0.006$ |
| XGBoost | $10.736 \pm 0.000$ | $10.736 \pm 0.000$ |
| LightGBM | $10.471 \pm 0.000$ | $10.471 \pm 0.000$ |

### visualizing soil ↓

| Method | Single model | Ensemble |
|---|---|---|
| *Tuned Hyperparameters* | | |
| MLP | $0.138 \pm 0.012$ | $0.132 \pm 0.010$ |
| MLP-PLR | $0.158 \pm 0.067$ | $0.144 \pm 0.060$ |
| TabR-S | $0.398 \pm 0.352$ | $0.387 \pm 0.375$ |
| TabR | $0.227 \pm 0.264$ | $0.202 \pm 0.147$ |
| CatBoost | $0.055 \pm 0.006$ | $0.047 \pm 0.006$ |
| XGBoost | $0.176 \pm 0.071$ | $0.154 \pm 0.054$ |
| LightGBM | $0.062 \pm 0.016$ | $0.062 \pm 0.017$ |
| *Default hyperparameters* | | |
| TabR-S | $0.327 \pm 0.254$ | $0.310 \pm 0.257$ |
| CatBoost | $0.064 \pm 0.005$ | $0.058 \pm 0.005$ |
| XGBoost | $0.066 \pm 0.009$ | $0.066 \pm 0.010$ |
| LightGBM | $0.061 \pm 0.013$ | $0.061 \pm 0.014$ |

### wine ↑

| Method | Single model | Ensemble |
|---|---|---|
| *Tuned Hyperparameters* | | |
| MLP | $0.769 \pm 0.015$ | $0.784 \pm 0.010$ |
| MLP-PLR | $0.771 \pm 0.016$ | $0.783 \pm 0.014$ |
| TabR-S | $0.794 \pm 0.011$ | $0.805 \pm 0.006$ |
| TabR | $0.780 \pm 0.015$ | $0.795 \pm 0.012$ |
| CatBoost | $0.799 \pm 0.013$ | $0.806 \pm 0.010$ |
| XGBoost | $0.795 \pm 0.018$ | $0.801 \pm 0.019$ |
| LightGBM | $0.789 \pm 0.016$ | $0.793 \pm 0.011$ |
| *Default hyperparameters* | | |
| TabR-S | $0.791 \pm 0.012$ | $0.800 \pm 0.008$ |
| CatBoost | $0.796 \pm 0.010$ | $0.799 \pm 0.010$ |
| XGBoost | $0.796 \pm 0.010$ | $0.796 \pm 0.010$ |
| LightGBM | $0.798 \pm 0.004$ | $0.798 \pm 0.004$ |

### wine quality ↓

| Method | Single model | Ensemble |
|---|---|---|
| *Tuned Hyperparameters* | | |
| MLP | $0.672 \pm 0.015$ | $0.659 \pm 0.016$ |
| MLP-PLR | $0.654 \pm 0.018$ | $0.634 \pm 0.018$ |
| TabR-S | $0.632 \pm 0.010$ | $0.620 \pm 0.010$ |
| TabR | $0.641 \pm 0.011$ | $0.620 \pm 0.007$ |
| CatBoost | $0.609 \pm 0.013$ | $0.606 \pm 0.014$ |
| XGBoost | $0.604 \pm 0.013$ | $0.602 \pm 0.014$ |
| LightGBM | $0.613 \pm 0.014$ | $0.612 \pm 0.014$ |
| *Default hyperparameters* | | |
| TabR-S | $0.628 \pm 0.015$ | $0.614 \pm 0.015$ |
| CatBoost | $0.628 \pm 0.012$ | $0.626 \pm 0.012$ |
| XGBoost | $0.648 \pm 0.008$ | $0.648 \pm 0.008$ |
| LightGBM | $0.641 \pm 0.011$ | $0.641 \pm 0.012$ |

### year ↓

| Method | Single model | Ensemble |
|---|---|---|
| *Tuned Hyperparameters* | | |
| MLP | $8.964 \pm 0.018$ | $8.901 \pm 0.003$ |
| MLP-PLR | $8.927 \pm 0.013$ | $8.901 \pm 0.006$ |
| TabR-S | $9.007 \pm 0.015$ | $8.913 \pm 0.009$ |
| TabR | $8.972 \pm 0.010$ | $8.917 \pm 0.003$ |
| CatBoost | $9.037 \pm 0.007$ | $9.005 \pm 0.003$ |
| XGBoost | $9.031 \pm 0.003$ | $9.024 \pm 0.001$ |
| LightGBM | $9.020 \pm 0.002$ | $9.013 \pm 0.001$ |
| *Default hyperparameters* | | |
| TabR-S | $9.067 \pm 0.022$ | $8.893 \pm 0.008$ |
| CatBoost | $9.073 \pm 0.008$ | $9.046 \pm 0.001$ |
| XGBoost | $9.376 \pm 0.000$ | $9.376 \pm 0.000$ |
| LightGBM | $9.214 \pm 0.000$ | $9.214 \pm 0.000$ |

| yprop 4 1 ↓ | | |
|---|---|---|
| Method | Single model | Ensemble |
| *Tuned Hyperparameters* | | |
| MLP | $0.027 \pm 0.001$ | $0.027 \pm 0.001$ |
| MLP-PLR | $0.027 \pm 0.001$ | $0.027 \pm 0.001$ |
| TabR-S | $0.027 \pm 0.000$ | $0.027 \pm 0.001$ |
| TabR | $0.027 \pm 0.000$ | $0.027 \pm 0.000$ |
| CatBoost | $0.027 \pm 0.000$ | $0.027 \pm 0.001$ |
| XGBoost | $0.027 \pm 0.001$ | $0.027 \pm 0.001$ |
| LightGBM | $0.027 \pm 0.000$ | $0.027 \pm 0.000$ |
| *Default hyperparameters* | | |
| TabR-S | $0.027 \pm 0.001$ | $0.027 \pm 0.001$ |
| CatBoost | $0.027 \pm 0.000$ | $0.027 \pm 0.000$ |
| XGBoost | $0.027 \pm 0.001$ | $0.027 \pm 0.001$ |
| LightGBM | $0.027 \pm 0.000$ | $0.027 \pm 0.000$ |

