# OpenReview forum: "TabR: Tabular Deep Learning Meets Nearest Neighbors"
_ICLR.cc/2024/Conference — ICLR 2024 poster_

### Official Review · Reviewer_EnVq · 2023-10-16

**Soundness:** 3 good
**Presentation:** 2 fair
**Contribution:** 2 fair
**Rating:** 3
**Confidence:** 4

**Summary:**

The authors meticulously designed a supervised deep learning model for tabular data prediction, which operates in a retrieval-like manner. It outperformed tree-based models on middle-scale datasets, as well as other retrieval-based deep learning tabular learning models. To achieve this, they introduced a k-Nearest-Neighbors-like idea in model design.

**Strengths:**

- As emphasized by the authors, their method has managed to outperform tree based models like xgboost on middle-scale datasets.

- Overall, the presentation is clear, and the experiments are comprehensive. The details are clear and the model is highly reproducible.

- This model is the best-performing retrieval based model.

**Weaknesses:**

- The motivations behind the module designs are not entirely clear. It appears that the authors made meticulous module (equation) optimization based on its performance on some datasets empirically. Then:

(1) Why does employing the L2 distance, instand of the dot product, lead to improved performance (as shown in Eq. 3)?

(2) Why is the T function required to use LinearWithoutBias?

(3) We are uncertain about the robustness of the designed modules. If the dataset characteristics are changed, is it likely that the performance rankings will change significantly? The performances only on middle-sized datasets cannot show the robustness.

...

I suggest that providing a theoretical analysis or intuitive motivation would enhance the reader's understanding of those details.

- Some sota DL approaches are not compared, such as T2G-Former (an improved version of FTT)[1], TabPFN [2], and TANGOS [3]. Especially, TabFPN is relatively similar to TabR. These papers are current SOTA, and may outperforms tree based models.

[1] T2G-Former: Organizing tabular features into relation graphs promotes heterogeneous feature interaction

[2] TabPFN: A Transformer That Solves Small Tabular Classification Problems in a Second

[3] TANGOS: Regularizing Tabular Neural Networks through Gradient Orthogonalization and Specialization

- The major comparison lies among middle-scale datasets, accompanied with some results on few other datasets shown in Table 3. In scenarios involving sparse, medium, and dense data-distributed datasets (which typically occur in small, medium-sized, and large-sized datasets, respectively), I suppose that there exists a variance in the nearest neighbor retrieval pattern. Hence, conducting tests solely on medium-sized datasets may not suffice. Furthermore, the issue of inefficiency when dealing with large-scaled datasets appears to have hindered the authors from proving the method's effectiveness in large-scaled datasets.

- The method proposed by the authors appears to have achieved slight performance advantages on certain datasets (although some SOTA are not compared). However, due to the lacks of explanation for the model details that are designed empirically, it seems unnecessary and risky to apply this method in real-world scenarios (for example, it's unclear whether L2 distance may fail when uninformative features are present; or, for instance, when a table has a feature with values [f_1, f_2, f_3, ..., f_n], and we take the logarithm of these values [log f_1, log f_2, log f_3, ..., log f_n] or their reciprocals, the method may perform poorly in such cases).

**Questions:**

- In Section 3.1, you mentioned "continuous (i.e., continuous) features." Could this be a typographical error?

- I am curious if the L2 design is sensitive to uninformative features? You can offer some analysis or conduct experiments by adding some gaussian noise columns (uninformative features are commonly seen in tabular datasets) and observe the change of performances. Some transformation like logarithm may impact the results.

- Some questions in weakness.

---

> ### Author Response · Authors · 2023-11-18
> **Rebuttal (part 1/N)**
>
> We thank the reviewer for the detailed review. Below, we address all the raised concerns and questions.
>
> > Some sota DL approaches are not compared ... T2G-Former ... TabPFN ... TANGOS ... These papers are current SOTA, and may outperforms tree based models.
>
> Our reply to this comment consists of three paragraphs.
>
> **First**, using the benchmark from `[1]` (43 classification and regression tasks; used in Figure 1 and Table 5 in our submission) we show that:
> 1. **Our model TabR significantly outperforms all the suggested baselines**.
> 2. **The baselines used in our paper are representative**.
>
> The results:
>
> | Model `[citation]`     | Average rank (+- std) | Wins/Ties/Losses vs TabR | Not applicable |
> | ---------------------- | --------------------- | ------------------------ | -------------- |
> | **Used in our paper:** |                       |                          |               |
> | TabR `[ours]`          | 1.36 (± 0.65)        | 0/43/0                   | 0              |
> | XGBoost `[11]`         | 1.93 (± 0.94)        | 7/13/23                     | 0              |
> | MLP-PLR `[2]`          | 1.95 (± 0.92)        | 5/19/19                  | 0              |
> | FT-Transformer `[4]`   | 2.45 (± 1.20)        | 2/14/27                  | 0              |
> | MLP                    | 3.07 (± 1.55)        | 2/12/29                  | 0              |
> | **Additional:**        |                       |                          |               |
> | T2G-Former `[10]`      | 2.76 (± 1.56)        |  3/14/23                | 3           |
> | MLP + TANGOS `[9]`     | 3.29 (± 1.69)        | 2/13/28                | 0           |
> | TabPFN `[8]`           | 4.38 (± 1.98)        | 0/4/5                | 34           |
>
> Technical details:
> - "Not applicable" (the righmost column) means that the method is fundamentally not applicable to a given task or it does not fit into A100-80GB or (the case of T2G-Former) the hyperparameter tuning did not finish on time.
> - If a method is not applicable to a task, it is assigned the largest rank.
>
> **Second**, a closer look at the suggested baselines reveals things that are not consistent with being SOTA *in the scope and niche of our paper*:
> - TANGOS
>     - **Different niche:** TANGOS is not a model, it is a regularization method, which is completely orthogonal to our niche (model).
>     - **Different scope**: the original work mostly focuses on small data with only two datasets having more than 1000 objects.
>     - **Significant limitations:** the method seems to make training order(s) of magnitude slower (40x times slower on average on our datasets).
> - TabPFN
>     - **Different scope:** the work fully focuses on small (<1000 objects) classification tasks.
>     - **Significant limitations:** not applicable to regression tasks, not applicable to more than 10 classes and more than 100 features, high memory consumption.
> - T2G-Former
>     - **Already tested** in its paper on many of the datasets used in our paper and in `[2]` (where embeddings for continuous features and, in particular, MLP-PLR were introduced). The results from `[10]` and `[2]` indicate that models from `[2]` perform at least on par (and often better) while being more efficient. And, the above table confirms that MLP-PLR is indeed a good baseline, while T2G-Former did not generalize well to the new benchmark.
>
> **Third**, in the new PDF, in Section 2 Paragraph "Tabular deep learning", we improved the communication of the niche of our work and, in particular, cited TANGOS and TabPFN.

---

> ### Author Response · Authors · 2023-11-18
> **Rebuttal (part 2/N)**
>
> > **[REGARDING THE BENCHMARK AND DATASET SIZES]**
> >
> > If the dataset characteristics are changed, is it likely that the performance rankings will change significantly?
> > ...
> > The performances only on middle-sized datasets cannot show the robustness
> > ...
> > conducting tests solely on medium-sized datasets may not suffice
> > ...
> > **FEW** other datasets
>
> Our understanding is that the above quoted comments express uncertainty about our benchmark and whether it is comprehensive enough to support our story.
>
> We did our best to build a **comprehensive, challenging and representative benchmark** to support our claims. Our reply consists of several points.
>
> **First**, let us show that **our experimental setup is competitive** by comparing it to other works on tabular DL:
>
> | Model `[citation]`   | The number of datasets used in the paper                                | Max. train set size               |
> | -------------------- | ----------------------------------------------------------------------- |:------------------------------- |
> | **TabR** `[ours]`    | **~50 datasets (roughly, `[1]` + some datasets from `[2]` and `[3]`)**  | **2.9M** (+ 10M in the rebuttal) |
> | NODE `[7]`           | 6 datasets                                                              | 10M                            |
> | TabNet  `[5]`        | <10 datasets                                                            | 10M                            |
> | FT-Transformer `[4]` | 11 datasets                                                             | 723K                            |
> | SAINT `[6]`          | 30 datasets                                                             | 321K                            |
> | \<survey\> `[12]`    | 5 datasets                                                              | 10M                            |
> | MLP-PLR `[2]`        | 11 datasets                                                             | 723K                            |
> | TabPFN `[8]`         | 67 small datasets (different scope)                                     | 1K                            |
> | \<benchmark\> `[1]`  | <50 *unique* datasets (>50 with multiple versions of the same datasets) | 50K                            |
> | TANGOS `[9]`         | 20 datasets                                                             | <100K                            |
> | T2G-FORMER `[10]`    | 12 datasets                                                             | 320K                            |
>
> **Second,** to illustrate that **our study is based on many diverse datasets**, we provide the structure of our benchmark:
> - The "default" benchmark based on prior work (mainly `[2]`):
>     - 11 datasets (binary classification, multiclass classification and regression)
>     - from 6.4K to 723K training objects
>     - from 8 to 136 features
> - The widely known and cited benchmark from `[1]`:
>     - 43 datasets (binary classification and regression)
>     - from 1.7K to 50K training objects
>     - from 3 to 613 features
> - One weather prediction regression task from `[3]`: 2.9M training objects and 119 features.
>
> **Third, crucially**, to make the benchmark challenging for TabR, we ensured that **there are many challenging tasks that favour GBDT over prior neural networks.** In particular:
> - On the benchmark from `[1]`, GBDT was shown to be superior to neural networks.
> - `[2]` also contains multiple GBDT-friendly tasks.
> - The big weather prediction dataset `[3]` favors GBDT over simple neural networks.
>
> **To sum up**, we truly hope that the above points demonstrate why we consider our benchmark to be of reasonable size, diversity and complexity to support the claims that we make in the abstract and in the list of contribution in Section 1.

---

> ### Author Response · Authors · 2023-11-18
> **Rebuttal (part 3/N)**
>
> > Furthermore, the issue of inefficiency when dealing with large-scaled datasets appears to have hindered the authors from proving the method's effectiveness in large-scaled datasets.
>
> However, **in the paper, we used datasets with up to three million training objects.**
> To further illustrate the applicability of TabR to large datasets, below, we report the performance of TabR on the Higgs dataset with **ten million training objects:**
>
> | Model   | Hyperparameter tuning time | Accuracy   |
> | ------- |:-------------------------- |:---------- |
> | XGBoost | 1.5 days                   | 0.783      |
> | TabR    | **0**                      | **0.797** |
>
> Technical details:
> - We used TabR with the default hyperparameters with the simplest `Linear-ReLU` embeddings for continuous features from `[2]` without any hyperparameter tuning (`d_embedding = 32`).
> - We applied the "context freeze" technique described in Section 5.1 of the paper for faster training.
>
> > The motivations behind the module designs are not entirely clear ... I suggest that providing a theoretical analysis or intuitive motivation would enhance the reader's understanding of those details.
>
> Providing clear motivation is of high importance to us. Our reply consists of two parts.
>
> **First**, *based on the content from the submission*, let us show how **the design steps behind TabR (Section 3.2) are motivated formally, informally and by prior work:**
>
> - **Step-0 (baseline) is fully motivated by related work.**
>     - (Section 3.2, Paragraph "Step-0") *"the self-attention operation was often used in prior work ... Then, instantiating retrieval module R as the vanilla self-attention ... is a reasonable baseline"*
> - **Step-1 (using labels) is highly intuitive:** labels is an extremely valuable signal, so we try using them to improve the model.
>     - (Section 3.2, Paragraph "Step-1") *"A natural attempt to improve the Step-0 configuration is to utilize labels of the context objects"*
> - **Step-2 (modifying the similarity module $\mathcal{S})$ has a whole range of motivations** provided in the main text ("Step-1" and "Step-2" paragraphs of Section 3.2) and appendix (Section A.1 referenced from Section 5.3, Section A.3 referenced from Section 3.2):
>     - **Intuitively,** it is motivated by the outcome of Step-1:  (Section 3.2, Paragraph "Step-1") *"Perhaps, the similarity module S taken from the vanilla attention does not allow benefiting from such a valuable signal as labels."*
>     - Section A.1.1 provides **formal motivation** for removing the query representations.
>     - Section A.1.1 provides **informal motivation** for using L2 instead of the dot product.
>     - **Empirically**, it is motivated by the exhaustive (literally) ablation study in Section A.3, which is references from the main text.
> - **Step-3 (modifying the value module $\mathcal{V})$**
>     - **Formal motivation:** using $\tilde{x}$ in the value module makes it strictly more expressive: (Section 3.2 Paragraph "Step-3"): *"we make the value module V more expressive by taking the target object’s representation $\tilde{x}$ into account"*
>     - **Related work:** (Section 3.2 Paragraph "Step-3") *"we take inspiration from DNNR -- the recently proposed generalization of the kNN algorithm"*
>     - **Intuitive interpretation:** (Section 3.2 Paragraph "Step-3"): *"\<the decomposition into the "raw" and "correction" terms\>"*
>     - **Quantifying the intuitive interpretation:** Section A.2 referenced from Section 5.3
> - **Step-4** is the only purely technical step. However, it is not uncommon for a new architecture to require different technical defaults.
>
> **Second,** we summarize how we further improve the communication and make the available content more visible in the new PDF:
> - In Section 3.2 Paragraph "Step-2", we added a reference to Section A.1.
> - In Section 3.2 Paragraph "Step-3", we added a reference to Section A.2.
> - In Section A.2.1, we added extended comments on Section 3.2 Paragraph "Step-3".
> - In Section A.8, we added a minor technical note on Section 3.2 Paragraph "Step-3".

---

> ### Author Response · Authors · 2023-11-18
> **Rebuttal (part 4/N)**
>
> > It appears that the authors made meticulous module (equation) optimization based on its performance on some datasets empirically.
>
> We consider the design-step-based storytelling to be a purely stylistic choice resembling an ablation study, and this is not uncommon in literature, for example, see Section 2 in (ConvNeXt) "A ConvNet for the 2020s", CVPR 2022, Liu et al. In our specific case, we find it to be a productive way to highlight the important insights behind TabR.
>
> > Why does employing the L2 distance, instand of the dot product, lead to improved performance (as shown in Eq. 3)?
>
> **The changes introduced in Equation 3 are discussed in the paper** in Section 3.2 Paragraph "Step-2" and Section A.1 referenced from Section 5.3 (and, in the new PDF, from Section 3.2):
>
> - (Section 3.2 Paragraph "Step-2") Using L2 alone is not enough: *"removing any of the three ingredients (context labels, key-only representation, L2 distance) results in a performance drop back to the level of MLP"*.
> - (Section A.1.1) Because of specific details of TabR's encoder and the nature of tabular data, L2 is a more reasonable default choice for TabR.
> - (Section A.1.2) Using L2 leads to more diverse neighbor patterns, which, without any additional assumptions, may be a better default behavior for retrieval-based models.
> - (Section A.1.3) For several datasets, we empirically observed that TabR with L2 was capable of uncovering much better neighbors than with the dot product.
>
> Also, in Section 3.2 Paragraph "Step-2", we add that L2 is just a reasonable default choice: *"While the L2 distance is unlikely to be the universally best choice for problems (even within the tabular domain), it seems to be a reasonable default choice for tabular data problems."*
>
> > Why is the T function required to use LinearWithoutBias?
>
> In the new PDF, we clarified this in Section A.8: using Linear instead of LinearWithoutBias would be redundant because of the term $W_Y$ that already implicitly contains the bias:
>
> $\mathcal{V}(\tilde{x}, \tilde{x}_i, y_i) = W_Y(y_i) + T(W_K(\tilde{x}) - W_K(\tilde{x}_i))$
>
> And, just in case, we avoid this redundancy.

---

> ### Author Response · Authors · 2023-11-18
> **Rebuttal (part 5/5, N=5)**
>
> > **[REGARDING ROBUSTNESS OF L2 TO UNINFORMATIVE AND IRREGULARLY DISTRIBUTED FEATURES]**
> >
> > it's unclear whether L2 distance may fail when uninformative features are present ... You can offer some analysis or conduct experiments by adding some gaussian noise columns
> > ...
> > or, for instance, when a table has a feature ... and we take the logarithm of these values ...
> > Some transformation like logarithm may impact the results
>
> Below, we provide results indicating that **the presense of L2 does not make TabR any worse than alternatives**, (dot-product-based TabR or simple MLP) in the presence of uninformative features or after the logarithmic transformation.
> Moreover, the alternatives continue lagging significantly behind TabR.
>
> Technical setup:
> - Datasets (see Table 1 in the paper): CA, HO, AD, DI, HI.
> - We use TabR with the default hyperparameters.
> - The uniformative (noisy) features are added as new features.
> - For the logarithmic transformation, original features are replaced with the transformed ones.
>
> The table below reports the average (over datasets) difference in `%` compared to the metric of the corresponding model trained on the original unmodified datasets.
>
> | Ratio of new uninformative features: | 0.1    | 0.25   | 0.5    |
> |:------------------------------------ |:------ |:------ |:------ |
> | TabR                                 | -0.6%  | -1.9%  | -2.87% |
> | TabR (dot product)                   | -1.27% | -2.26% | -3.77% |
> | MLP                                  | -1.32% | -2.63% | -4.37% |
>
> | Ratio of log-transformed features | 0.1    | 0.25   | 0.5    |
> |:--------------------------------- |:------ |:------ |:------ |
> | TabR                              | -0.15% | -0.77% | -1.07% |
> | TabR (dot product)                | -0.23% | -0.18% | -0.39% |
> | MLP                               | -0.05% | -0.55% | -0.45% |
>
> **Importantly**, in both of the above regimes, TabR continues to *significantly* outperform alternatives.
> For example, this is how the second of the above tables would look like if we reported the differences relative to the *same* baseline, namely, to TabR trained on unmodified original data:
>
> | Ratio of log-transformed features | 0.1    | 0.25   | 0.5    |
> |:--------------------------------- |:------ |:------ |:------ |
> | TabR                              | -0.15% | -0.77% | -1.07% |
> | TabR (dot product)                | -6.95% | -6.95% | -7.17% |
> | MLP                               | -6.65% | -7.23% | -7.15% |
>
> Note that, currently, TabR uses a simple MLP-like encoder, and naturally inherits all properties related to robustness to various data challenges  such as uniformative features, irregularly distributed features, etc. `[1]`, `[13]`, `[14]`. In particular:
> - These properties explain why, in the above table, there is a performance loss for all DL models regardless of the usage of L2.
> - It is also known how to avoid/alleviate these problems: by using transformer-like encoders `[1]`, `[13]` or by using embeddings for continuous features `[2]`.
>
> **References**
> - `[1]` "Why do tree-based models still outperform deep learning on tabular data?", NeurIPS 2022 Datasets and Benchmarks, Grinsztajn et al.
> - `[2]` "On Embeddings for Numerical Features in Tabular Deep Learning", NeurIPS 2022, Gorishniy et al.
> - `[3]` "Shifts: A Dataset of Real Distributional Shift Across Multiple Large-Scale Tasks", NeurIPS 2021 Datasets and Benchmarks , Malinin et al.
> - `[4]` "Revisiting Deep Learning Models for Tabular Data", NeurIPS 2021, Gorishniy et al.
> - `[5]` "TabNet: Attentive Interpretable Tabular Learning", AAAI 2021, Arik et al.
> - `[6]` "SAINT: Improved Neural Networks for Tabular Data via Row Attention and Contrastive Pre-Training", ICLR 2022 submission, Somepalli et al.
> - `[7]` "Neural Oblivious Decision Ensembles for Deep Learning on Tabular Data", ICLR 2020, Popov et al.
> - `[8]` "TabPFN: A Transformer That Solves Small Tabular Classification Problems in a Second", ICLR 2023, Hollmann et al.
> - `[9]` "TANGOS: Regularizing Tabular Neural Networks through Gradient Orthogonalization and Specialization", ICLR 2023, Jeffares et al.
> - `[10]` "T2G-FORMER: Organizing Tabular Features into Relation Graphs Promotes Heterogeneous Feature Interaction", AAAI 2023, Yan et al.
> - `[11]` "XGBoost: A Scalable Tree Boosting System", KDD 2016, Chen et al.
> - `[12]` "Deep Neural Networks and Tabular Data: A Survey"
> - `[13]` "A Performance-Driven Benchmark for Feature Selection in Tabular Deep Learning", NeurIPS 2023 Datasets and Benchmarks, Cherepanova et al.
> - `[14]` "When Do Neural Nets Outperform Boosted Trees on Tabular Data?", NeurIPS 2023 Datasets and Benchmarks, McElfresh et al.

---

> ### Comment · Reviewer_EnVq · 2023-12-05
>
> Thank you for your responses, which have partially addressed my concerns.
>
> 1. However, I find it puzzling why the hyperparameter tuning for T2G-Former did not complete on time, and other approaches were able to finish. The architecture of T2G-Former is very similar to FTT, and their training schemes are also similar. Nevertheless, this seems to be a minor issue, and I trust the author can provide a satisfactory clarification in their final version. Even in such a situation, deeming a model as the worst in performance solely because hyperparameter tuning was not completed is unfair. Even without completing the scheduled tuning iterations, a model may still achieve good performances, not necessarily the worst. Therefore, comparing with the state-of-the-art in this way is unreasonable and the results are not convincing.
>
> 2. This method exhibits inefficiency in training. While the author attempts to justify its use without the need of hyperparameter tuning, comparing the time cost of TabR (without hyperparameter tuning) to other works undergoing hyperparameter tuning seems unfair. There are several competitive deep learning models that do not require hyperparameter tuning and perform well (the comparison with sota has not been executed successfully). Moreover, CatBoost often achieves good performance with fewer iterations compared to XGBoost. I still feel hesitant about using this approach in practice.
>
> I would like to keep my scores.

---

### Official Review · Reviewer_Ly5o · 2023-10-22

**Soundness:** 3 good
**Presentation:** 3 good
**Contribution:** 3 good
**Rating:** 6
**Confidence:** 3

**Summary:**

The paper introduces TabR, a retrieval-augmented tabular deep learning model that outperforms gradient-boosted decision trees (GBDT) on various datasets. TabR incorporates a novel retrieval module that is similar to the attention mechanism, which helps the model achieve the best average performance among tabular deep learning models and is more efficient compared to prior retrieval-based models.

**Strengths:**

1. TabR demonstrates superior performance compared to GBDT and other retrieval-based tabular deep learning models on multiple datasets.
2. The new similarity module in TabR has a reasonable intuitive motivation, allowing it to find and exploit natural hints in the data for better predictions.

**Weaknesses:**

1. Some aspects are not clear, see the questions section.

**Questions:**

1.  What's the reason for choosing m to be 96? How does m affect the performance of TabR?
2.  What's the inference efficiency of TabR and how does it compare with other baselines (e.g., GBDT)?
3.  Is TabR applicable to categorical features? It seems like the paper only considers continuous features.

---

> ### Author Response · Authors · 2023-11-18
> **Rebuttal (part 1/2)**
>
> We thank the reviewer for the feedback!
>
> > What's the reason for choosing m to be 96? How does m affect the performance of TabR?
>
> **In the new PDF, this is addressed in Section A.6.** Below, we provide a summary for convenience and an explanation for the obtained results.
>
> **First,** to evaluate how performance depends on $m$, we consider TabR-S with default hyperparameters and, for each value of $m$, we report the average rank of TabR-S on the default benchmark (Table 1 of the main text). For each `(dataset, m)` pair, the performance is computed as the average over five random seeds.
>
> The choice of $m$  must be made based on the *validation* metrics (not on the *test* metrics), so we provide the ranks on the *validation* sets first:
>
> |     |  m=1 | m=2  | m=4  | m=8  | m=16 | m=32 | m=64 | m=96  | m=128 | m=256 |
> |-----|------|------|------|------|------|------|------|-------|-------| ------|
> | avg | 5    | 4.5  | 4    | 3.25 | 2.75 | 2.12 | 2.12 | 1.88  | 1.88  | 1.88 |
> | std | 2.45 | 1.87 | 1.73 | 1.71 | 1.56 | 1.45 | 1.45 | 0.93  | 1.05  | 1.36 |
>
> It seems that the m=96 turns out to be a reasonable default choice.
>
> Additionally, here are the ranks on the *test* sets:
>
> |     | m=1 | m=2  | m=4  | m=8 | m=16 | m=32 | m=64 | m=96 | m=128 | m=256 |
> |-----|------|------|-----|------|------|------|------|-------|-------|-----|
> | avg | 4.12 | 3.62 | 3.5 | 3.12 | 2.38 | 2.12 | 2    | 1.62  | 1.75  | 1.75 |
> | std | 2.2  | 1.8  | 1.5 | 1.05 | 1.41 | 1.27 | 1.41 | 0.99  | 0.97  | 1.09 |
>
> **Second**, we suggest an explanation for the obtained results. The presence of the softmax function in the retrieval module of TabR gives a hope that the only requirement for $m$ is to be large enough, and softmax will "automatically" choose the optimal value for each sample. That said, in this work, we don't analyse extreme cases like $m = |I_{train}|$ and recommend values like 96 as a starting point.
>
> > What's the inference efficiency of TabR and how does it compare with other baselines (e.g., GBDT)?
>
> **In the new PDF, this is addressed in Section A.4.2.** For convenience, here, we provide a summary.
>
> Below, we report the inference throughput of TabR and XGBoost.
>
> (The technical setup:
> - XGBoost and TabR-S with tuned hyperparameters as in Table 4 of the main text
> - Computation is performed on NVIDIA 2080 Ti.
> - For both models, objects are passed by batches of 4096 objects.)
>
> **The key observations:**
> - On the considered datasets, the throughputs of TabR and XGBoost are mostly comparable.
> - **Important**: our implementation of TabR is naive and lacks even basic optimizations.
>
> |                                | CH    | CA   | HO   | AD   | DI   | OT   | HI  | BL   | WE   | CO   | MI   |
> |--------------------------------|------|------|------|------|------|-----|------|------|------|------|------|
> | `#trainingObjects`                | 6400 | 13209 | 14581 | 26048 | 34521 | 39601 | 62751 | 106764 | 296554 | 371847 | 723412 |
> | `#features`                      | 11    | 8    | 16   | 14   | 9    | 93  | 28   | 9    |  119 | 54   | 136 |
> | `n_estimators` in XGBoost         | 121   | 3997 | 1328 | 988  | 802  | 524 | 1040 | 1751 | 3999 | 1258 | 3814 |
> | `max_depth` in XGBoost            | 5     | 9    | 7    | 10   | 13   | 13  | 11   | 8    | 13   | 12   | 12   |
> | XGBoost throughput (obj./sec.)  | 2197k | 33k  | 179k | 131k | 417k | 19k | 72k  | 84k  | 15k  | 10k  | 14k  |
> | TabR-S throughput (obj./sec.)  | 35k   | 35k  | 55k  | 33k  | 43k  | 40k | 37k  | 27k  | 34k  | 23k  | 11k  |
> | Overhead                        | 62.3  | 0.9  | 3.3  | 3.9  | 9.6  | 0.5 | 1.9  | 3.1  | 0.5  | 0.4  | 1.2  |
>
> **In more detail:**
> - On "non-simple" tasks (CA, OT, WE, CO), TabR is faster ("non-simple" = XGBoost needs many trees AND/OR XGBoost needs high depth AND/OR a dataset has more features).
> - On "simple" tasks, XGBoost is faster ("simple" = XGBoost is shallow AND/OR dataset has few features).
> - With the growth of training size (MI), TabR may become slower because of the retrieval, however, there is *a lot* of room for optimizations (caching candidate representations instead of recomputing them on each forward pass; using float16 instead of the current float32; doing approximate search instead of the current brute force; using only a subset of the training data as candidates, etc.)
>
> (P.S. in the paper, we also analyze *training* efficiency in Table 10)

---

> ### Author Response · Authors · 2023-11-18
> **Rebuttal (part 2/2)**
>
> > It seems like the paper only considers continuous features
>
> **In fact, the used datasets include both continuous and categorical features**:
> - For the default benchmark, Table 1 in the main text provides information on how many features of each type is presented in each dataset.
> - The benchmark from `[1]` also includes many datasets with categorical features, please, see the original paper for details.
>
> > Is TabR applicable to categorical features?
>
> **Yes, TabR is applicable to all kinds of features:** its input encoder consists of conventional common modules, as explained in the caption of Figure 3. In particular, categorical features are encoded with the one-hot encoding, and, if needed, one can use other encoding/embedding schemes.
>
> **References**
>
> - `[1]` "Why do tree-based models still outperform deep learning on tabular data?", NeurIPS 2022 Datasets and Benchmarks, Grinsztajn et al.

---

> ### Author Response · Authors · 2023-11-23
> **Inference efficiency: extended analysis**
>
> Dear reviewer,
>
> In our "Rebuttal (part 1/2)" reply, we provided some analysis on the inference efficiency.
>
> Now, we are excited to share an **extended study on the inference throughput covering SEVEN models in TWO modes.** We did not update the PDF document with the new content, but we are committed to do that based on feedback.
>
> **The main observations**:
> - TabR is faster than prior retrieval-based tabular DL (SAINT).
> - TabR is slower than simple retrieval-free models.
> - Notably, TabR is only moderately slower than FT-Transformer (a non-retrieval model).
> - *(not reflected in the tables below)* On large dataset, where the retrieval becomes the bottleneck, TabR can be made *significantly* faster (e.g. by order of magnitude) by using approximate nearest neighbor search instead of the current brute force.
>
> Technical details:
> - We did *not* use DL-specific optimizations like `torch.compile`, mixed precision computation, pruning and other techniques.
> - Models with tuned hyperparameters are used (that is, from Table 3 and Table 4). For FT-Transformer, the tuned hyperparameters were taken from `[1]` and `[2]` for most datasets, and the default hyperparemeters were used on two datasets.
>
> **The first mode: "Online predictions"**
> - Device: CPU Intel Core i7-7800X 3.50GHz
> - Thread count: 1
> - Batch size: 1
>
> *Notation*
> - *Values are aggregated over the 11 datasets from Table 1 (the "default" benchmark)*
> - *(A) ~ Absolute throughput (objects per second)*
> - *(R) ~ Relative throughput w.r.t. TabR-S*
>
> | [CPU & Batch size = 1]   | (A) Min   | (A) Median   | (A) Max   | (R) Min   |   (R) Median |   (R) Max |
> |------------------------|:-----------|:--------------|:-----------|:-----------|:--------------|:-----------|
> | TabR-S                 | 31        | 470          | 2.0K      | 1       |         1    |       1   |
> | TabR                   | 26        | 236          | 1.2K      | 0.2       |         0.5  |       1.3 |
> | SAINT                  | < 1       | 15           | 59        | < 0.01    |         0.02 |       0.1 |
> | XGBoost                | 270       | 2.2K         | 10K       | 0.6       |         5.5  |      17.5 |
> | MLP                    | 2.6K      | 4.8K         | 15K       | 4.0       |        13.9  |     387.1 |
> | MLP-PLR                | 532       | 1.6K         | 5.1K      | 0.9       |         5.2  |      47.2 |
> | FT-Transformer         | 219       | 694          | 1.7K      | 0.4       |         1.6  |      11.6 |
>
> **The second mode: "Offline batch processing"**
> - Device: GPU NVIDIA 2080Ti 12GB
> - Batch size: the largest possible batch size for a given model (but no larger then `2 ** 17 ~= 128_000`)
>
> It turns out that XGBoost can achieve better numbers in this mode than with the fixed batch size 4096.
>
> *Notation*
> - *Values are aggregated over the 11 datasets from Table 1 (the "default" benchmark)*
> - *(A) ~ Absolute throughput (objects per second)*
> - *(R) ~ Relative throughput w.r.t. TabR-S*
>
> | [CUDA & Batch size = max]   | (A) Min   | (A) Median   | (A) Max   |   (R) Min |   (R) Median |   (R) Max |
> |---------------------------|:-----------|:--------------|:-----------|:-----------|:--------------|:-----------|
> | TabR-S                    | 24K       | 123K         | 319K      |      1    |          1   |       1   |
> | TabR                      | 28K       | 91K          | 308K      |      0.4  |          0.8 |       2.5 |
> | SAINT                     | 2.1K      | 32K          | 133K      |      0.05 |          0.3 |       0.8 |
> | XGBoost                   | 63K       | 1.4M         | 16.6M     |      0.9  |         11.4 |     126.8 |
> | MLP                       | 3.2M      | 7.7M         | 47.8M     |     23.9  |         67.4 |     512   |
> | MLP-PLR                   | 63K       | 1.1M         | 8.5M      |      0.5  |          6.7 |     101.7 |
> | FT-Transformer            | 21K       | 198K         | 752K      |      0.2  |          2   |       4.5 |
>
> **References**
>
> - `[1]` "Revisiting Deep Learning Models for Tabular Data", Gorishniy et al.
> - `[2]` "On Embeddings for Numerical Features in Tabular Deep Learning", Gorishniy et al.

---

### Official Review · Reviewer_fJsx · 2023-11-01

**Soundness:** 4 excellent
**Presentation:** 4 excellent
**Contribution:** 2 fair
**Rating:** 6
**Confidence:** 2

**Summary:**

This work proposes a retrieval-augmented deep learning architecture for tabular regression/classification. The model passes $x$, the row to be classified/predicted, as well as additional retrieval context rows, through a learned encoder. TabR then retrieves the rows most similar to the encoded form of $x$, where similarity is defined as the Euclidean distance between the encoded versions of two rows, mapped through a linear layer. The top retrieval candidates and their respective labels are then sent through some more learned transformations before being aggregated and combined with the encoded form of the row to be classified/regressed. This combined embedding goes through more MLP layers to result in the output.

The paper goes through variants of the architecture and how each respective change impacts performance. It then compares against other deep learning-based models as well as gradient boosted decision trees. In both default-hyperparameter and tuned-hyperparameter settings, TabR performs well.

**Strengths:**

1. The extensive amount of open-sourcing and experiment reproducibility is greatly appreciated.
1. Strong results relative to both deep learning and boosted tree methods, and TabR-S's relatively strong performance relative to out-of-the-box boosted tree libraries suggests this isn't just excessive parameter tweaking and overfitting via architecture search.
1. Easy to read, with key pieces of information generally emphasized appropriately.

**Weaknesses:**

1. Paper doesn't go into detail describing differences with prior deep learning-based tabular methods. What might explain the performance differences? Ex. "prior work, where several layers with multi-head attention between objects and features are often used" but was this what led to retrieval's low benefit in the past?
1. Insufficient discussion of categorical variables. Is accuracy or training time particularly affected by their relative abundance relative to numerical features?
1. The steps of Section 3.2 seem rather arbitrary. Some of the detail could be compressed to make room for more intuition why the final architecture makes more sense (content from A.1.1). Description of architectural changes that didn't work would also be very insightful.
1. Paper describes training times in A.4, but I believe a summary of this is important enough to warrant inclusion in the main paper. Something like a mention of the geometric mean (over the datasets) of the ratio between TabR's training time to a gradient boosted methods, described in the conclusion, would be sufficient. While the ratio is likely >1, it is better to acknowledge this weakness than to hide it.

**Questions:**

See weaknesses. Also, what is $I_{cand}$? Is it all rows of the table that labels have been provided for? It's mentioned in page 3 that "we use the same set of candidates for all input objects" but what it the set of candidates exactly?

---

> ### Author Response · Authors · 2023-11-18
> **Rebuttal (part 1/2)**
>
> We thank the reviewer for the comments and questions!
>
> > What might explain the performance differences? ... but was this what led to retrieval's low benefit in the past?
>
> (1) **TL;DR: yes, the main reason behind the difference in the performance lies in the retrieval module designed in Section 3.2.** A quick summary of the main differences in the retrieval-related functionality:
> - Prior work: *multiple multi-head vanilla* attention modules, where the attention is performed *between objects and features*.
> - TabR: *one single-head custom* attention-like module, where the attention is performed *only between objects*.
>
> In the new PDF, in Section 2 on Page 3, we improved the wording and the formatting to better contrast TabR against prior solutions.
>
> (2) **The relevant parts from the original PDF** (here, we uppercased the key pieces for convenience):
>
> - (Paragraph "Step 2" Page 5, where we improved the similarity module commonly used in prior work) *This change is a turning point in our story, WHICH WAS OVERLOOKED IN PRIOR WORK.*
> - (abstract) *"NOVEL FINDINGS ... LIE IN THE ATTENTION-LIKE MECHANISM that is responsible for retrieving the nearest neighbors and extracting valuable signal from them."*
> - (Section 2 Page 3) *"Compared to prior work, where SEVERAL LAYERS WITH MULTI-HEAD ATTENTION ... TabR implements its retrieval component with just ONE SINGLE-HEAD ATTENTION-LIKE MODULE. Importantly ... module of TabR is CUSTOMIZED in a way that makes it BETTER SUITED FOR TABULAR DATA PROBLEMS"*
> - etc.
>
> > Insufficient discussion of categorical variables.
>
> However, as indicated by Figure 3 and its caption, **TabR is equally capable of handling all feature types, there is no conceptual difference between them.** In particular:
> - Categorical features are encoded with the one-hot encoding (as mentioned in the caption of Figure 3 and in Section D6). This is a simple and efficient operation, and one can switch to any other encoding scheme if needed.
> - Continuous features are normalized and, optionally, transformed with embeddings from `[2]` (which, in fact, makes continuous features more demanding than categorical features in terms of compute).
>
> Just in case, we will mention that **the used datasets include both continuous and categorical features**:
> - The default benchmark: Table 1 provides information on how many features of each type is presented in each dataset.
> - The benchmark from `[1]` also includes many datasets with categorical features.
>
> > Is accuracy or training time particularly affected by their relative abundance relative to numerical features?
>
> We did not notice any different behaviour caused by the presence of categorical features, which is consistent with the explanation provided above (*"... for TabR, there is no conceptual difference between feature types ..."*).
>
> > The steps of Section 3.2 seem rather arbitrary.
>
> **First**, just in case, we provide a quick informal recap of the story behind the design steps in Section 3.2:
> - "Step-0. The vanilla-attention-like baseline" is motivated by prior work.
> - "Step-1. Adding context labels" is a natural attempt to use the available labels of the neighbors. The important outcome: *"the similarity module $\mathcal{S}$ taken from the vanilla attention does not allow benefiting from such a valuable signal as labels"*.
> - "Step-2. Improving the similarity module $\mathcal{S}$" is fully motivated by the quoted outcome of Step-1.
> - "Step-3. Improving the value module $\mathcal{V}$" is inspired by a simple observation that $\mathcal{V}$ can be made more expressive by adding the dependency on the target object's representation $\tilde{x}$.
> - Step-4 is the only purely technical step where we identified better technical defaults for the new architecture.
>
> **Second**, how we improved the communication and made the available content more visible in the new PDF:
> - In Section A.2.1, we added extended motivation behind the design of the value module of TabR.
> - In the "Step-2" and "Step-3" paragraphs, we added references to the extended motivation and analysis available in the appendix.
> - In the "Step-3" paragraph, we improved the wording to highlight the transition of our focus from the similarity module to the value module.

---

> ### Author Response · Authors · 2023-11-18
> **Rebuttal (part 2/2)**
>
> > Paper describes training times in A.4, but I believe a summary of this is important enough to warrant inclusion in the main paper. Something like a mention of the geometric mean (over the datasets) of the ratio between TabR's training time to a gradient boosted methods, described in the conclusion, would be sufficient.
>
> We are excited to discuss the aspect of efficiency in more detail!
>
> Regarding the suggestion to announce Table 10 (training times), currently, we achieve that and prepare readers for Table 10 by:
>
> - repeatedly mentioning two things throughout the paper:
>     - TabR is more efficient than prior retrieval-based tabular DL.
>     - TabR is less efficient than simple models and may require special considerations on larger datasets.
> - directly mentioning Section A.4 and Section A.4.1 multiple times.
>
> **Importantly**, we believe that Table 10 (training times) needs a full-fledged presentation and nuanced discussion, and it may suffer from simplifications. In particular, announcing Table 10 with a single number can be misleading: for example, the suggested geometric mean of the ratios will completely hide absolute training times, which is what actually matters in practice. To avoid such effects, we report specific numbers for both "positive" (e.g. order(s) of magnitude improvements over prior work) and "negative" efficiency-related stories only in Table 10.
>
> >  it is better to acknowledge this weakness than to hide it.
>
> Based on the content from the original submission, let us illustrate that, in fact, **we care deeply about efficiency, invest heavily in this aspect of our work, and strive for complete transparency in this matter.** We truly consider the storyline about efficiency an important part of our paper. In particular, throughout the work, we repeatedly mention two things:
> - TabR is more efficient than prior retrieval-based tabular DL.
> - TabR is less efficient than simple models and may require special considerations on larger datasets.
>     - (The end of Introduction) *"Tree-based models, in turn, remain a cheaper solution"*
>     - **The whole Section 5.1 is dedicated to improving training times of TabR**.
>     - Even the seemingly unrelated Section 5.2 mentions that the continual updates can be used to make TabR train faster: *"Additionally, this approach can be used to scale TabR to large datasets by training the model on a subset of data and retrieving from the full data."*
>     - (Conclusion) *"An important direction for future work is improving the efficiency of retrieval-augmented models to make them faster in general and in particular applicable to tens and hundreds of millions of data points"*
>     - (Limitations on Page 6 refer to Section B, where we write:) *"the retrieval module R still causes overhead compared to purely parametric models, so TabR may not scale to truly large datasets as-is"*
>
> > Also, what is $I_{cand}$? Is it all rows of the table that labels have been provided for?
>
> 1. **In most of the experiments, yes**: the neighbors are retrieved from all training objects, which formally means $I_{cand} = I_{train}$.
> 2. Strictly speaking, we implicitly rely on the fact that  $I_{cand}$ can be different from $I_{train}$ in the following places:
>     - In Section 5.1, for *training* objects,  $I_{cand} = I_{train}$ is true only until the context freeze, after which the retrieval for *training* objects is not performed (instead, the same neighbors are reused until the end of training).
>     - In Section 5.2, for *test* objects, $I_{cand} = I_{train}$ is true only until new labeled objects are added to the set of candidates.
>     - The *"Second, depending on an application ..."* paragraph of Section B.
>
> > It's mentioned in page 3 that "we use the same set of candidates for all input objects"
>
> In the new PDF, we improved the wording, now it says: *"In this work, unless otherwise noted, we use the same set of candidates for all input objects and set $I_{cand} = I_{train}$ (which means retrieving from all training objects)"*.
>
> **References**
> - `[1]` "Why do tree-based models still outperform deep learning on tabular data?", NeurIPS 2022 Datasets and Benchmarks, Grinsztajn et al.
> - `[2]` "On Embeddings for Numerical Features in Tabular Deep Learning", NeurIPS 2022, Gorishniy et al.

---

### Official Review · Reviewer_aaV3 · 2023-11-01

**Soundness:** 3 good
**Presentation:** 3 good
**Contribution:** 3 good
**Rating:** 8
**Confidence:** 3

**Summary:**

This paper considers the problem of making predictions on tabular data. The authors propose a retrieval-augmented approach where a predictor takes the representation not of the table being predicted but also the representation of the nearest neighbors from a training dataset. The encoding representations and the predictors are training together and use straightforward architecture architectures. The main result is that a combination of the carefully crafted techniques outperforms GBDT on an ensemble of tasks. The training time is higher than GBDT but not unreasonable, and better compared to prior deep learning methods. The prediction times are better

**Strengths:**

1. The results seem to be a significant advance over prior work in tabular data predictions. In particular, the first deep learning model to outperform GBDT on an ensemble of datasets.
2. The experiments and analysis are quite extensive. Multiple datasets of different kinds of data, analysis of training and prediction times.
3. Clear articulation of which techniques helped. the techniques are overall not too complex.

**Weaknesses:**

A comparison of the inference and query complexity between the methods is lacking.

**Questions:**

1. Inference time and compexity -- are the studies based on normalized inference time between models? If not, could you comment more? How does the inference complexity depend on the size of the table data?

2. Could a different selection of datasets prove that the tabR is not superior to GBDT? In other words, are these datasets highly representative?

3. Is it not surprising that Step-1 (adding context labels) did not help that much? One would guess that this is a big component of signal in retrieval augmentation.

4. Not a question, but the methodology here reminds one of extreme classification and specifically this paper. https://arxiv.org/abs/2207.04452

---

> ### Author Response · Authors · 2023-11-18
> **Rebuttal (part 1/2)**
>
> We thank the reviewer for the positive feedback!
>
> > A comparison of the inference and query complexity between the methods is lacking.
>
> **In the new PDF, this is addressed in Section A.4.2.** For convenience, here, we provide a summary.
>
> Below, we report the inference throughput of TabR and XGBoost.
>
> (The technical setup:
> - XGBoost and TabR-S with tuned hyperparameters as in Table 4 of the main text
> - Computation is performed on NVIDIA 2080 Ti.
> - For both models, objects are passed by batches of 4096 objects.)
>
> **The key observations:**
> - On the considered datasets, the throughputs of TabR and XGBoost are mostly comparable.
> - **Important**: our implementation of TabR is naive and lacks even basic optimizations.
>
> |                                | CH    | CA   | HO   | AD   | DI   | OT   | HI  | BL   | WE   | CO   | MI   |
> |--------------------------------|------|------|------|------|------|-----|------|------|------|------|------|
> | `#trainingObjects`                | 6400 | 13209 | 14581 | 26048 | 34521 | 39601 | 62751 | 106764 | 296554 | 371847 | 723412 |
> | `#features`                      | 11    | 8    | 16   | 14   | 9    | 93  | 28   | 9    |  119 | 54   | 136 |
> | `n_estimators` in XGBoost         | 121   | 3997 | 1328 | 988  | 802  | 524 | 1040 | 1751 | 3999 | 1258 | 3814 |
> | `max_depth` in XGBoost            | 5     | 9    | 7    | 10   | 13   | 13  | 11   | 8    | 13   | 12   | 12   |
> | XGBoost throughput (obj./sec.)  | 2197k | 33k  | 179k | 131k | 417k | 19k | 72k  | 84k  | 15k  | 10k  | 14k  |
> | TabR-S throughput (obj./sec.)  | 35k   | 35k  | 55k  | 33k  | 43k  | 40k | 37k  | 27k  | 34k  | 23k  | 11k  |
> | Overhead                        | 62.3  | 0.9  | 3.3  | 3.9  | 9.6  | 0.5 | 1.9  | 3.1  | 0.5  | 0.4  | 1.2  |
>
> **In more detail:**
> - On "non-simple" tasks (CA, OT, WE, CO), TabR is faster ("non-simple" = XGBoost needs many trees AND/OR XGBoost needs high depth AND/OR a dataset has more features).
> - On "simple" tasks, XGBoost is faster ("simple" = XGBoost is shallow AND/OR dataset has few features).
> - With the growth of training size (MI), TabR may become slower because of the retrieval, however, there is *a lot* of room for optimizations (caching candidate representations instead of recomputing them on each forward pass; using float16 instead of the current float32; doing approximate search instead of the current brute force; using only a subset of the training data as candidates, etc.)
>
> > Inference time and compexity -- are the studies based on normalized inference time between models? If not, could you comment more?
>
> Strictly speaking, the results reported in all main tables are obtained without fixing the inference time budget: all methods can spend any time they need to make predictions. *However,* generally, the efficiency of TabR vs. prior work is an important storyline of our paper with significant wins over prior work, and, in the considered scope of datasets, the inference time of TabR is always reasonable (as indicated by the above table).
>
> > How does the inference complexity depend on the size of the table data?
>
> As indicated by the above table, for datasets of sizes up to (roughly) 500K objects, for TabR-S, the dataset size is not a major factor defining the inference throughput. For larger datasets, the dataset size becomes a more important factor (formally, for the current brute force search, the search complexity grows linearly with the dataset size). Luckily, the current implementation of TabR has a lot of room for simple optimizations:
> - caching candidate key representations instead of recomputing them on each forward pass.
> - performing the search in float16 instead of the current float32.
> - performing approximate similarity search instead of the current brute force.
> - using only a subset of the training data as candidates.
> - etc.

---

> > ### Comment · Reviewer_aaV3 · 2023-11-20
> > **inference time comparison**
> >
> > Is it reasonable to compare XGBoost with deep learning with 4096 batch size on GPU. How do the inference times compare on CPU or with small batching as is the case with inference in reality?

---

> > > ### Author Response · Authors · 2023-11-23
> > > **Inference efficiency: extended analysis**
> > >
> > > We thank the reviewer for the questions!
> > >
> > > We are excited to present an **extended study on the inference throughput covering SEVEN models in TWO modes.** We did not update the PDF document with the new content, but we are committed to do that based on feedback.
> > >
> > > **The main observations**:
> > > - TabR is faster than prior retrieval-based tabular DL (SAINT).
> > > - TabR is slower than simple retrieval-free models.
> > > - Notably, TabR is only moderately slower than FT-Transformer (a non-retrieval model).
> > > - *(not reflected in the tables below)* On large dataset, where the retrieval becomes the bottleneck, TabR can be made *significantly* faster (e.g. by order of magnitude) by using approximate nearest neighbor search instead of the current brute force.
> > >
> > > Technical details:
> > > - We did *not* use DL-specific optimizations like `torch.compile`, mixed precision computation, pruning and other techniques.
> > > - Models with tuned hyperparameters are used (that is, from Table 3 and Table 4). For FT-Transformer, the tuned hyperparameters were taken from `[1]` and `[2]` for most datasets, and the default hyperparemeters were used on two datasets.
> > >
> > > **The first mode: "Online predictions"**
> > > - Device: CPU Intel Core i7-7800X 3.50GHz
> > > - Thread count: 1
> > > - Batch size: 1
> > >
> > > *Notation*
> > > - *Values are aggregated over the 11 datasets from Table 1 (the "default" benchmark)*
> > > - *(A) ~ Absolute throughput (objects per second)*
> > > - *(R) ~ Relative throughput w.r.t. TabR-S*
> > >
> > > | [CPU & Batch size = 1]   | (A) Min   | (A) Median   | (A) Max   | (R) Min   |   (R) Median |   (R) Max |
> > > |------------------------|:-----------|:--------------|:-----------|:-----------|:--------------|:-----------|
> > > | TabR-S                 | 31        | 470          | 2.0K      | 1       |         1    |       1   |
> > > | TabR                   | 26        | 236          | 1.2K      | 0.2       |         0.5  |       1.3 |
> > > | SAINT                  | < 1       | 15           | 59        | < 0.01    |         0.02 |       0.1 |
> > > | XGBoost                | 270       | 2.2K         | 10K       | 0.6       |         5.5  |      17.5 |
> > > | MLP                    | 2.6K      | 4.8K         | 15K       | 4.0       |        13.9  |     387.1 |
> > > | MLP-PLR                | 532       | 1.6K         | 5.1K      | 0.9       |         5.2  |      47.2 |
> > > | FT-Transformer         | 219       | 694          | 1.7K      | 0.4       |         1.6  |      11.6 |
> > >
> > > **The second mode: "Offline batch processing"**
> > > - Device: GPU NVIDIA 2080Ti 12GB
> > > - Batch size: the largest possible batch size for a given model (but no larger then `2 ** 17 ~= 128_000`)
> > >
> > > It turns out that XGBoost can achieve better numbers in this mode than with the fixed batch size 4096.
> > >
> > > *Notation*
> > > - *Values are aggregated over the 11 datasets from Table 1 (the "default" benchmark)*
> > > - *(A) ~ Absolute throughput (objects per second)*
> > > - *(R) ~ Relative throughput w.r.t. TabR-S*
> > >
> > > | [CUDA & Batch size = max]   | (A) Min   | (A) Median   | (A) Max   |   (R) Min |   (R) Median |   (R) Max |
> > > |---------------------------|:-----------|:--------------|:-----------|:-----------|:--------------|:-----------|
> > > | TabR-S                    | 24K       | 123K         | 319K      |      1    |          1   |       1   |
> > > | TabR                      | 28K       | 91K          | 308K      |      0.4  |          0.8 |       2.5 |
> > > | SAINT                     | 2.1K      | 32K          | 133K      |      0.05 |          0.3 |       0.8 |
> > > | XGBoost                   | 63K       | 1.4M         | 16.6M     |      0.9  |         11.4 |     126.8 |
> > > | MLP                       | 3.2M      | 7.7M         | 47.8M     |     23.9  |         67.4 |     512   |
> > > | MLP-PLR                   | 63K       | 1.1M         | 8.5M      |      0.5  |          6.7 |     101.7 |
> > > | FT-Transformer            | 21K       | 198K         | 752K      |      0.2  |          2   |       4.5 |
> > >
> > > **References**
> > >
> > > - `[1]` "Revisiting Deep Learning Models for Tabular Data", Gorishniy et al.
> > > - `[2]` "On Embeddings for Numerical Features in Tabular Deep Learning", Gorishniy et al.

---

> ### Author Response · Authors · 2023-11-18
> **Rebuttal (part 2/2)**
>
> > are these datasets highly representative?
>
> We did our best to build a representative benchmark using **~50 datasets from prior work**.
> In particular, we ensured that there are **many challenging tasks that favour GBDT over traditional neural networks.** Examples:
> - We use the widely cited `[1]` where GBDT was shown to be superior to neural networks.
> - We use some datasets from `[2]` including multiple GBDT-friendly tasks.
> - We also use the real world large scale dataset for weather prediction from `[3]`.
>
> To put our work in the context of prior work, here are the main properties of the benchmarks used in some prior work on tabular deep learning:
>
> | Model `[citation]`   | The number of datasets used in the paper                                |
> | -------------------- | ----------------------------------------------------------------------- |
> | **TabR** `[ours]`    | **~50 datasets (roughly, `[1]` + some datasets from `[2]` and `[3]`)**  |
> | NODE `[7]`           | 6 datasets                                                              |
> | TabNet  `[5]`        | <10 datasets                                                            |
> | FT-Transformer `[4]` | 11 datasets                                                             |
> | SAINT `[6]`          | 30 datasets                                                             |
> | \<survey\> `[11]`    | 5 datasets                                                              |
> | MLP-PLR `[2]`        | 11 datasets                                                             |
> | TabPFN `[8]`         | 67 small datasets (different scope)                                     |
> | \<benchmark\> `[1]`  | <50 *unique* datasets (>50 with multiple versions of the same datasets) |
> | TANGOS `[9]`         | 20 datasets                                                             |
> | T2G-FORMER `[10]`    | 12 datasets                                                             |
>
>
> > Is it not surprising that Step-1 (adding context labels) did not help that much?
>
> This is discussed in the "Step-1" paragraph on Page 5:
>
> *> "Table 2 shows no improvements from using labels, which is counter-intuitive. Perhaps, the similarity module taken from the vanilla attention does not allow benefiting from such a valuable signal as labels."*
>
> In other words, yes, this is indeed surprising! However, after the subsequent Step-2 in Section 3.2 (and, in particular, after the ablation study in Section A.3), we can confidently say that *to benefit from a valuable signal (such as labels), good similarity module is required.*
>
> **References**
> - `[1]` "Why do tree-based models still outperform deep learning on tabular data?", NeurIPS 2022 Datasets and Benchmarks, Grinsztajn et al.
> - `[2]` "On Embeddings for Numerical Features in Tabular Deep Learning", NeurIPS 2022, Gorishniy et al.
> - `[3]` "Shifts: A Dataset of Real Distributional Shift Across Multiple Large-Scale Tasks", NeurIPS 2021 Datasets and Benchmarks , Malinin et al.
> - `[4]` "Revisiting Deep Learning Models for Tabular Data", NeurIPS 2021, Gorishniy et al.
> - `[5]` "TabNet: Attentive Interpretable Tabular Learning", AAAI 2021, Arik et al.
> - `[6]` "SAINT: Improved Neural Networks for Tabular Data via Row Attention and Contrastive Pre-Training", ICLR 2022 submission, Somepalli et al.
> - `[7]` "Neural Oblivious Decision Ensembles for Deep Learning on Tabular Data", ICLR 2020, Popov et al.
> - `[8]` "TabPFN: A Transformer That Solves Small Tabular Classification Problems in a Second", ICLR 2023, Hollmann et al.
> - `[9]` "TANGOS: Regularizing Tabular Neural Networks through Gradient Orthogonalization and Specialization", ICLR 2023, Jeffares et al.
> - `[10]` "T2G-FORMER: Organizing Tabular Features into Relation Graphs Promotes Heterogeneous Feature Interaction", AAAI 2023, Yan et al.
> - `[11]` "Deep Neural Networks and Tabular Data: A Survey"

---

### Author Response · Authors · 2023-11-18
**Dear Reviewers (rebuttal)**

We thank the reviewers for the feedback!

**In this post**:
- (1) We summarize positive things from the reviews.
- (2) We summarize the changes to the updated PDF document.

**In the individual replies**, we address other comments.

### (1) Positive things

- **Strong empirical performance of the proposed method**
    - `aaV3`: *"significant advance over prior work ... the first deep learning model to outperform GBDT on an ensemble of datasets."*
    - `Ly5o`: *"TabR demonstrates superior performance compared to GBDT and other retrieval-based tabular deep learning models ..."*
    - `EnVq`: *"method has managed to outperform tree based models like XGBoost on middle-scale datasets. ... This model is the best-performing retrieval based model."*
    - `fJsx`: *"Strong results relative to both deep learning and boosted tree methods ... TabR-S's relatively strong performance relative to out-of-the-box boosted tree libraries"*
- **The proposed method**
    - `fJsx`: *"this [strong performance] suggests this isn't just excessive parameter tweaking and overfitting via architecture search."*
    - `Ly5o`: *"The new similarity module in TabR has a reasonable intuitive motivation, allowing it to find and exploit natural hints in the data for better predictions."*
    - `aaV3`: *"the techniques are overall not too complex."*
- **Experimental setup**
    - `EnVq`: *"the experiments are comprehensive"*
    - `aaV3`: *"The experiments and analysis are quite extensive. Multiple datasets of different kinds of data ..."*
- **Presentation**
    - `fJsx`: *"Easy to read, with key pieces of information generally emphasized appropriately."*
    - `aaV3`: *"Clear articulation of which techniques helped."*
    - `EnVq`: *"Overall, the presentation is clear ... The details are clear ..."*
- **Reproducibility**
    - `fJsx`: *"The extensive amount of open-sourcing and experiment reproducibility is greatly appreciated."*
    - `EnVq`: *"the model is highly reproducible"*

### (2) Changes to the PDF

**Main text**

Only small fixes and wording improvements:

- `[fJsx]` (Section 2) An improved sentence contrasting our model TabR against prior solutions (the *"Retrieval-augmented models for tabular data problems"* paragraph).
- `[fJsx]` (Section 3.1) A small fix to notation.
- `[fJsx]` (Section 3.2) A similar notation-related fix (the "Retrieval module" paragraph).
- `[fJsx, EnVq]` (Section 3.2) An improved sentence for a smoother transition from Step-2 to Step-3 (the "Step-3" paragraph).
- `[EnVq]` (Section 3.1) Fixed the typo (the "Notation" paragraph).
- `[EnVq]` (Section 3.2) Point to the available analysis in Appendix (the "Step-2" and "Step-3" paragraphs).
- `[EnVq]` (Section 2) Added citations of TANGOS `[1]` and TabPFN `[2]` (the "Parametric deep learning models" paragraph was renamed to "Tabular deep learning").

**Appendix**

The content added based on the reviews:

- `[Ly5o]` (Section A.6) The ablation study for the context size $m=96$ used in the paper.
- `[EnVq]` (Section A.2.1) The extended motivation for the value module $\mathcal{V}$ (and, for the sake of symmetry, we changed to name of Section A.1.1 to "Motivation").
- `[EnVq]` (a new bullet in Section A.8) The explanation for why "LinearWithoutBias" is used in Equation 4.
- `[aaV3, Ly5o]` (Section A.4.2) The comparison of the inference efficiency of TabR and XGBoost.

**References**

- `[1]` "TANGOS: Regularizing Tabular Neural Networks through Gradient Orthogonalization and Specialization", ICLR 2023, Jeffares et ak.
- `[2]` "TabPFN: A Transformer That Solves Small Tabular Classification Problems in a Second", ICLR 2023, Hollmann et al.

---

### Meta-Review · Area_Chair_6yPy · 2023-12-08

**Metareview:**

This submission contributes a neural architecture dedicated to tabular learning based on combining a feed-forward network with a nearest neighbor mechanism. The submission generated many solid discussions and was seen as an interesting addition to the tabular-learning literature. The reviewers appreciated the extensive experiments, the writing, and the reproducibility of the work. More baselines could have been added, and more attention to categorical variables.

**Justification For Why Not Higher Score:**

The paper is already borderline with regards to acceptance. I do not think that we can push it further.

**Justification For Why Not Lower Score:**

The work seems solid, as acknowledged by 3 of the 4 reviewers. The answers to the review by the authors were also solid. The fourth reviewer seems unfair, and is the author of one of the competing methods puts forward in the critical review. The work seems to honestly position itself relative to this prior art.

---

### Decision · Program_Chairs · 2024-01-16

Accept (poster)